# Pessimistic Data Integration for Policy Evaluation

**Xiangkun Wu**[*]
School of Mathematical Sciences
Zhejiang University
Hangzhou, China
12235031@zju.edu.cn

**Ting Li**[*]
School of Statistics and Data Science
Shanghai University of Finance and Economics
Shanghai, China
tingli@mail.shufe.edu.cn

**Gholamali Aminian**
The Alan Turing Institute
London, UK
gaminian@turing.ac.uk

**Armin Behnamnia**
Sharif University of Technology
Tehran, Iran
arminbehnamnia@gmail.com

**Hamid R. Rabiee**
Department of Computer Engineering
Sharif University of Technology
Tehran, Iran
rabiee@sharif.edu

**Chengchun Shi** [†]
Department of Statistics
London School of Economics and Political Science
London, UK
C.Shi7@lse.ac.uk

## Abstract

This paper studies how to integrate historical control data with experimental data to enhance A/B testing, while addressing the distributional shift between historical and experimental datasets. We propose a pessimistic data integration method that combines two causal effect estimators constructed based on experimental and historical datasets. Our main idea is to conceptualize the weight function for this combination as a policy so that existing pessimistic policy learning algorithms are applicable to learn the optimal weight that minimizes the resulting weighted estimator's mean squared error. Additionally, we conduct comprehensive theoretical and empirical analyses to compare our method against various baseline estimators across five scenarios. Both our theoretical and numerical findings demonstrate that the proposed estimator achieves near-optimal performance across all scenarios.

## 1 Introduction

A/B testing is widely used by various technology companies such as Amazon, Google, Netflix, Uber, and Didi to evaluate the performance of new products, policies or treatments compared to existing controls. However, the effectiveness of such evaluations is often limited by short duration of online experiments. For instance, in ridesharing, most experiments last no more than two weeks [1]. Before conducting these experiments, companies usually have access to a substantial amount of historical data collected under the control policy. Recent work has demonstrated that integrating these historical control data with experimental data can largely improve the efficiency of A/B testing [2].

The primary challenge in data integration stems from the distributional shift between historical and experimental data, which can generally be categorized into three types: (i) covariate shift – the changes in the distribution of contextual covariates; (ii) policy shift – the changes in the behavior

---

[*]Equal contribution.
[†]Corresponding author.

39th Conference on Neural Information Processing Systems (NeurIPS 2025).

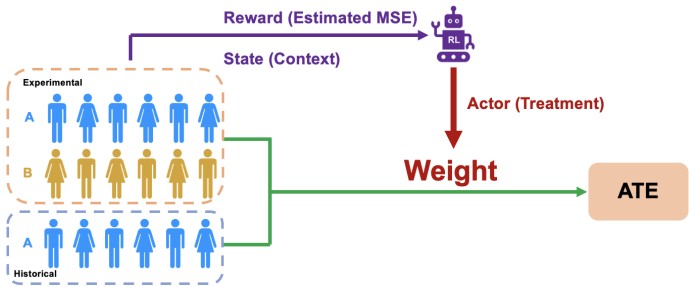

Figure 1: Workflow of the proposed estimator.

policy or propensity score; and (iii) posterior shift – the changes in the outcome distribution given covariates and treatment. Such distributional shifts can substantially bias the resulting treatment effect estimator and hinder the effective use of historical data for A/B testing.

This paper studies the integration of historical data to improve A/B testing, while addressing all three types of distributional shifts simultaneously. Our contributions are as follows.

**Methodologically**, we propose a weighted average treatment effect (ATE) estimator that optimally combines information from both experimental and historical datasets. Our main idea is the development of a pessimistic data integration approach that conceptualizes the weight function for data combination as a policy, which enables existing pessimistic policy learning algorithms to learn this optimal weight function; see Figure 1 for an illustration of our methodology.

**Theoretically**, we conduct a comprehensive comparative analysis to compare our proposed weighted ATE estimator against five baseline estimators across five different scenarios, reflecting differing degrees of posterior shift between experimental and historical datasets, and different levels of heavy-tailedness in reward residuals. Our analysis reveals that while each baseline estimator may perform optimally under certain scenarios, they often fail in others. In contrast, the proposed estimator is adaptive: it satisfies the *oracle* property across all scenarios, meaning that it achieves comparable performance to the optimal scenario-specific estimator, working effectively as if it knew the optimal weight function. Our theories are further supported by synthetic and real-world data analyses. Table 1 summarizes our theoretical and numerical findings, with detailed descriptions of the baseline estimators and scenarios provided in Sections 3.2 and 4.

## 2 Related work

Our paper is closely related to data integration, the pessimistic principle in offline policy learning and off-policy evaluation (OPE), which we elaborate below.

**Data integration**. Data integration is related to various fields in statistics and machine learning, ranging from the classical meta-analysis [3], to more recent advancements in transfer learning [4], and their applications to numerous downstream tasks such as multi-task learning [5], multimodal learning [6, 7], fusion learning [8], individualized treatment regime estimation [9, 10], and RL [11].

Our proposal is particularly related to those methods tailored for causal inference; see [12] and [13] for recent reviews. Based on their approach to handling distributional shifts, these methods can roughly be classified into two categories:

1. The first category addresses the covariate and policy shifts under the assumption of no posterior shift [14–19]. A primary example is given by Li et al. [20], who integrated experimental data with historical controls and developed an estimator that achieves the asymptotically smallest MSE.

2. The second category handles all three types of distributional shifts simultaneously [21–24]. Among these methods, a notable subset applied an $\ell_1$-type penalty function for selecting external data [25, 26]. Recent advancements include federated causal inference approaches [27, 28], which employed such penalization to estimate ATE in a manner that preserves privacy.

Our approach differs from both categories in the following ways. Methodologically, we overcome several critical limitations of the aforementioned methods, resulting in a more flexible and practical

Table 1: MSEs of different ATE estimators across five scenarios. Green indicates that the estimator achieves the oracle property (its MSE is asymptotically equivalent to that of the optimal estimator). Yellow indicates that the estimator may generally have a high MSE but can attain the oracle property in some special cases. Red indicates that the estimator exhibits a generally large MSE.

| Scenarios | EDO | HDB | MVE | CWE | NonPessi | Proposed |
|---|---|---|---|---|---|---|
| (i) Heavy-tailed historical rewards | | | | | | |
| (ii) Heavy-tailed experimental rewards | | | | | | |
| (iii)Small posterior shifts | | | | | | |
| (iv) Moderate posterior shifts | | | | | | |
| (v) Large posterior shifts | | | | | | |

solution. First, unlike the first category of methods which assumes no posterior shift, our proposal accounts for this shift. Second, whereas the second category of methods requires the posterior shift to be either zero or sufficiently large for consistent data selection – failing in intermediate regimes [2] – our method remains robust even with moderate posterior shifts (see Corollary 4).

In terms of applications, our work focuses on combining experimental and historical data for A/B testing, whereas most works either integrate data from multiple treatment centers or trials for meta analysis [29–32], or combine RCT and observational data to handle unmeasured confounding [33–35].

**Pessimistic policy learning**. The pessimistic principle is fundamental to most existing offline policy learning algorithms, which aim to learn an optimal policy from a pre-collected historical dataset. This principle originates from the seminal works of Swaminathan and Joachims [36, 37], who proposed a counterfactual risk minimizing approach that incorporates the uncertainty of a policy's value estimator as a penalty term to learn policies with lower-variance value estimates. It has been widely employed in contextual bandits [38–41], dynamic treatment regimes [42], RL [43–53], and more recently in the training of large language models [see e.g., 54–56] to prevent value function overestimation and encourage the learning policy to stay close to the behavior policy.

A recent proposal by Li et al. [2] applied the pessimistic principle to data integration for policy evaluation. In particular, they proposed to linearly combine policy value estimators computed from experimental and historical datasets using a weighted average. While their approach is closely related to ours, a key difference lies in their use of a fixed weight function for data integration. In contrast, our approach employs a covariate-dependent weighting, leading to a more accurate estimator than theirs, as demonstrated analytically in Sections 4 and empirically in Section 5.

**Off-policy evaluation**. There is a huge literature on OPE in bandits and RL; see [57] and [58] for reviews. The goal of OPE is to estimate the expected outcome of a target policy using offline data collected under a potentially different behavior policy. Existing approaches can be classified into four main categories: (i) Model-based methods that estimate a dynamic model (e.g., a Markov decision process) from offline data and compute the target policy's expected outcome via dynamic programming or Monte Carlo [59–62]; (ii) Direct methods that estimate the expected outcome by learning either a reward or value function from the offline data [63–73]; (iii) Importance sampling (IS) methods that reweight observed rewards using the IS ratio (the density ratio between target and behavior policies) [74–84]; (iv) Doubly or multiply robust methods that combine the estimated reward or value function from direct methods with the IS ratio from IS methods, and require only the reward/value function or the IS ratio to be consistent [85–99].

Many of these approaches have been recently adopted for A/B testing [see e.g., 100–111]. However, these works rely solely on experimental datasets, without leveraging historical datasets to improve policy evaluation.

More recently, [112] proposed a doubly robust (DR) estimator by combining data from multiple experimental studies. Their approach requires the target outcome and covariate distribution to match those in at least one experimental dataset. We also notice that there is an emerging line of research that studied how to improve OPE estimation by strategically combining multiple base OPE estimators to leverage their strengths [see e.g., 38, 113–115]. While similar in spirit to our combination of OPE estimators from experimental and historical datasets, the base estimators in these papers were derived from a single dataset (thus avoiding distributional shifts). On the contrary, we need to address the challenges posed by the distributional shift between experimental and historical datasets.

# 3 Pessimistic data integration

We first introduce two baseline OPE estimators (formally defined in Equations (1) and (2)). We next introduce our proposed estimator, which builds upon these baseline estimators.

## 3.1 Two baseline estimators

Suppose we are given an experimental dataset $\mathcal{D}^{(e)}$ and a historical dataset $\mathcal{D}^{(h)}$. During the experiment, at each time $t$, the decision maker observes certain contextual covariates (e.g., market features), and assigns an action between a new treatment strategy (denoted by 1) and a baseline control (0), resulting in a reward that measures the company's profit at that time. Thus, the experimental data can be summarized as a set of context-action-reward $O^{(e)} = (S^{(e)}, A^{(e)}, R^{(e)})$ triplets, which are assumed to be i.i.d. over time. Similarly, the historical dataset consists of another set of i.i.d. triplets $O^{(h)} = (S^{(h)}, A^{(h)}, R^{(h)})$, but differs in distribution from $\mathcal{D}^{(e)}$ in the following three aspects:

1. **Covariate shift**: the probability mass function of $S^{(e)}$ (denoted by $p_e$) might differ from that of $S_h$ (denoted by $p_h$), leading to the IS ratio $\mu(s) = p_e(s)/p_h(s)$ generally deviating from 1.

2. **Policy shift**: actions in the historical dataset are exclusively generated under the control policy such that $A^{(h)} = 0$ almost surely, whereas actions in the experimental dataset are generated under both the control and the treatment for A/B testing.

3. **Posterior shift**: the reward function $r^{(e)}(a, s) = \mathbb{E}(R^{(e)}|A^{(e)} = a, S^{(e)} = s)$ in the experimental dataset might differ from that in the historical dataset (denoted by $r^{(h)}$).

Our objective lies in estimating the ATE – the difference between the expected reward under the treatment and that under the control, i.e.,

$$\text{ATE} = \mathbb{E}[r^{(e)}(1, S^{(e)}) - r^{(e)}(0, S^{(e)})],$$

using both experimental and historical datasets.

The first baseline estimator for ATE we introduce is the experimental-data-only (EDO) estimator, which uses exclusively the experimental dataset $\mathcal{D}^{(e)}$ to learn the ATE. This estimator is simple to describe: we construct two OPE estimators using $\mathcal{D}^{(e)}$ to estimate the expected outcomes under treatment 1 and 0, respectively, and compute their difference to obtain the ATE estimator. Notice that any OPE method discussed in Section 2 can be applied for estimation.

As a concrete example, consider the IS estimator with the estimating function $\psi_a^{(e)}(O^{(e)}) = \mathbb{I}(A = a)R^{(e)}/\pi(a|S^{(e)})$ where $\mathbb{I}(A = a)/\pi(a|S)$ denotes the IS ratio of the target policy over the behavior policy (i.e., propensity score) $\pi$ in the experimental data. Using the change of measure theorem, it can be shown that $\psi_a^{(e)}(O^{(e)})$ is unbiased to the expected outcome under treatment $a$. This motivates the use of $\mathbb{E}_n[\psi_a^{(e)}(O^{(e)})]$ to estimate this expected outcome, leading to the following EDO estimator for the ATE,

$$\text{EDO} = \mathbb{E}_n[\psi_1^{(e)}(O^{(e)})] - \mathbb{E}_n[\psi_0^{(e)}(O^{(e)})], \tag{1}$$

where $\mathbb{E}_n$ denotes the empirical average over the offline dataset.

The second baseline estimator is the historical-data-based (HDB) estimator. Similar to EDO, it uses $\mathbb{E}_n[\psi_1^{(e)}(O^{(e)})]$ to estimate the expected outcome under the new treatment. For the control policy, the corresponding estimator $\mathbb{E}_n[\psi_0^{(h)}(O^{(h)})]$ with $\psi_0^{(h)}(O^{(h)}) = \mu(S^{(h)})R^{(h)}$ is constructed using solely the historical data. Here, the IS ratio $\mu(\cdot)$ denotes the density ratio of the probability mass/density function of $S^{(e)}$ over that of $S^{(e)}$. It depends only on the contextual variable, since the historical data is exclusively generated under the control policy, leading to a propensity score $\pi(A^{(h)}|S^{(h)})$ of 1 almost surely. Similarly, it can be shown that this estimator is unbiased to $\mathbb{E}[r^{(h)}(0, S^{(h)})]$. In summary, we have

$$\text{HDB} = \mathbb{E}_n[\psi_1^{(e)}(O^{(e)})] - \mathbb{E}_n[\psi^{(h)}(O^{(h)})]. \tag{2}$$

To conclude this section, we remark that there is a bias-variance trade-off between the two estimators. Specifically, the HDB estimator is generally biased due to the incorporation of historical data.

Table 2: A numerical example demonstrating the bias–variance trade-off (see Appendix A for details). As shown, EDO achieves the lowest bias, while HDB attains the lowest variance. The proposed estimator strikes a balance between bias and variance, resulting in the lowest overall MSE.

| Method | MSE( 95% CI ) | Bias( 95% CI ) | Variance( 95% CI ) |
|--------|---------------|----------------|--------------------|
| EDO | 1.701 (1.598–1.804) | **0.007** (-0.064–0.051) | 1.701 (1.598–1.804) |
| HDB | 2.372 (2.289–2.455) | 1.400 (1.372–1.428) | **0.413** (0.388–0.438) |
| Proposed | **1.394** (1.312–1.476) | 0.221 (0.170–0.272) | 1.345 (1.262–1.428) |

Although it addresses the covariate shift through the use of the IS ratio $\mu$, the posterior shift from the experimental data is extremely challenging to correct, resulting in a non-negligible bias. In contrast, the EDO estimator, derived exclusively from the experimental data, remains asymptotically unbiased. On the other hand, HDB typically achieves lower variance by leveraging the historical data, which usually has a much larger sample size than the experimental data. Finally, our proposed estimator, which we introduce in the following section, effectively strikes a balance between bias and variance and outperforms both baseline estimators; see Table 2 for an illustration.

### 3.2 A pessimistic estimator for data integration

We begin with a summary of our proposal. Our approach is to linearly combine the two baseline estimators presented in Section 3.1 for data integration, while taking into account the posterior shift between the experimental and historical data. The key here is to determine the optimal weight (see e.g., Equation (3) below for the definition) for data combination. Our main idea is to transform this weight selection problem into offline policy learning. Specifically, we conceptualize the choice of weight as an 'action', which could vary as a function of the contextual information. This conceptualization effectively frames the weight selection as a policy learning problem where the goal is to identify an optimal policy that maximizes reward or minimizes cost, the latter of which corresponds to the MSE of the weighted ATE estimator. Figure 1 gives an overview of the proposed estimator pipeline. Adopting this perspective enables us to employ state-of-the-art pessimistic policy learning algorithms such as counterfactual risk minimization to effectively determine the weight.

We next detail our methodology. Similar to EDO and HDB, our estimator employs $\mathbb{E}_n[\psi_1^{(e)}(O^{(e)})]$ to estimate the mean outcome under the treatment policy. As for the control, it uses a weight function $w(s)$ to linearly combine the estimating functions used in EDO and HDB. Specifically, we define the following estimating function,

$$\psi_w(O^{(e)}, O^{(h)}) = w(S^{(e)})\psi_0^{(e)}(O^{(e)}) + [1 - w(S^{(h)})]\psi^{(h)}(O^{(h)}). \tag{3}$$

It is immediate to see that setting $w = 1$ recovers EDO's estimating function $\psi_0^{(e)}$ whereas setting $w = 0$ recovers the HDB's estimating function $\psi^{(h)}$. This leads to the following weighted ATE estimator,

$$\widehat{\text{ATE}}(w) = \mathbb{E}_n[\psi_1^{(e)}(O^{(e)})] - \mathbb{E}_n[\psi_w(O^{(e)}, O^{(h)})], \tag{4}$$

where the second empirical average $\mathbb{E}_n$ is taken over all pairs of $(O^{(e)}, O^{(h)}) \in \mathcal{D}^{(e)} \times \mathcal{D}^{(h)}$.

It remains to identify the optimal weight function $w^*$ that optimally balances the bias and variance of the ATE estimator to minimize its MSE. As mentioned earlier, we adopt an offline policy learning framework and view each value of $w$ – bounded between 0 and 1 – as an arm in a contextual bandit model. Given that $w$ is a function of the covariates, it defines a policy on this contextual space. The identification of $w^*$ is thus equivalent to the identification of the optimal policy that minimizes the cost, which in our case equals the MSE.

Although the oracle MSE is unknown, it can be estimated from the offline data. Specifically, since EDO is derived solely from $\mathcal{D}_e$, it is expected to be asymptotically unbiased. Thus, its deviation from $\widehat{\text{ATE}}(w)$ (i.e., $\widehat{\text{bias}}(w) = \widehat{\text{ATE}}(w) - \text{EDO}$) can be used to measure $\widehat{\text{ATE}}(w)$'s bias (denoted by bias($w$)). Additionally, its variance (denoted by Var($w$)) can be estimated using the sampling variance formula (see Appendix B.3 for details). Denote the resulting variance estimator by $\widehat{\text{Var}}(w)$. Given a parametric function class $\mathcal{W}$, the optimal $w^*$ can be estimated by minimizing

$$\widehat{\text{MSE}}(w) = \widehat{\text{bias}}^2(w) + \widehat{\text{Var}}(w), \tag{5}$$

over $w \in \mathcal{W}$. Following the pessimistic principle, we instead minimize an upper bound of the estimated MSE given by

$$\widehat{\text{MSE}}_U(w) = \widehat{\text{bias}}_U^2(w) + \widehat{\text{Var}}_U(w), \tag{6}$$

where $\widehat{\text{bias}}_U$ and $\widehat{\text{Var}}_U$ are required to upper bound the oracle bias and variance so that the following assumption is satisfied.

**Assumption 1** (Coverage probability). *Assume* $\mathbb{P}(\cap_{w \in \mathcal{W}}\{\widehat{\text{bias}}_U(w) \geq |\text{bias}(w)|\}) \geq 1 - \alpha$ *and* $\mathbb{P}(\cap_{w \in \mathcal{W}}\{\widehat{\text{Var}}_U(w) \geq \text{Var}(w)\}) \geq 1 - \alpha$ *for some* $0 < \alpha < 1$.

In practice, $\widehat{\text{bias}}_U$ and $\widehat{\text{Var}}_U$ can be constructed using concentration inequalities [116] or Wald-type confidence intervals [117]. We detail our implementation in Appendix B.4. We also remark that for clarity of presentation, we focus on IS estimators for ATE estimation in this section. However, our actual implementation employs DR estimators, which are known to be more efficient than IS with well-specified reward models [118]. The detailed formulas are relegated to Appendix B.2

Let $\widehat{w}$ denote the minimizer of (6), which yields our proposed estimator $\widehat{\text{ATE}}(\widehat{w})$. To conclude, we remark that our framework unifies several baseline estimators through specific choices of $\widehat{w}$:

1. **EDO**: Setting $\widehat{w}$ to 1 recovers the experimental-data-only estimator;

2. **HDB**: Setting $\widehat{w}$ to 0 yields the historical-data-based estimator;

3. **MVE**: Omitting the bias term in (5) and minimizing (5) leads to the minimal-variance estimator in Li et al. [20];

4. **NonPessi**: Minimizing (5) as opposed to (6) produces the non-pessimistic estimator;

5. **CWE** (short for constant weight estimator): Restricting $\mathcal{W}$ to constant functions of the context and setting $\widehat{\text{Var}}_U$ to $\widehat{\text{Var}}$ result in the pessimistic estimator in Li et al. [2].

We will analytically compare these estimators in the following section.

## 4 Statistical properties and analytical comparisons

We first analyze the MSE of our proposed estimator. We next analytically compare it against other baseline estimators. Our analysis covers five different scenarios:

(i) **Heavy-tailed historical rewards**, where the reward residual $R^{(h)} - r(A^{(h)}, S^{(h)})$ exhibits substantial variability;

(ii) **Heavy-tailed experimental rewards**, where the reward residual from the control group exhibits substantial variability;

(iii) **Small posterior shifts**, where the bias due to posterior shift $\mathbb{E}[\psi^{(h)}(O^{(h)}) - \psi_0^{(e)}(O^{(e)})]$ is much smaller than the standard deviation of its estimator $\mathbb{E}_n[\psi^{(h)}(O^{(h)}) - \psi_0^{(e)}(O^{(e)})]$;

(iv) **Moderate posterior shifts**, with the bias being much larger than the estimator's standard deviation, yet falling within its high-confidence bound, making it undetectable from the data;

(v) **Large posterior shifts**, where the bias is larger than the upper confidence bound of $\mathbb{E}_n[\psi^{(h)}(O^{(h)}) - \psi_0^{(e)}(O^{(e)})]$, allowing it to be detected from the data.

See the formal definitions of these scenarios in Corollaries 1 – 4. While each of the aforementioned baseline estimators might be optimal in certain scenarios, they can perform poorly in others. In contrast, the proposed estimator is adaptive and robust: it performs comparably to the scenario-specific optimal estimators in most cases. See Table 1 for a summary.

We begin by introducing a boundedness assumption and presenting an upper bound for the MSE of the proposed estimator. To simplify the theoretical analysis, we follow [2] and study a sample-split version of the ATE estimator where half of the data triplets in $\mathcal{D}^{(e)}$ and $\mathcal{D}^{(h)}$ are used to estimate $\widehat{w}$ by solving (6), while the remaining half are used to construct the ATE estimator in (4).

**Assumption 2** (ATE boundedness). *There exists some constant $B > 0$ such that both the absolute values of ATE and our estimator $\widehat{\text{ATE}}(\widehat{w})$ are upper bounded by $B$.*

**Theorem 1** (MSE of the proposed estimator). *Under Assumptions 1 and 2, we have for any $w \in \mathcal{W}$, $\text{MSE}(\widehat{\text{ATE}}(\widehat{w})) - \text{MSE}(\widehat{\text{ATE}}(w))$ can be bounded by:*

$$\mathbb{E}[\widehat{\text{bias}}_U^2(w) - \text{bias}^2(w)] + \mathbb{E}[\widehat{\text{Var}}_U(w) - \text{Var}(w)] + O(\alpha B^2). \tag{7}$$

Theorem 1 is generic in that it applies to any OPE estimator – direct, IS or DR – used to learn the ATE, provided that Assumptions 1 and 2 are satisfied. We also remark that Assumption 2 is mild. In practice, the size of the ATE is typically very small in A/B testing [102, 119, 108, 107]. Equation (7) upper bounds the difference in MSE between the proposed ATE estimator and any weighted estimator with a fixed weight function $w$. Under the realizability assumption [see e.g., 120] where $w^* \in \mathcal{W}$, setting $w = w^*$ in Theorem 1 leads to an upper bound on the difference between the MSE of our estimator and that of the optimal weighted estimator. According to (7), this upper bound can be decomposed into three parts: the first two terms quantify the estimation errors of the squared bias and the two variances respectively, and the last term, being proportional to $\alpha$, represents the probability of under-coverage – the probability that $\widehat{\text{bias}}_U$ or $\widehat{\text{Var}}_U$ fails to upper bound the oracle bias or variance.

Notice that through the use of concentration inequalities, the last term can be made arbitrarily small without largely inflating the estimation errors of the bias and variance. As for the first two terms, a key observation is that the bias and variance upper bounds in these terms depend on the weight function only through a fixed $w$, rather than the estimated weight $\widehat{w}$. This arises from the pessimistic principle, which, in policy learning, ensures that the regret of the estimated policy depends only on the reward estimation error under the optimal action, rather than under the *estimated* optimal action [121, 122]. In our setting, this principle is crucial for enabling the proposed pessimistic estimator to achieve adaptivity. To elaborate, we impose the following conditions.

**Assumption 3** (Coverage). *The probability mass functions of both $(A^{(e)}, S^{(e)})$ and $S^{(h)}$ are bounded from below by some constant $\epsilon > 0$.*

**Assumption 4** (Additive noise). *Assume $R^{(h)} = r^{(h)}(0, S^{(h)}) + \epsilon^{(h)}$ for some mean-zero random error $\epsilon^{(h)}$ independent of $S^{(h)}$. Similarly, assume $R^{(e)} = r^{(e)}(A^{(e)}, S^{(e)}) + \epsilon_{A^{(e)}}^{(e)}$ for mean-zero random errors $\epsilon_0^{(e)}$ and $\epsilon_1^{(e)}$ independent of $S^{(e)}$ and $A^{(e)}$.*

**Assumption 5** (Reward function boundedness). *The reward functions $r^{(h)}$ and $r^{(e)}$ are uniformly bounded in absolute value by some constant $r_{\max} > 0$.*

The coverage and boundedness assumptions are commonly imposed in RL and OPE [see e.g., 123, 124]. Note that the boundedness condition applies only to the *reward function*, not to the *reward* itself. The reward – being the sum of the reward function and the residual – can be unbounded due to the potential heavy-tailedness of the residual. The additive noise assumption is widely imposed in machine learning and statistics [see e.g., 125, 126]. Under this assumption, we use $\sigma^{(h)}$ and $\sigma^{(e)}$ to denote the standard deviations of $\epsilon^{(h)}$ and $\epsilon_0^{(e)}$, respectively. These standard deviations are used to characterize the tails of these error distributions. Specifically, in the first two scenarios with heavy-tailed reward residuals, $\sigma^{(h)}$ and $\sigma^{(e)}$ can be substantially large. These two cases naturally favor EDO and HDB as optimal estimators, respectively, since they avoid incorporating heavy-tailed rewards for ATE estimation. In the last three scenarios, we measure the posterior shift by the reward difference $b(s) = r^{(h)}(0, s) - r^{(e)}(0, s)$. When $|b(s)|$ is small so that variance dominates the squared bias, MVE is asymptotically optimal since it is designed for variance minimization. With moderate-to-large values of $|b(s)|$, EDO becomes again the optimal estimator as it avoids bias by excluding the historical dataset from the ATE estimation. The following corollaries demonstrate that our proposed estimator performs comparably to these optimal estimators across all scenarios, maintaining robustness with either heavy-tailed reward residuals or posterior shift.

**Corollary 1** (Scenario (i)). *Assume Assumptions 1 − 5 hold. Let $\delta = |\mathcal{D}^{(h)}|/|\mathcal{D}^{(e)}|$ denote the ratio between the sample sizes of the two datasets. Then with heavy-tailed historical rewards where $\sigma^{(h)} \gg [\epsilon^{-1}\sqrt{\delta}(\sigma^{(e)} + r_{\max})]$, $\omega^*(s) \to 1$ for any $s$ so that EDO becomes the asymptotically optimal estimator. By setting $w$ in Theorem 1 to 1, the difference in MSE between the proposed estimator and EDO is*

$$\mathbb{E}[\widehat{\text{Var}}_U(\text{EDO}) - \text{Var}(\text{EDO})] + O(\alpha B^2), \tag{8}$$

*which is much smaller than MSE(EDO) itself under mild conditions specified in Appendix C.4.*

**Corollary 2** (Scenario (ii))**.** *Assume Assumptions 1 – 5 hold. Then with heavy-tailed experimental rewards where $\sigma^{(e)} \gg [\epsilon^{-1/2}(\sigma^{(h)}\delta^{-1/2} + \sqrt{|\mathcal{D}^{(e)}|}r_{\max})]$, $\omega^*(s) \to 0$ for any $s$ so that HDB becomes the asymptotically optimal estimator. Additionally, the difference in MSE between the proposed estimator and HDB is much smaller than MSE(HDB) itself under mild conditions specified in Appendix C.5.*

**Corollary 3** (Scenario (iii))**.** *Assume Assumptions 1 – 5 hold. Then with small posterior shifts such that $|b(s)| \ll \min(\sigma^{(e)}/\sqrt{|\mathcal{D}^{(e)}|}, \sigma^{(h)}/\sqrt{|\mathcal{D}^{(h)}|})$, MVE achieves the smallest MSE. Additionally, the difference in MSE between the proposed estimator and MVE is much smaller than MSE(MVE) itself under certain conditions specified in Appendix C.6.*

**Corollary 4** (Scenarios (iv) and (v))**.** *Assume Assumptions 1 – 5 hold and that either $b(s) > 0$ for all $s$ or $b(s) < 0$ for all $s$. Then with either moderate posterior shifts such that*

$$\frac{\sigma^{(e)} + r_{\max}}{\sqrt{\epsilon|\mathcal{D}^{(e)}|}} + \frac{\sigma^{(h)} + r_{\max}}{\sqrt{\epsilon|\mathcal{D}^{(h)}|}} \ll |b(s)| = O\Big(\frac{\sigma^{(e)} + r_{\max}}{\sqrt{\epsilon|\mathcal{D}^{(e)}|}}\sqrt{\log|\mathcal{D}^{(e)}|} + \frac{\sigma^{(h)} + r_{\max}}{\sqrt{\epsilon|\mathcal{D}^{(h)}|}}\sqrt{\log|\mathcal{D}^{(h)}|}\Big)$$

*for any $s$, or large posterior shifts such that*

$$|b(s)| \gg \Big(\frac{\sigma^{(e)} + r_{\max}}{\sqrt{\epsilon|\mathcal{D}^{(e)}|}}\sqrt{\log|\mathcal{D}^{(e)}|} + \frac{\sigma^{(h)} + r_{\max}}{\sqrt{\epsilon|\mathcal{D}^{(h)}|}}\sqrt{\log|\mathcal{D}^{(h)}|}\Big),$$

*for any $s$, $\omega^*(s) \to 1$ for any $s$ so that EDO becomes the asymptotically optimal estimator. Additionally, the difference in MSE between the proposed estimator and EDO is upper bounded by* (8)*, which is much smaller than MSE(EDO) itself under mild conditions specified in Appendix C.7.*

Corollaries 1 – 4 upper bound the excess MSE of the proposed estimator over the scenario-specific optimal estimators across Scenarios (i)-(v). Importantly, the excess MSEs in (i), (iv) and (v) are independent of $\sigma^{(h)}$ or $b(s)$, which demonstrates our estimator's robustness when these parameters become (moderately) large in their respective scenarios. Furthermore, these corollaries establish the *oracle* property of our estimator: it asymptotically achieves the same MSE as the optimal estimator for each scenario, working efficiently as if it knew the underlying scenario.

We next compare against the baseline estimators mentioned in Section 3.2 analytically.

- **EDO**: According to Corollaries 1 and 4, EDO is asymptotically optimal in Scenarios (i), (iv) and (v). However, it underperforms our estimator in Scenarios (ii) and (iii), where incorporating historical data yields more accurate ATE estimation.

- **HDB**: As demonstrated in Corollary 2, HDB is asymptotically optimal in Scenario (ii). However, unlike the proposed estimator, it generally fails in Scenarios (i), (iii), (iv) and (v).

- **MVE**: Corollary 3 shows that MVE is asymptotically optimal in Scenario (iii). However, it suffers from a large bias in Scenarios (iv) and (v), due to the posterior shifts.

- **NonPessi**: Similar to Corollaries 3 and 4, we can show that NonPessi is asymptotically optimal in scenarios (iii) and (v) when the posterior shift is either small or large. However, it is not optimal for moderate shifts in Scenario (iv). This is because without adopting the pessimistic principle, its excess MSE depends on the estimation errors $\widehat{\mathrm{bias}}(\widehat{w})$ and $\widehat{\mathrm{Var}}(\widehat{w})$ at the estimated weight $\widehat{w}$. Although the optimal population-level weight $w^* \to 1$ with moderate posterior shifts, the estimated $\widehat{w}$ may not, since the bias is not large enough to be detectable [2]. Similarly, in the first two scenarios with heavy-tailed rewards, NonPessi – unlike the pessimistic estimator – can suffer from a large MSE when $\sigma^{(e)}$ and $\sigma^{(h)}$ are large, yet not sufficiently so to be detected from the data.

- **CWE**: While [2] showed that CWE is optimal in Scenarios (iv) and (v), it differs from our method in two ways: (a) it restricts $\mathcal{W}$ to constant weight functions, and (b) it applies the pessimistic principle only partially – to upper bound the squared bias term but not the variance. (a) leads to its sub-optimality in Scenario (iii), where the optimal weight for MVE is typically context-dependent rather than being constant. Similar to NonPessi, (b) makes CWE sub-optimal in the first two scenarios.

We have so far focused exclusively on the estimation of the ATE. To conclude this section, we remark that our proposal also accommodates valid inference for A/B testing. By employing sample-splitting

and doubly robust ATE estimation, valid p-values can be readily obtained when combined with. Specifically, we use one half of the data to estimate the weight function $\widehat{w}$ and nuisance functions (the reward and density ratio), and the other half to construct the ATE estimator. Following [127], one can show that the resulting ATE estimator is asymptotically normal under suitable regularity conditions. As a result, standard $z$-tests based on normal approximation can be used to obtain valid p-values. Our numerical studies in Section 5 confirm that the resulting p-values remain valid across all experimental settings.

# 5 Numerical experiments

In this section, we evaluate the finite-sample performance of the proposed estimator, comparing it against **EDO**, **MVE**, **CWE**, and **NonPessi** (introduced in Section 3.2). We exclude **HDB** as it performs similarly or worse than MVE in our experiments. Instead, we include **LASSO**, proposed by Cheng and Cai [25], which selects weights by minimizing the estimated variance of the ATE estimator with a Lasso penalty. MSEs are computed over 100 simulation replications. Details of the data generating process are provided in Appendix A.

**Example 5.1** (**Synthetic-data simulation**). We design settings to cover all five scenarios mentioned in Section 4 and Table 1. Specifically, we model the difference between reward functions $b(s)$ as $\mu_{\text{diff}} \times d(s)$ for some nonzero function $d$ and a scalar parameter $\mu_{\text{diff}} \in [0, 5]$ controlling the degree of posterior shift. When $\mu_{\text{diff}} = 0$, the reward functions are identical, indicating no posterior shift. Increasing $\mu_{\text{diff}}$ leads to larger shifts. This covers Scenarios (iii) – (v), which range from small to moderate to large posterior shifts. We also consider two forms for the function $d$: (i) a piecewise function of the context variable, and (ii) a linear function of the context variable, resulting in piecewise and linear shifts, respectively. Finally, we allow the reward residuals to follow either a normal distribution (light-tailed) or a Student's $t$-distribution with 6 degrees of freedom (heavy-tailed). This covers the first two scenarios.

The top panels of Figure 2 report the MSEs of all ATE estimators under piecewise shifts, while the middle panels exclude MVE, which exhibits large MSEs even under moderate to large posterior shifts, to allow for a clearer comparison of the remaining estimators. It can be seen that when $\mu_{\text{diff}}$ is small, MVE achieves the lowest MSEs, and the proposed method performs comparably. As $\mu_{\text{diff}}$ increases, the proposed estimator outperforms all baseline alternatives in most cases. Notably, even when $\mu_{\text{diff}}$ is large – where EDO is expected to perform best – our proposal still achieves lower MSEs under normally distributed experimental reward residuals. This benefits from its use of a context-adaptive weight function. When the reward difference $b(s)$ is negative for some contexts and positive for others, a properly chosen context-adaptive weight can still incorporate historical data to reduce variance while effectively cancelling out bias. In contrast, CWE and LASSO, which rely on constant weights, tend to converge toward EDO's performance.

Bottom panels of Figure 2 show similar trends under linear shifts (excluding MVE). We also consider a nonlinear form of $d(s)$ and conduct additional experiments in Appendix A, which confirm similar patterns under nonlinear posterior shifts. Finally, we remark that the LASSO estimator is implemented using a carefully chosen tuning parameter to ensure competitive performance. Additional results in Appendix A (Figure 8) reveal LASSO's sensitivity to this hyperparameter.

**Example 5.2** (**Ridesharing-data-based simulation**). In this example, we construct a simulation environment based on a real-world A/A dataset collected from a ridesharing platform. The contextual information includes two variables: the total online time of drivers and the number of order requests across one day. The reward is defined as the total daily income earned by each driver. We first learn the outcome model using these variables from this A/A dataset, and then generate synthetic experimental and historical data based on this model, following scenarios similar to those in Example 5.1. Results reported in Figures 3 and additional results in Appendix A (Figures 9-12) align with the findings in Example 5.1, where the proposed estimator achieves the lowest MSEs in most cases.

Additionally, we conduct a clinical data–based simulation in Example A.1 (Appendix). The results exhibit patterns consistent with those in Examples 5.1–5.2, where the proposed estimator outperforms competing methods in most cases. Furthermore, we assess the inference procedure by testing the nullity of the ATE and comparing it with those based on EDO and CWE. As shown in Table 3, all three methods adequately control the Type I error under the null (ATE = 0), while the proposed test demonstrates higher power under the alternative.

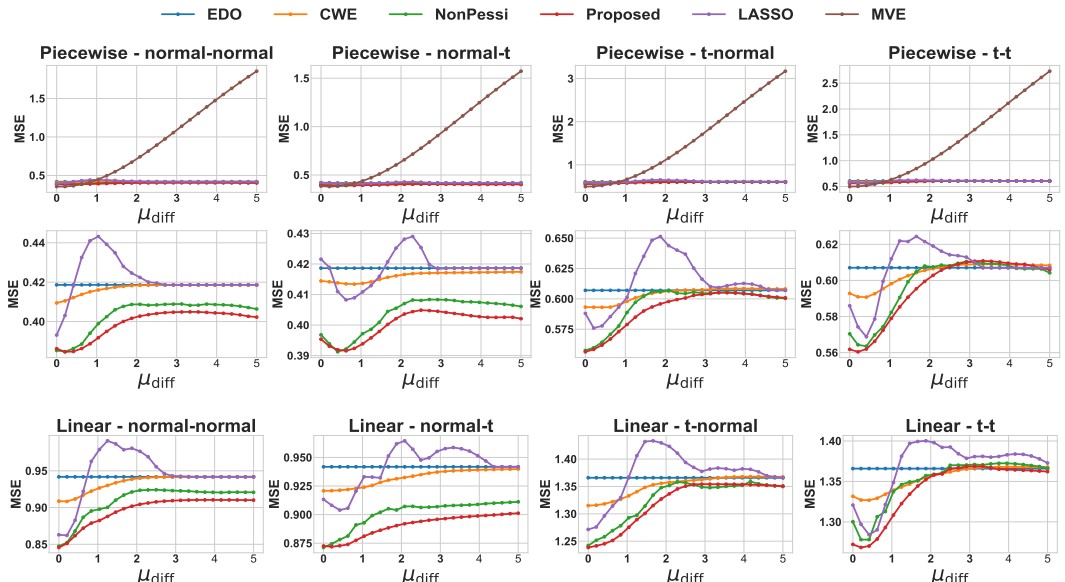

Figure 2: MSEs in Example 5.1. Top panels show all estimators under piecewise shifts; middle panels zoom in without MVE; bottom panels present results for linear shifts excluding MVE.

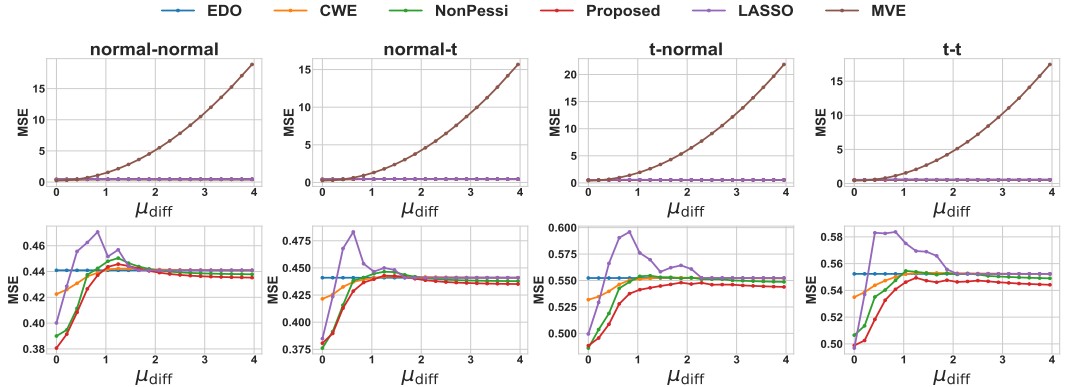

Figure 3: MSEs of ATE estimators in Example 5.2. Top: all estimators; Bottom: exclude MVE.

## Discussion

In this work, we study how to integrate historical control data with experimental data to enhance A/B testing. We proposed a covariate-dependent weighting scheme that treats the weight as a policy and learns it by minimizing a pessimistic upper bound on the estimator's MSE. We establish MSE bounds for the resulting estimator. We evaluate it against competitive baselines across five representative settings. Both our theoretical analysis and empirical results demonstrate greater robustness to heavy-tailed rewards and near-optimal handling of diverse posterior shifts.

Several directions merit future exploration. The proposed method focuses on a non-dynamic setting, whereas in many practical applications, treatments are sequential and may influence future outcomes. A natural extension is to accommodate dynamic settings with carryover effects by explicitly modeling the underlying dynamics. Our theoretical analysis also relies on bounded-reward assumptions, which could be relaxed to handle unbounded outcomes. Beyond IS and DR estimators for the ATE, the proposed framework can be extended to more general off-policy evaluation and reinforcement learning objectives, including least-squares temporal-difference methods and fitted $Q$-evaluation.

## Acknowledgments

Xiangkun Wu's research is supported by the National Key Research and Development Program of China (Grant No. 2024YFC2511003). Ting Li's research is partially supported by the National Natural Science Foundation of China (No. 12571304), the Shanghai Pujiang Program (No. 24PIC030), CCF-DiDi GAIA Collaborative Research Funds and the Program for Innovative Research Team of Shanghai University of Finance and Economics.

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

# Appendix

## Appendix Contents

# A    Experimental Settings and Additional Results

In this section, we present details of the data generating process for the toy example in Table 2 and Section 5, and additional experimental results. All experiments were conducted on a high-performance computing node equipped with dual AMD EPYC 7742 64-Core Processors (128 logical cores). Each experiment with 100 replications can be typically completed within 90 minutes.

**Toy Example in Table 2.** We define the reward functions for the experimental and historical data as follows

$$R_e = \begin{cases} 10 + 3S_e + 6\epsilon_e, & \text{if } A_e = 0 \\ 10 + A_e + 3S_e + 2\epsilon_e, & \text{if } A_e = 1 \end{cases}, \quad R_h = 11.4 + S_h + 6\epsilon_h,$$

where the state variables $S_h, S_e$ and the errors $\epsilon_h, \epsilon_e$ are independently drawn from the standard normal distribution $N(0,1)$. In the experimental data, actions are evenly split between the two treatment groups, with half assigned to 1 and the other half to 0.

**Example 5.1 (Continued).** In this example, we consider five posterior shift scenarios as follows.

(1) **Piecewise Shifts:** The reward functions for the historical and experimental data are:

$$R_e = 10 + A_e + S_e + 2\epsilon_e, \quad R_h = 10 + d_\mu(S_h)\mu_{\text{diff}} + S_h + (2 + d_S(S_h))\epsilon_h$$

where

$$d_S(S) = \begin{cases} -1, & S < -1 \\ -1, & -1 \le S < 0 \\ 2, & S > 0 \end{cases}, \quad d_\mu(S) = \begin{cases} 0, & S < -1 \\ 1, & -1 \le S < 0 \\ 1, & S > 0 \end{cases},$$

and $S_e$ and $S_h$ are sampled from $N(0,1)$. The noise terms $\epsilon_e$ and $\epsilon_h$ follow four distribution combinations: **normal-normal**, **normal-t**, **t-normal**, and **t-t**, where **normal** is $N(0,1)$ and **t** is the $t$-distribution with 6 degrees of freedom.

The parameter $\mu_{\text{diff}}$ captures distributional differences between datasets, discretized into 25 values from 0 to 5. In the experimental dataset, the action $A_e$ alternates deterministically between 0 and $1 - 0$ for the first, third, and fifth samples, and 1 for the second, fourth, and sixth samples. The shift magnitude depends on the state $S$ through $d_\mu(S)$, and the noise variance varies with the state via $d_S(S)$.

(2) **Linear Shifts:** The reward functions for the two datasets are

$$R_e = 10 + A_e + S_e + 3\epsilon_e, \quad R_h = 10 + \mu_{\text{diff}} + \mu_{\text{diff}}S_h + 3\epsilon_h,$$

where $S_e, S_h, \epsilon_e, \epsilon_h$, and $A_e$ follow the same configurations to Setting (1).

(3) **Cosine Shifts:** The reward functions are given by

$$R_e = 10 + A_e + \cos(S_e) + 3\epsilon_e, \quad R_h = 10 + \mu_{\text{diff}} + \mu_{\text{diff}}\cos(S_h) + S_h + 3\epsilon_h.$$

All other configurations remain the same to Setting (1).

(4) **Quadratic Shifts:** The reward functions are given by

$$R_e = 10 + A_e + S_e^2 + 3\epsilon_e, \quad R_h = 10 + \mu_{\text{diff}} + \mu_{\text{diff}}S_e^2 + S_h + 3\epsilon_h.$$

All other configurations remain the same.

(5) **Absolute Shifts:** The reward functions are given by

$$R_e = 10 + A_e + |S_e| + 3 \cdot \epsilon_e, \quad R_h = 10 + \mu_{\text{diff}} + \mu_{\text{diff}}|S_h| + |S_h| + 3\epsilon_h.$$

All other configurations remain the same.

Figure 4 shows the empirical MSEs of all estimators, including MVE, under linear shifts. When $\mu_{\text{diff}}$ is small, MVE performs well due to minimal bias in the historical data. However, as $\mu_{\text{diff}}$ grows, this bias increases sharply, causing MVE's MSE to rise substantially. Figures 5–7 present the empirical MSEs under cosine, quadratic, and absolute shift settings. The results remain consistent across varying levels of posterior shift. The MVE method shows marginal gains only when $\mu_{\text{diff}}$ is very small; however, its performance quickly deteriorates as $\mu_{\text{diff}}$ increases, because it ignores the bias from the historical data, resulting in substantially large MSE.

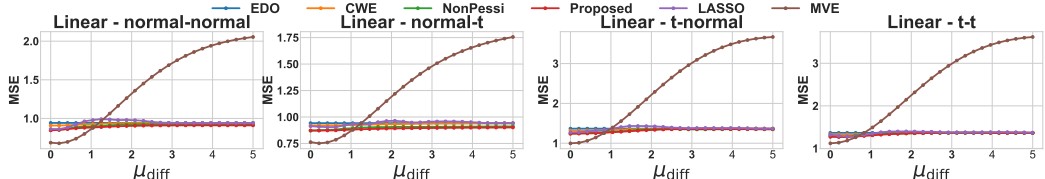

Figure 4: Empirical means of MSE for various estimators, including MVE, under the piecewise shifts and linear shifts.

When the experimental data follow a normal distribution, our method consistently outperforms all baselines over the entire range of $\mu_{\text{diff}} \in [0, 5]$, regardless of the data generating process of the historical dataset. It achieves a much smaller MSE compared to EDO and Pessi when $\mu_{\text{diff}}$ is small. Although its MSE increases as $\mu_{\text{diff}}$ grows, our method still maintains a clear advantage over other baselines. This gain arises from the use of a non-constant, learned weight function that adapts to distributional shifts, in contrast to EDO's fixed weight.

When the experimental data follow a $t$-distribution, our method continues to perform well, particularly when $\mu_{\text{diff}} < 2$. As $\mu_{\text{diff}}$ increases, the performance of all methods converges to that of EDO.

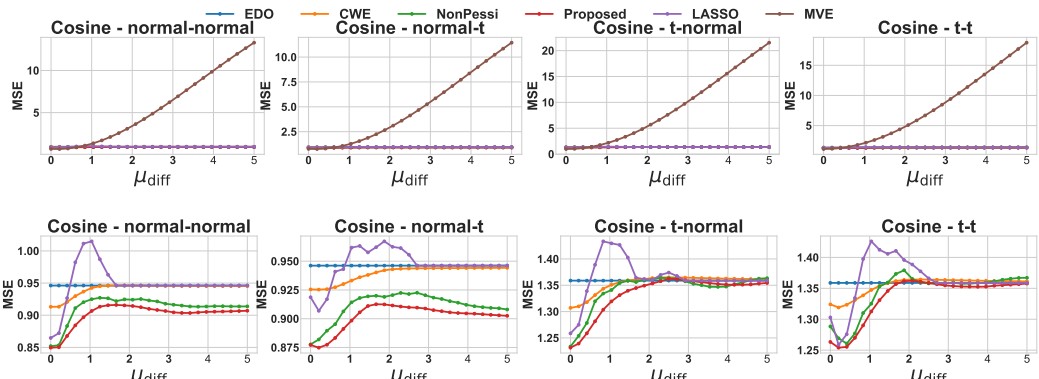

Figure 5: Empirical means of MSEs in Example 5.1 under cosine shifts. The top panel shows all estimators; the bottom panel zooms in by excluding the MVE method.

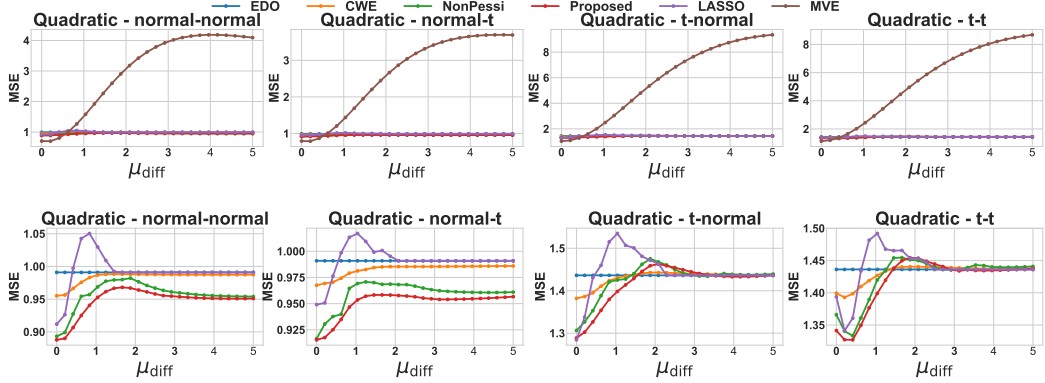

Figure 6: Empirical means of MSEs in Example 5.1 under quadratic shifts. The top panel shows all estimators; the bottom panel zooms in by excluding the MVE method.

**Sensitivity Analysis of the Lasso Tuning Parameter.** We investigate the performance of the Lasso estimator across a range of tuning parameters $\lambda \in \{0.1, 0.2, \dots, 1.0\}$, using the same data generating process as in the piecewise shifts setting. Figure 8 reports the performance of Lasso and EDO across varying $\lambda$ values. For small values of $\mu_{\text{diff}}$, Lasso with a small $\lambda$ outperforms EDO. However, its

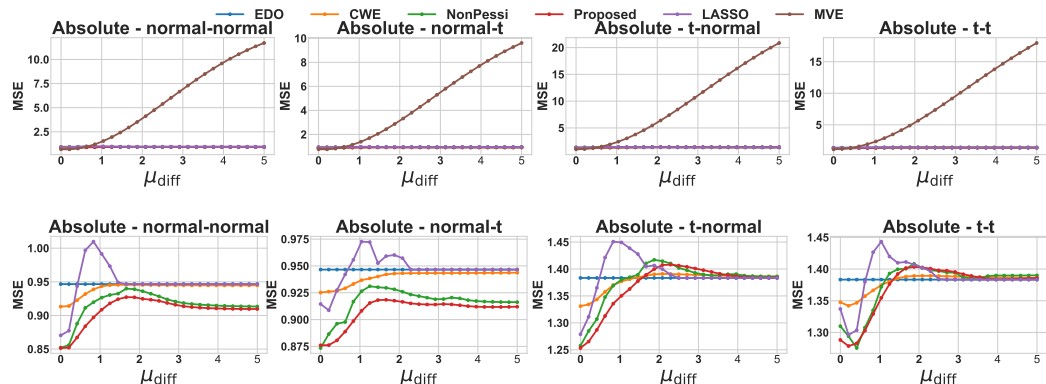

Figure 7: Empirical means of MSEs in Example 5.1 under absolute shifts. The top panel shows all estimators; the bottom panel zooms in by excluding the MVE method.

performance deteriorates as $\mu_{\text{diff}}$ increases. In contrast, Lasso with a large $\lambda$ performs comparably to EDO.

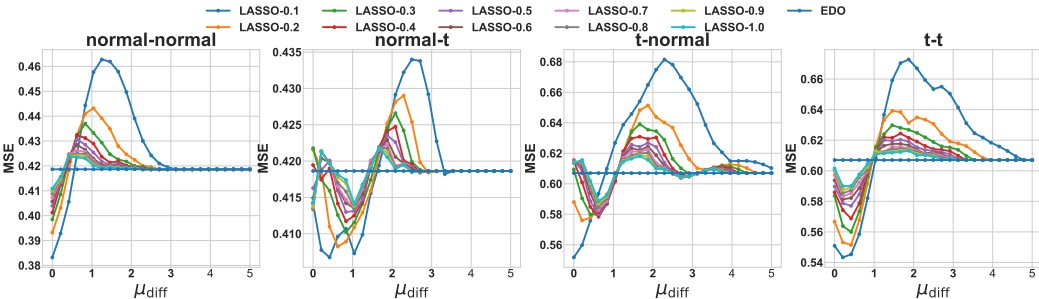

Figure 8: Hyperparameter Sensitivity Analysis

**Example 5.2 (Continued).** This example uses a real-world A/A experiment dataset from a leading ridesharing platform. The reward $R$ represents daily driver income, and the contextual variables $S_1$ and $S_2$ denote the number of ride requests and total online time during the first hour of the day, respectively.

For privacy concerns, company and city identifiers are omitted, and all variables are scaled. The contextual features $S_1$ and $S_2$ are normalized to have unit standard deviation. We fit the following linear model:

$$R = \beta_0 + \beta_1 S_1 + \beta_2 S_2 + \epsilon,$$

and obtain the estimates $\widehat{\beta}_0, \widehat{\beta}_1$ and $\widehat{\beta}_2$. Based on the estimated coefficients, we generate the experimental and historical datasets as follows:

$$R_e = \widehat{\beta}_0 + A_e + \widehat{\beta}_1 S_{e1} + \widehat{\beta}_2 S_{e2} + \delta\epsilon_e,$$
$$R_h = \widehat{\beta}_0 + \widehat{\beta}_1 S_{h1} + \widehat{\beta}_2 S_{h2} + \mu_{\text{diff}} + \mu_{\text{diff}} \cdot (S_{h1} + S_{h2})/20 + \delta\epsilon_h,$$

where $S_{e1}, S_{h1}$ are sampled from $N(\mu_1, 1)$ and $S_{e2}, S_{h2}$ from $N(\mu_2, 1)$, with $\mu_1$ and $\mu_2$ being the empirical means of $S_1$ and $S_2$ from the real dataset. To ensure privacy, we do not report $\mu_1$ and $\mu_2$ individually, but their sum lies between 10 and 20. The action $A_e$ is binary, assigned deterministically: even-indexed samples receive $A_e = 1$, odd-indexed samples $A_e = 0$. The noise terms $\epsilon_e$ and $\epsilon_h$ follow four combinations: normal-normal, normal-$t$, $t$-normal, and $t$-$t$, where the $t$-distribution has 6 degrees of freedom. The experimental dataset contains $|\mathcal{D}_e| = 48$ samples, and the historical dataset has $|\mathcal{D}_h| = m \cdot |\mathcal{D}_e|$, with $m \in \{1, 2, 3\}$. A noise scaling constant $\delta \in \{1, 2, 3\}$ controls the noise magnitude.

Figure 9 reports the empirical MSEs across methods, showing consistent patterns with the $m = 1$ case in the main text. Across all settings, our method consistently achieves strong performance.

When the experimental data is heavy-tailed, it significantly outperforms non-pessimistic baselines with lower MSEs across all $\mu_{\text{diff}}$ values. With small posterior shifts, it clearly outperforms both EDO and Pessi; in the moderate shift regime, it outperforms Lasso; and under large posterior shifts, it remains stable and performs slightly better than EDO.

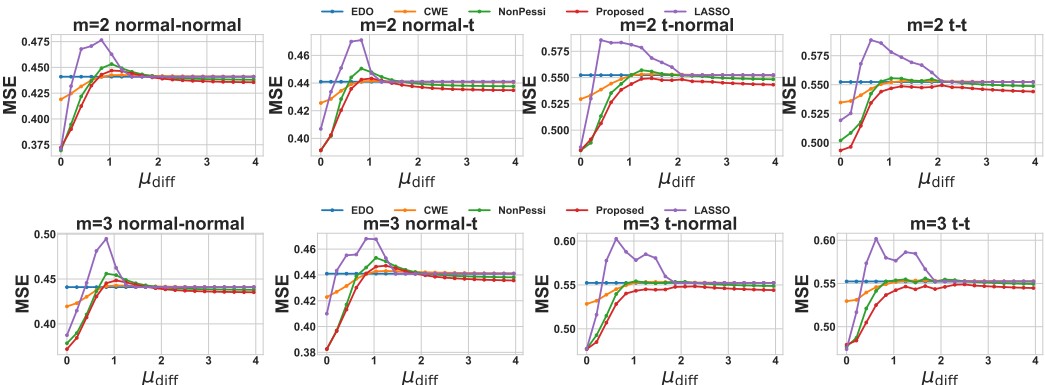

Figure 9: Empirical means of MSEs of various methods with $\delta = 1$ in Example 5.2 for $m = 2$ (top) and $m = 3$ (bottom).

Figure 10 shows the results under varying noise magnitudes. As expected, the MSE of all methods increases with higher residual variance (characterized by $\delta$), reflecting the impact of noise on estimation accuracy. Nevertheless, our method consistently outperforms all baselines across the full range of $\mu_{\text{diff}}$ values.

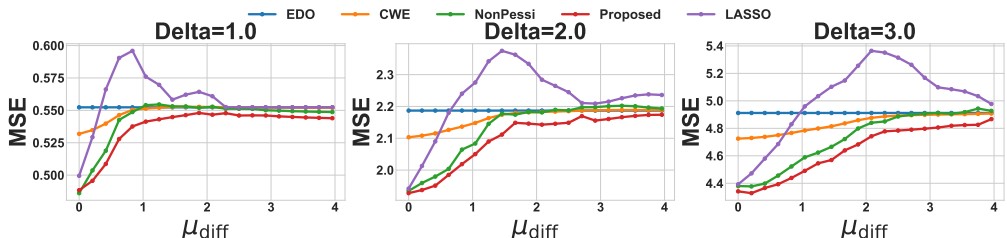

Figure 10: Empirical means of MSEs of various methods with $m = 1$ in Example 5.2 across different $\delta$s.

Figure 11 examines the impact of treatment assignment under varying probabilities $\mathbb{P}(A_e = 1) = \text{prob}$, with $\text{prob} \in \{0.3, 0.5, 0.7\}$, in the normal-normal setting with $m = \delta = 1$. The results show consistent performance across all methods, with our approach remaining robust and effective under different assignment probabilities.

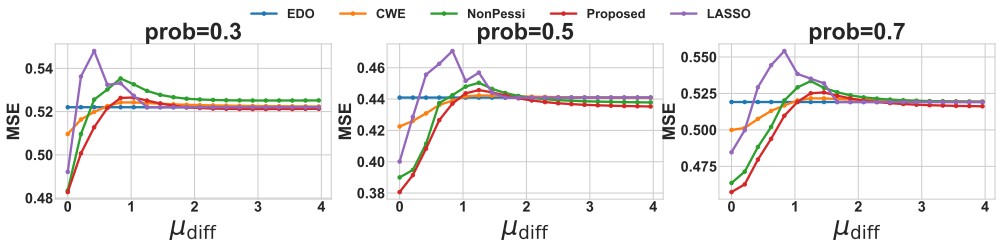

Figure 11: Empirical means of MSEs of various methods with $m = 1, \delta = 1$ in Example 5.2 in different prob scales.

Figure 12 examines the performance of various methods as sample size increases, focusing on the $t$-normal noise setting. As expected, MSEs decrease with larger sample sizes due to reduced variance. Notably, our method consistently performs well across all subplots and levels of $\mu_{\text{diff}}$.

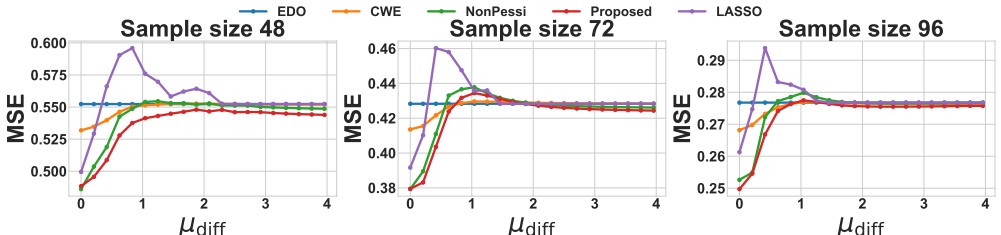

Figure 12: Empirical means of MSEs of various methods with $m = 1, \delta = 1$ in Example 5.2 across $|\mathcal{D}_e| = 48, 72, 96$.

**Example A.1 (Clinical-data-based simulation).** In this section, we construct a simulation environment based on the public real-world dataset ACTG175, which consists of 2,139 HIV-positive individuals randomized to four treatments. We focus on comparing ZDV+ddI ($n = 522$) and ZDV+zal ($n = 524$), treating them as actions 1 and 0, respectively. The outcome of interest is the rescaled CD4 count, and we consider three covariates: age ($S_1$), homosexual activity ($S_2$), and hemophilia ($S_3$). We construct a simulator similar to Example 5.2, and generate both experimental and historical data for evaluation.

Specifically, the outcome model is specified as:

$$R = f(S) = \beta_0 + \beta_1 S_1^2 + \beta_2 S_1 + \beta_3 S_2 + \beta_4 S_3 + \gamma A,$$

We use the data to fit the model. Using fitted parameters, we generate synthetic outcomes based on real data:

$$R_e = f(S_e) + 0.8\delta\epsilon_e, \quad R_h = f(S_h) + \mu_d + 0.05\mu_d S_{h,1} + 0.8\delta\epsilon_h,$$

where $\epsilon_e$ is drawn from $\mathcal{N}(0,1)$ and $\epsilon_h$ from either $\mathcal{N}(0,1)$ or the heavy-tailed $t_6$ distribution; alternatively, $\epsilon_e$ is drawn from $t_9$ and $\epsilon_h$ from either $\mathcal{N}(0,1)$ or $t_9$. This setting yields four possible scenarios, corresponding to all combinations of $\epsilon_e$ and $\epsilon_h$ being sampled from a standard normal or from a heavy-tailed distribution. Here, $S_1'$ represents covariates generated from a normal distribution fitted using the empirical mean and variance of all observed $S_1$. Here, $S_2'$ and $S_3'$ are sampled from Bernoulli distributions with parameters set to the empirical means of $S_2$ and $S_3$, respectively. We estimate the variance parameter $\delta$ as the average of the squared residuals from the fitted model. In experimental data, treatment assignments are randomized ($A \sim \text{Bernoulli}(0.5)$), whereas in historical data they are fixed at $A = 0$. Both datasets contain 48 samples. We vary $\mu_d \in [0,5]$ (25 points) and compare the empirical average MSE over 100 simulations. Results in Figure 13 reveal that the proposed estimator achieves the lowest MSEs in most cases.

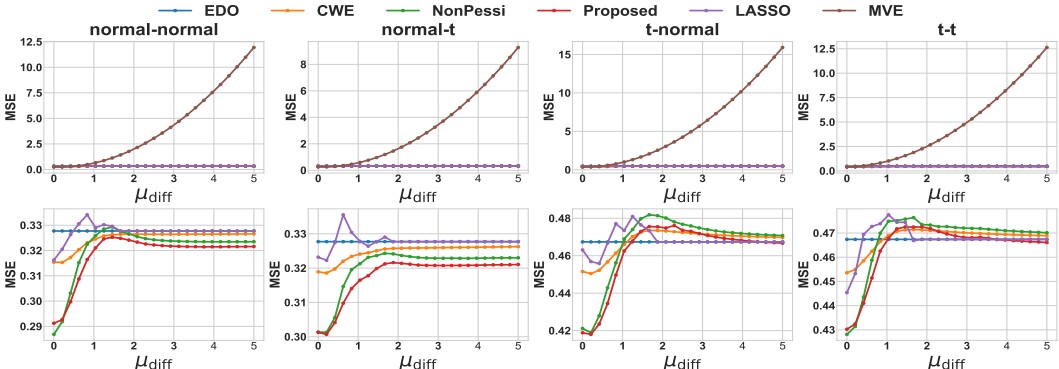

Figure 13: MSEs of ATE estimators in Clinical example. Top: all estimators; Bottom: exclude MVE.

We also consider the following hypothesis testing problem:

$$H_0 : \text{ATE} = 0 \quad \text{vs.} \quad H_1 : \text{ATE} \neq 0.$$

To conduct valid inference, we estimate the ATE using doubly robust procedures and compute p-values using a Wald test. To compare the proposed method against EDO and Pessi, we first set the true ATE to 0 and conduct 1,000 Monte Carlo simulations to estimate the Type I error rate of each method at the 5% significance level. To assess empirical power, we gradually increase the signal strength by setting the ATE to $\{0.5, 1.0, 1.5, 2.0, 2.5, 3.0\}$. We further examine three levels of posterior shifts: small, moderate, and large. Table 3 reports the empirical rejection rates. Under the null where ATE equals zero, all three methods control Type I error. Under the alternative, the proposed test is more powerful than the two competitors.

Table 3: Empirical Type-I error rates and power under different bias levels. The best results in each row are highlighted in bold.

| ATE | Metric | Small Posterior Shifts | | | Moderate Posterior Shifts | | | Large Posterior Shifts | | |
|---|---|---|---|---|---|---|---|---|---|---|
| | | EDO | CWE | Proposed | EDO | CWE | Proposed | EDO | CWE | Proposed |
| 0.0 | Type-I Error | 0.030 | 0.032 | **0.024** | 0.030 | 0.035 | **0.029** | 0.030 | 0.030 | **0.027** |
| 0.5 | Power | 0.081 | **0.097** | 0.093 | 0.081 | 0.103 | **0.114** | 0.081 | 0.084 | **0.088** |
| 1.0 | Power | 0.267 | 0.310 | **0.320** | 0.267 | 0.311 | **0.335** | 0.267 | 0.276 | **0.286** |
| 1.5 | Power | 0.546 | 0.592 | **0.611** | 0.546 | 0.586 | **0.613** | 0.546 | 0.558 | **0.572** |
| 2.0 | Power | 0.810 | 0.847 | **0.867** | 0.810 | 0.835 | **0.863** | 0.810 | 0.815 | **0.825** |
| 2.5 | Power | 0.955 | 0.967 | **0.974** | 0.955 | 0.965 | **0.970** | 0.955 | 0.960 | **0.962** |
| 3.0 | Power | 0.994 | 0.996 | **0.997** | 0.994 | 0.996 | **0.997** | 0.994 | 0.994 | **0.994** |

## B  Implementation Details

In this section, we provide detailed procedures for implementing our proposal based on importance sampling estimator (Section B.1) and doubly robust estimator (Section B.2).

### B.1  Importance Sampling Method

Recall that $\psi_a^{(e)}(O^{(e)}) = \mathbb{I}(A = a)R^{(e)}/\pi(a|S^{(e)})$, and $\psi^{(h)}(O^{(h)}) = \mu(S^{(h)})R^{(h)}$. The proposed weighted ATE estimator is given by (4), and can be obtained as follows.

- Step 1: estimate $\psi_a^{(e)}(O^{(e)})$ and $\psi^{(h)}(O^{(h)})$ using the methodology detailed in B.5 and B.6
- Step 2: Estimate $w(S)$ using Section B.7 by minimizing the upper bound of the estimated MSE derived in B.4.
- Step 3: Obtain the weighted ATE estimator by plugging in the unknown terms with their estimates in (4).

To simplify notation, we define

$$Z^{(e)}(w) = \psi_1^{(e)}(O^{(e)}) - w(S^{(e)})\psi_0^{(e)}(O^{(e)}), \qquad Z^{(h)}(w) = (1 - w(S^{(h)}))\psi^{(h)}(O^{(h)}). \qquad (9)$$

The weighted ATE estimator for a given weight function $w$ can thus be written as

$$\widehat{\text{ATE}}(w) = \mathbb{E}_n(Z^{(e)}(w)) - \mathbb{E}_n(Z^{(h)}(w)).$$

### B.2  Doubly Robust Method

We present two baseline DR estimators before introducing the proposed weighted DR estimator. The first baseline estimator is constructed using only experimental data,

$$\tau_{dr}^{(e)} = \frac{1}{|\mathcal{D}_e|} \sum_{O_e \in \mathcal{D}_e} [\psi_{dr,1}^{(e)}(O^{(e)}) - \psi_{dr,0}^{(e)}(O^{(e)})],$$

where the estimating function $\psi_{dr,a}^{(e)}$ for $a \in \{0, 1\}$ is given by

$$\psi_{dr,a}^{(e)}(O^{(e)}) = \frac{\mathbb{I}(A^{(e)} = a)}{\pi(a \mid S^{(e)})} \left[ R^{(e)} - r^{(e)}(A^{(e)}, S^{(e)}) \right] + r^{(e)}(a, S^{(e)}).$$

It can be shown that $\psi_{\mathrm{dr},a}^{(e)}(O^{(e)})$ is unbiased to the mean outcome under treatment $a$ as long as either the propensity score model $\pi$ or the outcome model $r^{(e)}$ is correctly specified. This yields the doubly robustness property.

The second estimator incorporates historical data into the ATE estimation and is defined as

$$\tau_{\mathrm{dr}}^{(h)} = \frac{1}{|\mathcal{D}^{(e)}|} \sum_{O^{(e)} \in \mathcal{D}^{(e)}} \psi_{\mathrm{dr},1}^{(e)}(O^{(e)}) - \frac{1}{|\mathcal{D}^{(e)}|} \sum_{S^{(e)} \in \mathcal{D}^{(e)}} r^{(h)}(0, S^{(e)}) - \frac{1}{|\mathcal{D}^{(h)}|} \sum_{O^{(h)} \in \mathcal{D}^{(h)}} \psi_{\mathrm{dr}}^{(h)}(O^{(h)}),$$

where the estimating function $\psi_{\mathrm{dr}}^{(h)}(O^{(h)})$ is given by

$$\psi_{\mathrm{dr}}^{(h)}(O^{(h)}) = \mu(S^{(h)}) \left[ R^{(h)} - r^{(h)}(0, S^{(h)}) \right].$$

Next, given a weight function $w$, we define the weighted DR estimator as:

$$
\begin{aligned}
\widehat{\mathrm{ATE}}_{\mathrm{dr}}(w) \;=\; & \frac{1}{|\mathcal{D}^{(e)}|} \sum_{O^{(e)} \in \mathcal{D}^{(e)}} \psi_{\mathrm{dr},1}^{(e)}(O^{(e)}) - \frac{1}{|\mathcal{D}^{(e)}|} \sum_{O^{(e)} \in \mathcal{D}^{(e)}} w(S^{(e)}) \psi_{\mathrm{dr},0}^{(e)}(O^{(e)}) \\
& - \frac{1}{|\mathcal{D}^{(e)}|} \sum_{S^{(e)} \in \mathcal{D}^{(e)}} \left( 1 - w(S^{(e)}) \right) r^{(h)}(0, S^{(e)}) \\
& - \frac{1}{|\mathcal{D}^{(h)}|} \sum_{O^{(h)} \in \mathcal{D}^{(h)}} \left( 1 - w(S^{(h)}) \right) \psi_{\mathrm{dr}}^{(h)}(O^{(h)}).
\end{aligned}
\tag{10}
$$

Similar to IS, we define $Z_{\mathrm{dr}}^{(e)}(w)$ and $Z_{\mathrm{dr}}^{(h)}(w)$ as

$$Z_{dr}^{(e)}(w) \;=\; \psi_{\mathrm{dr},1}^{(e)}(O^{(e)}) - w(S^{(e)}) \psi_{\mathrm{dr},0}^{(e)}(O^{(e)}) - (1 - w(S^{(e)})) r^{(h)}(0, S^{(e)}) \tag{11}$$

$$Z_{dr}^{(h)}(w) \;=\; (1 - w(S^{(h)})) \psi_{dr}^{(h)}(O^{(h)}). \tag{12}$$

Using these notations, the final ATE estimator is given by

$$\widehat{\mathrm{ATE}}_{dr}(w) = \mathbb{E}_n(Z_{dr}^{(e)}(w)) - \mathbb{E}_n(Z_{dr}^{(h)}(w)).$$

To summarize, our DR ATE estimator can be constructed through the following steps.

- Step 1: estimate $\psi_{\mathrm{dr},a}^{(e)}(O^{(e)})$ for $a \in \{0,1\}$, $r^{(h)}(0, S^{(e)})$ and $\psi_{\mathrm{dr}}^{(h)}(O^{(h)})$ using the method in B.5 and B.6.
- Step 2: Estimate $w(S)$ using Section B.7 by minimizing the upper bounded of the estimated MSE B.4.
- Step 3: Obtain the weighted ATE estimator by plugging in the unknown terms with their estimates in (10).

### B.3 Explicit Form of $\mathrm{Var}(w)$ and its Estimator $\widehat{\mathrm{Var}}(w)$

In this part, we present the detailed expression for the variance term $\mathrm{Var}(w)$ in (5), and its estimator $\widehat{\mathrm{Var}}(w)$. Since the experimental and historical dataset are mutually independent, and samples within each dataset are independently and identically distributed, $\mathrm{Var}(w)$ can be written as:

$$\mathrm{Var}(w) = \mathrm{Var}(\widehat{\mathrm{ATE}}(w)) = \frac{\mathrm{Var}(Z^{(e)}(w))}{|\mathcal{D}^{(e)}|} + \frac{\mathrm{Var}(Z^{(h)}(w))}{|\mathcal{D}^{(h)}|}, \tag{13}$$

Therefore, to estimate $\mathrm{Var}(w)$, it suffices to estimate $\mathrm{Var}(Z^{(e)}(w))$ and $\mathrm{Var}(Z^{(h)}(w))$.

Their estimators $\widehat{\mathrm{Var}}(Z^{(e)}(w))$ and $\widehat{\mathrm{Var}}(Z^{(h)}(w))$ can be obtained using the standard sample variance formula,

$$\widehat{\mathrm{Var}}(Z^{(e)}(w)) := \frac{1}{|\mathcal{D}^{(e)}| - 1} \sum_{i=1}^{|\mathcal{D}^{(e)}|} \left( Z_i^{(e)}(w) - \mathbb{E}_n[Z^{(e)}(w)] \right)^2,$$

$$\widehat{\mathrm{Var}}(Z^{(h)}(w)) := \frac{1}{|\mathcal{D}^{(h)}| - 1} \sum_{j=1}^{|\mathcal{D}^{(h)}|} \left( Z_j^{(h)}(w) - \mathbb{E}_n[Z^{(h)}(w)] \right)^2,$$

where $\mathbb{E}_n[Z^{(e)}(w)]$ and $\mathbb{E}_n[Z^{(h)}(w)]$ are defined as:

$$\mathbb{E}_n[Z^{(e)}(w)] = \frac{1}{|\mathcal{D}^{(e)}|} \sum_{i=1}^{|\mathcal{D}^{(e)}|} Z_i^{(e)}(w), \quad \text{and} \quad \mathbb{E}_n[Z^{(h)}(w)] = \frac{1}{|\mathcal{D}^{(h)}|} \sum_{j=1}^{|\mathcal{D}^{(h)}|} Z_j^{(h)}(w).$$

This yields the variance estimator

$$\widehat{\text{Var}}(w) := \frac{\widehat{\text{Var}}(Z^{(e)}(w))}{|\mathcal{D}^{(e)}|} + \frac{\widehat{\text{Var}}(Z^{(h)}(w))}{|\mathcal{D}^{(h)}|}. \tag{14}$$

## B.4 Explicit Form of $\widehat{\text{MSE}}_U(w)$

In this section, we derive $\widehat{\text{MSE}}_U(w)$ in (6). It consists of the two terms: $\widehat{\text{Var}}_U(w)$ and $\widehat{\text{bias}}_U(w)$. We seek these two terms that satisfy the coverage probability in Assumption 1. In what follows, we take the bias term as an example and present three approaches for its construction: one based on empirical process theory [128], another based on Markov inequality and Bonferroni's inequality, and a third based on the multiplier bootstrap [129].

For the first one, define the following function classes:

$$\mathcal{F}^{(e)} = \left\{ f_w^{(e)}(o) := (1 - w(s))\psi^{(e)}(o) : w \in \mathcal{W} \right\},$$

$$\mathcal{F}^{(h)} = \left\{ f_w^{(h)}(o) := (1 - w(s))\psi^{(h)}(o) : w \in \mathcal{W} \right\}.$$

Then, $\widehat{\text{bias}}(w) - \text{bias}(w)$ is the difference between two empirical processes indexed by $w$. If the function class $\mathcal{W}$ satisfies certain complexity properties (e.g., being a VC class), one can apply empirical process theory to construct a uniform upper bound $U$ such that

$$\mathbb{P}\left( \sup_{w \in \mathcal{W}} |\widehat{\text{bias}}(w) - \text{bias}(w)| \leq U \right) \geq 1 - \alpha, \tag{15}$$

or

$$\mathbb{P}\left( \sup_{w \in \mathcal{W}} \frac{|\widehat{\text{bias}}(w) - \text{bias}(w)|}{\widehat{\sigma}(w)} \leq U \right) \geq 1 - \alpha, \tag{16}$$

where $\widehat{\sigma}(w)$ is a consistent estimator of the asymptotic variance of $\widehat{\text{bias}}(w)$. Accordingly, we can set:

- $\widehat{\text{bias}}_U(w) = U + |\widehat{\text{bias}}(w)|$ in the unnormalized case;
- $\widehat{\text{bias}}_U(w) = \widehat{\sigma}(w)U + |\widehat{\text{bias}}(w)|$ in the normalized case.

More specifically, when the error terms $\epsilon^{(e)}$, $\epsilon^{(h)}$ are sub-Gaussian, we may set:

$$U = \frac{c r_{\max}}{\epsilon} \sqrt{\frac{v \log n_{\max} + \log(1/\alpha)}{n_{\min}}},$$

where $c > 0$ is a constant, $v$ denotes the VC dimension of the function class $\mathcal{W}$, $n_{\max} = \max(|\mathcal{D}^{(e)}|, |\mathcal{D}^{(h)}|)$ and $n_{\min} = \min(|\mathcal{D}^{(e)}|, |\mathcal{D}^{(h)}|)$. This can also be extended to heavy-tailed errors [see e.g., 129, Section 5].

Alternatively, under the following finite-hypothesis-class assumption, $\widehat{\text{bias}}_U$ and $\widehat{\text{Var}}_U$ can be constructed based on asymptotic normality and Bonferroni's inequality.

**Assumption 6** (Finite hypothesis class). *The number of elements in $\mathcal{W}$ is finite.*

Assumption 6 is commonly employed in machine learning to simplify the theoretical analysis [see e.g., 123].

**Assumption 7.** *The error terms $\epsilon_0^{(e)}$, $\epsilon_1^{(e)}$ and $\epsilon_h$ are assumed to have finite eighth moments.*

Assumption 7 is standard in high-order moment analysis and is commonly adopted in the literature on finite-sample concentration .The class of distributions with a finite eighth moment is very broad. We define $\widehat{\mathrm{Var}}_U\big(Z^{(e)}(w_i)\big) = \widehat{\mathrm{Var}}\big(Z^{(e)}(w_i)\big) + U^{(e)}(w_i)$ as the upper bound for $\mathrm{Var}\big(Z^{(h)}(w_i)\big)$ and $\widehat{\mathrm{Var}}_U\big(Z^{(h)}(w_i)\big) = \widehat{\mathrm{Var}}\big(Z^{(h)}(w_i)\big) + U^{(h)}(w_i)$ as the upper bound for $\mathrm{Var}\big(Z^{(e)}(w_i)\big)$. We consider a set $\mathcal{W}$ with $K$ elements. Since the samples are independent and identically distributed (i.i.d.) and the experimental dataset $\mathcal{D}^{(e)}$ is independent of the historical dataset $\mathcal{D}^{(h)}$, we have:

$$
\begin{aligned}
\widehat{\mathrm{Var}}_U(w_i) &= \frac{\widehat{\mathrm{Var}}_U\big(Z^{(e)}(w_i)\big)}{|\mathcal{D}^{(e)}|} + \frac{\widehat{\mathrm{Var}}_U\big(Z^{(h)}(w_i)\big)}{|\mathcal{D}^{(h)}|} \\
&= \frac{\widehat{\mathrm{Var}}\big(Z^{(e)}(w_i)\big) + U^{(e)}(w_i)}{|\mathcal{D}^{(e)}|} + \frac{\widehat{\mathrm{Var}}\big(Z^{(h)}(w_i)\big) + U^{(h)}(w_i)}{|\mathcal{D}^{(h)}|} \\
&= \widehat{\mathrm{Var}}(w_i) + \frac{U^{(e)}(w_i)}{|\mathcal{D}^{(e)}|} + \frac{U^{(h)}(w_i)}{|\mathcal{D}^{(h)}|}.
\end{aligned}
$$

Assume there exist positive constants $\alpha_1, \alpha_2, \alpha_3, \alpha_4 > 0$, and define $\alpha_{\mathrm{var}} := \alpha_1 + \alpha_2$ and $\alpha_{\mathrm{bias}} := \alpha_3 + \alpha_4$. For each $i \in \{1, \ldots, K\}$, we construct a confidence interval via Markov's inequality applied to the fourth moment:

$$
\mathbb{P}\left(\mathrm{Var}\big(Z^{(e)}(w_i)\big) \geq \widehat{\mathrm{Var}}_U\big(Z^{(e)}(w_i)\big)\right) \leq \frac{\mathbb{E}\left[\left(\mathrm{Var}\big(Z^{(e)}(w_i)\big) - \widehat{\mathrm{Var}}\big(Z^{(e)}(w_i)\big)\right)^4\right]}{U^{(e)}(w_i)^4} = \frac{\alpha_1}{K}. \quad (17)
$$

$$
\mathbb{P}\left(\mathrm{Var}\big(Z^{(h)}(w_i)\big) \geq \widehat{\mathrm{Var}}_U\big(Z^{(h)}(w_i)\big)\right) \leq \frac{\mathbb{E}\left[\left(\mathrm{Var}\big(Z^{(h)}(w_i)\big) - \widehat{\mathrm{Var}}\big(Z^{(h)}(w_i)\big)\right)^4\right]}{U^{(h)}(w_i)^4} = \frac{\alpha_2}{K}. \quad (18)
$$

Direct calculations lead to:

$$
\begin{aligned}
&\mathbb{P}\left(\mathrm{Var}(w_i) \geq \widehat{\mathrm{Var}}_U(w_i)\right) \\
\leq\ &\mathbb{P}\left(\mathrm{Var}\big(Z^{(e)}(w_i)\big) \geq \widehat{\mathrm{Var}}\big(Z^{(e)}(w_i)\big) + U^{(e)}(w_i)\right) \\
&\bigcup \mathrm{Var}\big(Z^{(h)}(w_i)\big) \geq \widehat{\mathrm{Var}}\big(Z^{(h)}(w_i)\big) + U^{(h)}(w_i) \\
\leq\ &\mathbb{P}\left(\mathrm{Var}\big(Z^{(e)}(w_i)\big) \geq \widehat{\mathrm{Var}}\big(Z^{(e)}(w_i)\big) + U^{(e)}(w_i)\right) \\
+\ &\mathbb{P}\left(\mathrm{Var}\big(Z^{(h)}(w_i)\big) \geq \widehat{\mathrm{Var}}\big(Z^{(h)}(w_i)\big) + U^{(h)}(w_i)\right) \\
\leq\ &\frac{\alpha_1}{K} + \frac{\alpha_2}{K} = \frac{\alpha_{var}}{K},
\end{aligned} \quad (19)
$$

where the first inequality follows from the relationship between the sets, and the second inequality follows from the probability of the union bound.

We can write $\widehat{\mathrm{bias}}_U(w_i)$ as:

$$
\widehat{\mathrm{bias}}_U(w_i) = |\widehat{\mathrm{bias}}(w_i)| + \left(U_b^{(e)}(w_i) + U_b^{(h)}(w_i)\right) \quad (20)
$$

This bound can also be derived via the fourth-moment version of Markov's inequality,

$$\mathbb{P}\left(|\text{bias}(w_i) - \widehat{\text{bias}}(w_i)| \leq U_b^{(e)}(w_i) + U_b^{(h)}(w_i)\right)$$

$$\leq \frac{\mathbb{E}\left[\left((1 - w(S^{(e)})\psi_0^{(e)}(O^{(e)}) - \mathbb{E}(1 - w(S^{(e)})\psi_0^{(e)}(O^{(e)}))\right)^4\right]}{U_b^{(e}(w_i)^4}$$

$$+ \frac{\mathbb{E}\left[\left((1 - w(S^{(h)})\psi_0^{(h)}(O^{(h)}) - \mathbb{E}(1 - w(S^{(h)})\psi_0^{(h)}(O^{(h)}))\right)^4\right]}{U_b^{(h}(w_i)^4}$$

$$= \frac{\alpha_3}{K} + \frac{\alpha_4}{K} = \frac{\alpha_{bias}}{K}. \tag{21}$$

According to Bonferroni's inequality, we have

$$P\left(\bigcap_{i=1}^{K}\left\{\widehat{\text{bias}}_U(w_i) \geq |\text{bias}(w_i)|\right\}\right) \geq 1 - \sum_{w=1}^{K} P\left(\widehat{\text{bias}}_U(w_i) < |\text{bias}(w_i)|\right) \geq 1 - \alpha_{bias}. \tag{22}$$

$$P\left(\bigcap_{i=1}^{K}\left\{\widehat{\text{Var}}_U(w_i) \geq \text{Var}(w_i)\right\}\right) \geq 1 - \sum_{w=1}^{K} P\left(\widehat{\text{Var}}_U(w_i) < \text{Var}(w_i)\right) \geq 1 - \alpha_{Var}. \tag{23}$$

Then, combining (22) and (23) and setting $\alpha = \alpha_{\text{var}} + \alpha_{\text{bias}}$, we obtain the claim. This is the procedure we adopt to construct $\widehat{\text{bias}}_U$ and $\widehat{\text{Var}}_U$ in Corollaries 1–4

Finally, one may use the high-dimensional multiplier bootstrap to construct $\widehat{\text{bias}}_U$ [129, 130]. Under mild regularity conditions, the distribution of the supremum in (15) or (16) converges to that of a Gaussian process [129, Theorem 2.1]. Furthermore, the supremum of this Gaussian process can be approximated via the multiplier bootstrap, which enables us to set to set $U$ to the $\alpha$-quantile of the supremum of the following bootstrapped process

$$\sup_{w \in \mathcal{W}}\left[\frac{1}{|\mathcal{D}^{(e)}|}\sum_{i=1}^{|\mathcal{D}^{(e)}|}(1 - w(S^{(e)}))\psi^{(e)}(O_i^{(e)})g_i - \frac{1}{|\mathcal{D}^{(h)}|}\sum_{j=1}^{|\mathcal{D}^{(h)}|}(1 - w(S^{(h)}))\psi^{(h)}(O_j^{(h)})g_{|\mathcal{D}^{(e)}|+j}\right],$$

in the unnormalized case, and

$$\sup_{w \in \mathcal{W}}\frac{1}{\widehat{\sigma}(w)}\left[\frac{1}{|\mathcal{D}^{(e)}|}\sum_{i=1}^{|\mathcal{D}^{(e)}|}(1 - w(S^{(e)}))\psi^{(e)}(O_i^{(e)})g_i - \frac{1}{|\mathcal{D}^{(h)}|}\sum_{j=1}^{|\mathcal{D}^{(h)}|}(1 - w(S^{(h)}))\psi^{(h)}(O_j^{(h)})g_{|\mathcal{D}^{(e)}|+j}\right],$$

in the normalized case, where $g_i$s are i.i.d. standard Gaussian variables. Repeating the process over multiple bootstrap samples provides an empirical estimate of the quantile.

Similar to the first approach, the uniform bound can then be defined as:

$$\widehat{\text{bias}}_U(w) = U + |\widehat{\text{bias}}(w)| \quad \text{or} \quad \widehat{\text{bias}}_U(w) = U + \widehat{\sigma}(w)|\widehat{\text{bias}}(w)|.$$

As for the double robust estimator, to obtain $\widehat{\text{Var}}_U(w)$, we only need to replace $Z^{(e)}(w)$ and $Z^{(h)}(w)$ with $Z_{\text{dr}}^{(e)}(w)$ and $Z_{\text{dr}}^{(h)}(w)$. The bias upper bound can be similarly established.

In our implementation, we find that the uniform upper bound over $w \in \mathcal{W}$ tends to be overly conservative. Therefore, instead of enforcing a global bound, we adopt pointwise, non-uniform upper bounds for $\widehat{\text{Var}}_U(w)$ and $\widehat{\text{bias}}_U(w)$, computed individually at each $w$ based on normal approximation.

## B.5   Estimation of Nuisance Function

Accurate estimation of the propensity score $\pi(a \mid S^{(e)})$ is essential for both DR and IS. In our implementation, we estimate this nuisance function via logistic regression. The outcome functions $r^{(h)}(0, S^{(h)})$, $r^{(e)}(0, S^{(e)})$, and $r^{(e)}(1, S^{(e)})$ can be flexibly estimated using a variety of regression models, including basis function expansions, random forests, and neural networks.

## B.6 Estimation of $\mu(S)$

To estimate the density ratio $\mu(S)$, we adopt a moment-matching approach. Specifically, we seek a function $\mu(S)$ such that the following moment conditions are satisfied:

$$\mathbb{E}[\mu(S^{(e)})\,\Phi_k(S^{(e)})] = \mathbb{E}[\Phi_k(S^{(h)})], \quad \text{for } k = 1, \dots, K,$$

where $\Phi_k(S)$ denotes the $k$-th test function.

Let $\Phi(S) = [\Phi_1(S), \dots, \Phi_K(S)]^\top$ be the corresponding feature map. We approximate the density ratio $\mu(S)$ by a linear model of the form $\mu(S) \approx \Phi(S)^\top \gamma$, and estimate the coefficient vector $\gamma \in \mathbb{R}^K$ by solving a sample moment-matching equation between the historical and experimental datasets.

$$\frac{1}{|\mathcal{D}_h|} \sum_{S_i^{(h)} \in \mathcal{D}_h} \Phi(S_i^{(h)})\Phi(S_i^{(h)})^\top \gamma = \frac{1}{|\mathcal{D}_e|} \sum_{S_j^{(e)} \in \mathcal{D}_e} \Phi(S_j^{(e)}).$$

In practice, the feature function $\Phi(\bullet)$ can be set to polynomials, splines, or neural network features. Under mild or negligible covariate shift, the density ratio can be simplified to 1.

## B.7 Estimation of the weight function $w(S)$

In our implementation, we parameterized $w(S)$ using a logistic model,

$$w(S) = \frac{1}{1 + e^{-\theta^\top S}},$$

which ensures that $w(S) \in (0, 1)$ for all $S$. Alternatively, a neural network can be employed. Given a parameterized $w$, the pessimistic objective function derived in Section B.4 can be optimized using gradient-based methods to learn the parameters.

# C Proofs of the Theorems and Corollaries

In this section, we present the proofs of Theorem 1 and Corollaries 1 – 4. Recall that Theorem 1 applies to any ATE estimator, while Corollaries 1 – 4 are specific to the IS estimator.

## C.1 Proof of Theorem 1

*Proof of Theorem 1.* To facilitate the analysis, we define the following events:

$$A := \bigcap_{w \in \mathcal{W}} \left\{ \widehat{\text{bias}}_U(w) \geq |\text{bias}(w)| \right\}, \quad B := \bigcap_{w \in \mathcal{W}} \left\{ \widehat{\text{Var}}_U(w) \geq \text{Var}(w) \right\}, \quad C := A \cap B.$$

For a given $w$, we define $\text{MSE}(w)$ as the MSE of the ATE estimator $\widehat{\text{ATE}}(w)$. Since the estimated weight $\widehat{w}$ itself is random, $\text{MSE}(\widehat{w})$ is a random variable. This MSE is well-defined due to the use of sample splitting which ensures that the ATE estimator is independent of $\widehat{w}$. The MSE of our proposed estimator is given by the expected value of $\text{MSE}(\widehat{w})$,

$$\text{MSE}(\widehat{\text{ATE}}(\widehat{w})) = \mathbb{E}[\text{MSE}(\widehat{w})],$$

where the expectation on the right-hand-side (RHS) is taken with respect to the randomness of $\widehat{w}$.

We decompose the difference in MSE into two parts:

$$\text{MSE}(\widehat{\text{ATE}}(\widehat{w})) - \text{MSE}(\widehat{\text{ATE}}(w)) = \underbrace{\mathbb{E}\left[(\text{MSE}(\widehat{w}) - \text{MSE}(w)) \cdot 1_C\right]}_{M_1}$$

$$+ \underbrace{\mathbb{E}\left[(\text{MSE}(\widehat{w}) - \text{MSE}(w)) \cdot 1_{C^c}\right]}_{M_2}. \tag{24}$$

**Bounding $M_1$:** Under Assumption 1, on event $C$, we have:

$$\text{bias}^2(\widehat{w}) + \text{Var}(\widehat{w}) \leq \widehat{\text{bias}}_U^2(\widehat{w}) + \widehat{\text{Var}}_U(\widehat{w}).$$

Moreover, since $\widehat{w}$ minimizes the pessimistic objective defined in (6), it follows that

$$\widehat{\text{bias}}_U^2(\widehat{w}) + \widehat{\text{Var}}_U(\widehat{w}) \leq \widehat{\text{bias}}_U^2(w) + \widehat{\text{Var}}_U(w).$$

Thus, $M_1$ can be bounded as follows:

$$
\begin{aligned}
M_1 &= \quad \mathbb{E}\left[\left(\text{bias}^2(\widehat{w}) + \text{Var}(\widehat{w}) - \text{bias}^2(w) - \text{Var}(w)\right) \cdot 1_C\right] \\
&\leq \quad \mathbb{E}\left[\left(\widehat{\text{bias}}_U^2(w) - \text{bias}^2(w) + \widehat{\text{Var}}_U(w) - \text{Var}(w)\right) \cdot 1_C\right] \\
&\leq \quad \mathbb{E}\left[\widehat{\text{bias}}_U^2(w) - \text{bias}^2(w)\right] + \mathbb{E}\left[\widehat{\text{Var}}_U(w) - \text{Var}(w)\right].
\end{aligned}
\tag{25}
$$

**Bounding $M_2$:** We next bound the term associated with the complement event $C^c$. By the definition of the mean squared error (MSE), we have:

$$\text{MSE}(\widehat{\text{ATE}}(\widehat{w})) = \mathbb{E}\left[\left(\widehat{\text{ATE}}(\widehat{w}) - \text{ATE}\right)^2\right] \leq \mathbb{E}\left[\left(|\widehat{\text{ATE}}(\widehat{w})| + |\text{ATE}|\right)^2\right] = O(B^2),$$

where the first inequality follows from the triangle inequality, and the second follows from Assumption 2. Similarly, we have $\text{MSE}(w) \leq O(B^2)$ for any $w \in \mathcal{W}$.

Using this and the union bound on probabilities:

$$\mathbb{P}(C^c) = \mathbb{P}(A^c \cup B^c) \leq \mathbb{P}(A^c) + \mathbb{P}(B^c) \leq 2\alpha.$$

Hence, we bound term $M_2$ as:

$$M_2 = \mathbb{E}\left[(\text{MSE}(\widehat{w}) - \text{MSE}(w)) \cdot 1_{C^c}\right] \leq 2B^2 \cdot \mathbb{P}(C^c) \leq O(\alpha B^2). \tag{26}$$

Plugging the bounds for terms $M_1$ in (25) and $M_2$ in (26) into (24), the result follows. $\square$

## C.2  Supporting Lemmas

**Lemma 1** (MSE decomposition). *For a given weight function $w$, the MSE of the weighted estimator can be decomposed as:*

$$\text{MSE}(w) = \frac{\text{Var}(Z^{(e)}(w))}{|\mathcal{D}^{(e)}|} + \frac{\text{Var}(Z^{(h)}(w))}{|\mathcal{D}^{(h)}|} + \left(\mathbb{E}\left[(1 - w(S^{(e)}))b(S^{(e)})\right]\right)^2. \tag{27}$$

*We interpret the three terms on the right-hand-side of* (27) *as follows: (1) variance from the experimental data, (2) variance from the historical data, and (3) the squared bias introduced by incorporating historical data.*

*Proof of Lemma 1.* We decompose the MSE into the sum of variance and squared bias $\text{MSE}(w) = \text{Var}(w) + \text{bias}^2(w)$. We have already derived the closed-form expression of the variance term in (13). As for the bias, we note that:

$$
\begin{aligned}
&\mathbb{E}[(1 - w(S^{(e)}))\psi_0^{(e)}(O^{(e)})] - \mathbb{E}[(1 - w(S^{(h)}))\psi^{(h)}(O^{(h)})] \\
=&\mathbb{E}\left[(1 - w(S^{(e)})) \cdot \mathbb{E}[(r^{(e)}(0, S^{(e)}) + \epsilon_0^{(e)} \mid S^{(e)}]\right] \\
&-\mathbb{E}\left[(1 - w(S^{(h)}) \cdot \frac{p_e(S^{(e)})}{p_h(S^{(h)})} \cdot \mathbb{E}[r^{(h)}(0, S^{(h)}) + \epsilon^{(h)} \mid S^{(h)}]\right] \\
=&\mathbb{E}\left[(1 - w(S^{(e)}))(r^{(e)}(0, S^{(e)}) - r^{(h)}(0, S^{(e)}))\right] = \mathbb{E}\left[(w(S^{(e)}) - 1) \cdot b(S^{(e)})\right].
\end{aligned}
\tag{28}
$$

The second equality follows from Assumption 4 that $\epsilon_0^{(e)}$ is independent of $S^{(e)}$ with $\mathbb{E}[\epsilon_0^{(e)} \mid S^{(e)}] = 0$, and that $\epsilon^{(h)}$ is independent of $S^{(h)}$ with $\mathbb{E}[\epsilon^{(h)} \mid S^{(h)}] = 0$. Combining (13) and (28) completes the proof. $\square$

**Lemma 2** (Variance from Experiment Data). *For a given weight function $w$, the variance of $Z^{(e)}(w)$ defined in* (9) *is given by*

$$
\begin{aligned}
\text{Var}\left(Z^{(e)}(w)\right) =& \mathbb{E}\left[\frac{r^{(e)}(1, S^{(e)})^2 + (\sigma_1^{(e)})^2}{\pi(1 \mid S^{(e)})}\right] + \mathbb{E}\left[\frac{w(S^{(e)})^2(r^{(e)}(0, S^{(e)})^2 + (\sigma_0^{(e)})^2)}{\pi(0 \mid S^{(e)})}\right] \\
&- \left(\mathbb{E}[r^{(e)}(1, S^{(e)})] - \mathbb{E}[w(S^{(e)})r^{(e)}(0, S^{(e)})]\right)^2.
\end{aligned}
\tag{29}
$$

*Proof.* Direct calculation leads to

$$\mathrm{Var}\left(Z^{(e)}(w)\right) = \mathrm{Var}(\psi_1^{(e)}(O^{(e)})) + \mathrm{Var}(w(S^{(e)})\psi_0^{(e)}(O^{(e)}))$$
$$- 2\mathrm{Cov}(\psi_1^{(e)}(O^{(e)}), w(S^{(e)})\psi_0^{(e)}(O^{(e)})).$$

We proceed to compute each term on the RHS, respectively.

**1. Variance of $\psi_1^{(e)}(O^{(e)})$:** We use the law of total variance:

$$\mathrm{Var}(X) = \mathbb{E}[\mathrm{Var}(X \mid S)] + \mathrm{Var}(\mathbb{E}[X \mid S]).$$

In our case, $X = \psi_1^{(e)}(O^{(e)})$. According to Assumption 4, we have:

$$\mathbb{E}[\psi_1^{(e)}(O^{(e)}) \mid S^{(e)}] = r^{(e)}(1, S^{(e)}), \quad \mathbb{E}[(\psi_1^{(e)}(O^{(e)}))^2 \mid S^{(e)}] = \frac{r^{(e)}(1, S^{(e)})^2 + (\sigma_1^{(e)})^2}{\pi(1 \mid S^{(e)})}.$$

It follows from the total variance formula that

$$\mathrm{Var}(\psi_1^{(e)}(O^{(e)})) = \mathbb{E}\left[\frac{r^{(e)}(1, S^{(e)})^2 + (\sigma_1^{(e)})^2}{\pi(1 \mid S^{(e)})}\right] + \mathrm{Var}(r^{(e)}(1, S^{(e)})) - \mathbb{E}[r^{(e)}(1, S^{(e)})^2]. \quad (30)$$

**2. Variance of $w(S^{(e)})\psi_0^{(e)}(O^{(e)})$:** Under Assumption 4, direct calculation yields

$$\mathbb{E}[w(S^{(e)})\psi_0^{(e)}(O) \mid S^{(e)}] = w(S^{(e)})r^{(e)}(0, S^{(e)}),$$

$$\mathbb{E}[w(S^{(e)})^2\psi_0^{(e)}(O)^2 \mid S^{(e)}] = \frac{w(S^{(e)})^2(r^{(e)}(0, S^{(e)})^2 + \sigma_0^{(e)^2})}{\pi(0 \mid S^{(e)})}.$$

Here, we use $\sigma_1^{(e)}$ to denote the standard deviation of $\epsilon_1^{(e)}$, and $\sigma_0^{(e)}$ to denote the standard deviation of $\epsilon_0^{(e)}$.

We next apply the total variance formula, which leads to

$$\mathrm{Var}(w(S^{(e)})\psi_0^{(e)}(O^{(e)})) = \mathbb{E}\left(w(S^{(e)})^2 \cdot \frac{r^{(e)}(0, S^{(e)})^2 + (\sigma_0^{(e)})^2}{\pi(0 \mid S^{(e)})}\right)$$
$$+ \mathrm{Var}(w(S^{(e)})r^{(e)}(0, S^{(e)})) - \mathbb{E}\left(w(S^{(e)})^2 r^{(e)}(0, S^{(e)})^2\right) \quad (31)$$

**3. Covariance term:** Since $\psi_1^{(e)}(O^{(e)})$ is nonzero only when $A = 1$ and $\psi_0^{(e)}(O^{(e)})$ only when $A = 0$, their supports are disjoint; hence $\mathbb{E}[\psi_1^{(e)}(O^{(e)}) \psi_0^{(e)}(O^{(e)})] = 0$. Therefore, their covariance simplifies to

$$\mathrm{Cov}(\psi_1^{(e)}(O^{(e)}), w(S^{(e)})\psi_0^{(e)}(O^{(e)})) = -\mathbb{E}[\psi_1^{(e)}(O^{(e)})] \cdot \mathbb{E}[w(S^{(e)})\psi_0^{(e)}(O^{(e)})]$$
$$= -\mathbb{E}[r^{(e)}(1, S^{(e)})] \cdot \mathbb{E}[w(S^{(e)})r^{(e)}(0, S^{(e)})]. \quad (32)$$

Combining (30)–(32) completes the proof of the lemma.

$\square$

**Lemma 3** (Variance from Historical Data)**.** *For a given weight function $w$, the variance of $Z^{(h)}(w)$ defined in* (9) *is given by*

$$\mathrm{Var}\left(Z^{(h)}(w)\right) = \mathrm{Var}\left((1 - w(S^{(h)}))\mu(S^{(h)})r^{(h)}(0, S^{(h)})\right) + \mathbb{E}\left[(1 - w(S^{(h)}))^2\mu(S^{(h)})^2\right](\sigma^{(h)})^2.$$

*Proof.* By definition,

$$(1 - w(S^{(h)}))\psi^{(h)}(S^{(h)}) = (1 - w(S^{(h)}))\mu(S^{(h)})\left(r^{(h)}(0, S^{(h)}) + \epsilon^{(h)}\right).$$

Under Assumption 4, the conditional mean of $\epsilon^{(h)}$ given $S^{(h)}$ is zero. We obtain that

$$
\begin{aligned}
\mathrm{Var}\left((1-w(S^{(h)}))\cdot\psi^{(h)}(S^{(h)})\right) \;=\;\; &\mathrm{Var}\left((1-w(S^{(h)})\mu(S^{(h)})r^{(h)}(0,S^{(h)})\right) \\
+\;\; &\mathrm{Var}\left((1-w(S^{(h)})\mu(S^{(h)})\epsilon^{(h)}\right).
\end{aligned}
$$

As for the second term, by the variance formula we have

$$
\mathbb{E}\left[\left((1-w(S^{(h)}))\mu(S^{(h)})\epsilon^{(h)}\right)^2\right]-\left(\mathbb{E}\left[(1-w(S^{(h)}))\mu(S^{(h)})\epsilon^{(h)}\right]\right)^2
$$

$$
=\mathbb{E}\left[\left((1-w(S^{(h)}))\mu(S^{(h)})\epsilon^{(h)}\right)^2\right] \quad(\text{since }\mathbb{E}[\epsilon^{(h)}]=0\text{ and }\epsilon^{(h)}\text{ is dependent of }S^{(h)})
$$

$$
=\mathbb{E}\left[(1-w(S^{(h)}))^2\mu(S^{(h)})^2\right]\cdot\mathbb{E}\left[(\epsilon^{(h)})^2\right]=\mathbb{E}\left[(1-w(S^{(h)}))^2\mu(S^{(h)})^2\right](\sigma^{(h)})^2.
$$

$\qquad\qquad\qquad\qquad\qquad\qquad\qquad\qquad\qquad\qquad\qquad\qquad\qquad\qquad\qquad\qquad\qquad\qquad\square$

### C.3 Preliminaries for the Proofs of Corollaries

We first derive the closed-form expression of the MSEs of EDO and HDB.

**The MSE of EDO estimator:** When $w=1$, the weighted ATE estimator becomes EDO, and its MSE simplifies to,

$$
\mathrm{MSE(EDO)}=\frac{1}{|\mathcal{D}^{(e)}|}\left(\mathbb{E}\left[\frac{r^{(e)}(1,S^{(e)})^2+(\sigma_1^{(e)})^2}{\pi(1\mid S^{(e)})}\right]+\mathbb{E}\left[\frac{r^{(e)}(0,S^{(e)})^2+(\sigma_0^{(e)})^2}{\pi(0\mid S^{(e)})}\right]\right.
$$
$$
\left.-\left(\mathbb{E}[r^{(e)}(1,S^{(e)})]-\mathbb{E}[r^{(e)}(0,S^{(e)})]\right)^2\right). \tag{33}
$$

**The MSE of HDB estimator:** When $w=0$, the weighted estimator becomes HDB, and its MSE can be expressed as,

$$
\mathrm{MSE(HDB)}=\frac{1}{|\mathcal{D}^{(e)}|}\left(\mathbb{E}\left[\frac{r^{(e)}(1,S^{(e)})^2+(\sigma_1^{(e)})^2}{\pi(1\mid S^{(e)})}\right]-\left(\mathbb{E}[r^{(e)}(1,S^{(e)})]\right)^2\right)
$$
$$
+\frac{1}{|\mathcal{D}^{(h)}|}\left(\mathrm{Var}(\mu(S^{(h)})r^{(h)}(0,S^{(h)}))+\mathbb{E}[\mu(S^{(h)})^2]\cdot(\sigma^{(h)})^2\right)+\left(\mathbb{E}[b(S^{(e)})]\right)^2. \tag{34}
$$

Next, we notice that although the MSE of the weighted estimator varies with the weight function $w$, certain components of the MSE remain constant with respect to $w$. To focus on the terms that vary with $w$, we define a new loss function $\mathcal{L}(w)$ as follows:

$$
\mathcal{L}(w)=\frac{1}{|\mathcal{D}^{(e)}|}\left(\mathbb{E}\left[\frac{w(S^{(e)})^2\cdot(r^{(e)}(0,S^{(e)})^2+(\sigma_0^{(e)})^2)}{\pi(0\mid S^{(e)})}\right]\right.
$$
$$
+2\mathbb{E}[r^{(e)}(1,S^{(e)})]\cdot\mathbb{E}[w(S^{(e)})r^{(e)}(0,S^{(e)})]-\left.\left(\mathbb{E}[w(S^{(e)})r^{(e)}(0,S^{(e)})]\right)^2\right)
$$
$$
+\frac{1}{|\mathcal{D}^{(h)}|}\mathrm{Var}((1-w(S^{(h)}))\mu(S^{(h)})r^{(h)}(0,S^{(h)})) \tag{35}
$$
$$
+\frac{1}{|\mathcal{D}^{(h)}|}\mathbb{E}[(1-w(S^{(h)}))^2\mu(S^{(h)})^2]\cdot(\sigma^{(h)})^2+\left(\mathbb{E}\left[(1-w(S^{(e)}))b(S^{(e)})\right]\right)^2,
$$
$$
=\frac{1}{|\mathcal{D}^{(e)}|}(\mathcal{L}_1+\mathcal{L}_2-\mathcal{L}_3)+\frac{1}{|\mathcal{D}^{(h)}|}(\mathcal{L}_4+\mathcal{L}_5)+\mathcal{L}_6,
$$

where $\mathcal{L}_1,\ldots,\mathcal{L}_6$ denote the above six terms, respectively.

**Remark.** To simplify the proof of the corollaries, we consider the case where the contextual variables $S^{(e)},S^{(h)}$ are discrete. Our results can be extended to settings with continuous contextual variables.

## C.4 Proof of Corollary 1

***Optimality of EDO.*** We prove the optimality by contradiction. Suppose the minimal MSE is achieved by a weight function $w$ that does not converge to one. Then, there must exist a non-empty set $\mathcal{S}_1 := \{S : 1 - w(S) \geq \Delta, \Delta > 0\}$. Under this assumption, we derive a lower bound of $\mathcal{L}_5$ in (35):

$$
\begin{aligned}
\mathcal{L}_5 &= \sum_{S^{(h)} \in \mathcal{S}} (1 - w(S^{(h)}))^2 \mu(S^{(h)})^2 p_h(S^{(h)}) \cdot (\sigma^{(h)})^2 \\
&= \sum_{S^{(h)} \in \mathcal{S}_1} (1 - w(S^{(h)}))^2 \cdot \frac{(p_e(S^{(e)}))^2}{p_h(S^{(h)})} \cdot (\sigma^{(h)})^2,
\end{aligned}
\tag{36}
$$

where the third equality follows from the definition of $\mu(S^{(h)}) = \frac{p_e(S^{(e)})}{p_h(S^{(h)})}$.

As for other terms in the objective function $\mathcal{L}(w)$ in (35), under Assumption 5, it is easy to show that

$$
|\mathcal{L}_2| \leq 2|\mathbb{E}[r^{(e)}(1, S^{(e)})] \cdot \mathbb{E}[w(S^{(e)})r^{(e)}(0, S^{(e)})]| \leq 2r_{max}^2,
$$

$$
\mathcal{L}_3 = \left(\mathbb{E}[w(S^{(e)})r^{(e)}(0, S^{(e)})]\right)^2 \leq r_{max}^2,
$$

leading to $\mathcal{L}_2 - \mathcal{L}_3 \geq -3r_{max}^2$. Since $\mathcal{L}_4$ and $\mathcal{L}_6$ are always non-negative, we have

$$
\begin{aligned}
\mathcal{L}(w) &\geq \frac{1}{|\mathcal{D}^{(e)}|}(\mathcal{L}_1 + \mathcal{L}_2 - \mathcal{L}_3) + \frac{1}{|\mathcal{D}^{(h)}|}\mathcal{L}_5 \geq \frac{1}{|\mathcal{D}^{(e)}|}(\mathcal{L}_1 - 3r_{max}^2) + \frac{1}{|\mathcal{D}^{(h)}|}\mathcal{L}_5 \\
&= \frac{1}{|\mathcal{D}^{(e)}|}\mathbb{E}\left[\frac{w(S^{(e)})^2 \cdot (r^{(e)}(0, S^{(e)})^2 + (\sigma_0^{(e)})^2)}{\pi(0 \mid S^{(e)})}\right] - \frac{O(r_{max}^2)}{|\mathcal{D}^{(e)}|} + \frac{1}{|\mathcal{D}^{(h)}|}\mathcal{L}_5
\end{aligned}
$$

Consider the EDO estimator. By setting $w = 1$, its objective function $\mathcal{L}(1)$ becomes

$$
\frac{1}{|\mathcal{D}^{(e)}|}\left(\mathbb{E}\left[\frac{r^{(e)}(0, S^{(e)})^2 + (\sigma_0^{(e)})^2}{\pi(0 \mid S^{(e)})}\right] + 2\mathbb{E}[r^{(e)}(1, S^{(e)})]\mathbb{E}[r^{(e)}(0, S^{(e)})] - \left(\mathbb{E}[r^{(e)}(0, S^{(e)})]\right)^2\right)
$$

and it is smaller than

$$
\frac{1}{|\mathcal{D}^{(e)}|}\left(\mathbb{E}\left[\frac{r^{(e)}(0, S^{(e)})^2 + (\sigma_0^{(e)})^2}{\pi(0 \mid S^{(e)})}\right] + 2\left|\mathbb{E}[r^{(e)}(1, S^{(e)})]\mathbb{E}[r^{(e)}(0, S^{(e)})]\right| + \left(\mathbb{E}[r^{(e)}(0, S^{(e)})]\right)^2\right)
$$

Futhermore,it can be bounded by

$$
\frac{1}{|\mathcal{D}^{(e)}|}\left(\mathbb{E}\left[\frac{r^{(e)}(0, S^{(e)})^2 + (\sigma_0^{(e)})^2}{\pi(0 \mid S^{(e)})}\right] + 3r_{max}^2\right).
$$

where the first inequality follows from the triangle inequality, and the second holds by the condition $|r^{(e)}(\cdot, \cdot)| < r_{max}$ in Assumption 5. Thus,

$$
\mathcal{L}(w) - \mathcal{L}(1) \geq G_1 + \frac{O(r_{max}^2)}{|\mathcal{D}^{(e)}|},
\tag{37}
$$

where

$$
\begin{aligned}
G_1 &= \frac{1}{|\mathcal{D}^{(h)}|}\sum_{S^{(h)} \in \mathcal{S}_1} (1 - w(S^{(h)}))^2 \cdot \frac{(p_e(S^{(e)}))^2}{p_h(S^{(h)})} \cdot (\sigma^{(h)})^2 \\
&\quad - \frac{1}{|\mathcal{D}^{(e)}|}\sum_{S^{(e)} \in \mathcal{S}_1} (1 - w(S^{(e)})^2) \cdot p_e(S^{(e)})\frac{r^{(e)}(0, S^{(e)})^2 + (\sigma_0^{(e)})^2}{\pi(0 \mid S^{(e)})} \\
&= \sum_{S^{(e)} \in \mathcal{S}_1} \frac{(1 - w(S^{(e)}))^2 p_e^2(S^{(e)})}{|\mathcal{D}^{(h)}|p_h(S^{(h)})}\left((\sigma^{(h)})^2 - \delta\frac{r^{(e)}(0, S^{(e)})^2 + (\sigma_0^{(e)})^2}{\pi(0 \mid S^{(e)})}\frac{p_h(S^{(h)})}{p_e(S^{(e)})}\right)
\end{aligned}
$$

where the first equality follows using similar arguments to (36), and the second equality follows from the definition of $\delta = |\mathcal{D}^{(h)}|/|\mathcal{D}^{(e)}|$. Under Assumptions 3 and 5, it is easy to show that the condition $\sigma^{(h)} \gg \epsilon^{-1}\sqrt{\delta}(r_{\max} + \sigma_0^{(e)})$ implies $G_1 > 0$, which in turn ensures that $\mathcal{L}(w) - \mathcal{L}(1) > 0$ for any $w \neq 1$.

Since for any weight function $w$, the objective $\mathcal{L}(w)$ differs from $\mathrm{MSE}(w)$ only by a constant independent of $w$. This in turn ,
$$\mathrm{MSE}(w) > \mathrm{MSE}(1).$$
which leads to a contraction with the property of the optimal weight $w^* = \arg\min_w \mathrm{MSE}(w)$. Therefore,
$$w^* \to 1 \quad \text{for all } S.$$

$\square$

***Oracle property***. We next show that the difference in MSE between our proposed estimator and MSE(EDO) is smaller than MSE(EDO) itself.

Define the $\ell_p$-norm of $\epsilon_a^{(e)}$ (for $a \in \{0, 1\}$)and $\epsilon^{(h)}$ as

$$\|\epsilon_a^{(e)}\|_p := \left( \mathbb{E}\left[ |\epsilon_a^{(e)}|^p \right] \right)^{1/p}, \quad \text{for } p = 2, 4, 8.$$

$$\|\epsilon^{(h)}\|_p := \left( \mathbb{E}\left[ |\epsilon^{(h)}|^p \right] \right)^{1/p}, \quad \text{for } p = 2, 4, 8.$$

Under Assumption 7, we show the proposed estimator achieves the oracle property. By (33), assumptions 3 and 5, MSE(EDO) has such a lower bound.

$$\mathrm{MSE}(\mathrm{EDO}) = \mathrm{MSE}(1) = \frac{\mathrm{Var}(Z^{(e)}(1))}{|\mathcal{D}^{(e)}|} = \Omega\left( \frac{(\sigma_1^{(e)} + \sigma_0^{(e)})^2}{|\mathcal{D}^{(e)}|} \right). \tag{38}$$

According to Theorem 1, we can deduce that

$$
\begin{aligned}
\mathrm{MSE}(\widehat{w}) - \mathrm{MSE}(1) &\leq \mathbb{E}[\widehat{\mathrm{bias}}_U^2(1) - \mathrm{bias}^2(1)] + \mathbb{E}[\widehat{\mathrm{Var}}_U(1) - \mathrm{Var}(1)] + O(\alpha B^2) \\
&= \mathbb{E}[\widehat{\mathrm{Var}}_U(1) - \mathrm{Var}(1)] + O(\alpha B^2).
\end{aligned}
\tag{39}
$$

We next focus on orders of the above two terms, respectively.

By definition, we have

$$\mathbb{E}\left[ \widehat{\mathrm{Var}}_U(1) - \mathrm{Var}(1) \right] = \frac{1}{|\mathcal{D}^{(e)}|} \cdot \mathbb{E}\left[ \widehat{\mathrm{Var}}_U\left( Z^{(e)}(1) \right) - \mathrm{Var}\left( Z^{(e)}(1) \right) \right].$$

For analytical convenience, we express the empirical variance estimator of $Z^{(e)}(1)$ as

$$\widehat{\mathrm{Var}}\left( Z^{(e)}(1) \right) = \frac{1}{|\mathcal{D}^{(e)}|} \sum_{i \in \mathcal{D}^{(e)}} \left( Z_i^{(e)}(1) - \mu \right)^2, \quad \text{where } \mu = \mathbb{E}\left[ Z^{(e)}(1) \right].$$

Then $\mathbb{E}[\widehat{\mathrm{Var}}\left( Z^{(e)}(1) \right)] = \mathrm{Var}\left( Z^{(e)}(1) \right)$.

According to (18), we need to calculate $\mathbb{E}\left[ \widehat{\mathrm{Var}}\left( Z^{(e)}(1) \right) - \mathrm{Var}\left( Z^{(e)}(1) \right) \right]^4$. Through a standard moment expansion, we obtain:

$$
\mathbb{E}\left[ \left( \widehat{\mathrm{Var}}\left( Z^{(e)}(1) \right) - \mathrm{Var}(Z^{(e)}(1)) \right)^4 \right] = O\left( \frac{1}{|\mathcal{D}^{(e)}|^3} \cdot \mathbb{E}\left[ \left( (Z_i^{(e)}(1) - \mu)^2 - \mathrm{Var}(Z^{(e)}(1)) \right)^4 \right] \right)
$$
$$
+ O\left( \frac{1}{|\mathcal{D}^{(e)}|^2} \cdot \left( \mathrm{Var}\left( (Z_i^{(e)}(1) - \mu)^2 \right) \right)^2 \right). \tag{40}
$$

$\mathbb{E}\left[ \left( (Z_i^{(e)}(1) - \mu)^2 - \mathrm{Var}(Z^{(e)}(1)) \right)^4 \right]$ and $\left( \mathrm{Var}\left( (Z_i^{(e)}(1) - \mu)^2 \right) \right)^2$ are of the same order, as their leading terms involve the eighth moments of $\epsilon_0^{(e)}$ and $\epsilon_1^{(e)}$. For the remaining terms, under

Assumptions 3 and 5, they can be uniformly bounded the same order of $r_{\max}$ and $\epsilon$. Therefore, both expressions share the same asymptotic order. So the second term is the dominant one, and the overall expression is of order $O\left(\frac{\mathrm{Var}\left((Z^{(e)}(1)-\mu)^2\right)^2}{|\mathcal{D}^{(e)}|^2}\right)$. If we choose $U$ as

$$U = O\left(\frac{\log^{1/4}|\mathcal{D}^{(e)}| \cdot \mathrm{Var}^{1/2}\left((Z^{(e)}(1)-\mu)^2\right)}{|\mathcal{D}^{(e)}|^{1/4}}\right), \quad \alpha = O\left(|\mathcal{D}^{(e)}|^{-1}\log^{-1}|\mathcal{D}^{(e)}|\right),$$

we can get:

$$\left|\widehat{\mathrm{Var}}\left(Z^{(e)}(1)\right) - \mathrm{Var}\left(Z^{(e)}(1)\right)\right| = O(U) = O\left(\frac{\log^{1/4}|\mathcal{D}^{(e)}|\mathrm{Var}^{1/2}\left((Z_i^{(e)}(1)-\mu)^2\right)}{|\mathcal{D}^{(e)}|^{1/4}}\right).$$

Therefore, we obtain the following bound on the expected deviation between the upper-bound estimator and the true variance of the EDO estimator:

$$\mathbb{E}\left[\widehat{\mathrm{Var}}_U(1) - \mathrm{Var}(1)\right] = O\left(\frac{\log^{1/4}|\mathcal{D}^{(e)}|}{|\mathcal{D}^{(e)}|^{5/4}} \cdot \mathrm{Var}^{1/2}\left((Z^{(e)}(1)-\mu)^2\right)\right). \tag{41}$$

We use the fact that lower-order moments can be controlled by higher-order moments. Specifically, we have

$$
\begin{aligned}
\mathrm{Var}^{1/2}\left(\left(Z^{(e)}(1)-\mu\right)^2\right) &\leq O\left(\mathbb{E}\left[Z^{(e)}(1)^4\right]^{1/2}\right) = O\left(\|Z^{(e)}(1)\|_4^2\right) \\
&\leq O\left(\|\psi_1^{(e)}(O^{(e)})\|_4^2\right) + O\left(\|\psi_0^{(e)}(O^{(e)})\|_4^2\right).
\end{aligned}
\tag{42}
$$

The first equality follows directly from the definition of the $\ell_4$-norm, while the final inequality is a consequence of the triangle inequality for norms and the definition in (9). According to Assumptions 7, 2, and 3, we obtain the following bound:

$$
\begin{aligned}
\mathrm{Var}^{1/2}\left(\left(Z^{(e)}(1)-\mu\right)^2\right) &\leq O\left(\|\psi_1^{(e)}(O^{(e)})\|_4^2 + \|\psi_0^{(e)}(O^{(e)})\|_4^2\right) \\
&\leq O\left(\frac{(r_{\max}+\sigma_1^{(e)}+\sigma_0^{(e)})^2}{\epsilon}\right).
\end{aligned}
\tag{43}
$$

Furthermore, we have:

$$O(\alpha B^2) = O\left(|\mathcal{D}^{(e)}|^{-1}\log|\mathcal{D}^{(e)}|^{-1}B^2\right). \tag{44}$$

We treat $r_{\max}, \sigma_0^{(e)}, \sigma_1^{(e)}$, and $\epsilon$ as constants, and consider the asymptotic regime where $|\mathcal{D}^{(e)}| \to \infty$. Plugging (41), (43), (42), and (44) into (39), we have

$$\mathrm{MSE}(\widehat{w}) - \mathrm{MSE}(1) \leq O\left(\frac{\log^{1/4}|\mathcal{D}^{(e)}|}{|\mathcal{D}^{(e)}|^{5/4}} \cdot \frac{(r_{\max}+\sigma_1^{(e)}+\sigma_0^{(e)})^2}{\epsilon} + |\mathcal{D}^{(e)}|^{-1}\log|\mathcal{D}^{(e)}|^{-1}B^2\right).$$

Comparing it with the MSE of the EDO estimator given in (38), one can deduce that

$$\frac{\mathrm{MSE}(\widehat{w}) - \mathrm{MSE}(1)}{\mathrm{MSE}(1)} \to 0$$

as $|\mathcal{D}^{(e)}| \to \infty$. We conclude that the gap between our proposed method and the EDO baseline vanishes at a faster rate than the MSE of the EDO estimator itself. $\square$

## C.5 Proof of Corollary 2

*Oracle property.* We prove the corollary by contradiction. Suppose that optimal $w(S)$ does not converge uniformly to zero. Then, there must exist a non-empty set $\mathcal{S}_2 := \{S : w(S) \geq \Delta, \Delta > 0\}$.

Under this assumption, we can derive a lower bound for the objective $\mathcal{L}(w^*)$ (35) as follows:

$$\mathcal{L}(w) \geq \frac{1}{|\mathcal{D}^{(e)}|} \left( \sum_{S^{(e)} \in \mathcal{S}} \frac{w(S^{(e)})^2 (\sigma_0^{(e)})^2}{\pi(0 \mid S^{(e)})} p_e(S^{(e)}) - O(r_{\max}^2) \right)$$

$$+ \frac{1}{|\mathcal{D}^{(h)}|} \left( \mathbb{E}[(1 - w(S^{(h)}))^2 \mu(S^{(h)})^2] \cdot (\sigma^{(h)})^2 \right)$$

$$= \frac{1}{|\mathcal{D}^{(e)}|} \left( \sum_{S^{(e)} \in \mathcal{S}_2} \frac{w(S^{(e)})^2 (\sigma_0^{(e)})^2}{\pi(0 \mid S^{(e)})} p_e(S^{(e)}) - O(r_{\max}^2) \right)$$

$$+ \frac{1}{|\mathcal{D}^{(h)}|} \left( \mathbb{E}[(1 - w(S^{(h)}))^2 \mu(S^{(h)})^2] \cdot (\sigma^{(h)})^2 \right)$$

Here, the first inequality holds since $\mathcal{L}_4$ and $\mathcal{L}_6$ in (35) are always non-negative, and the remaining terms can be upper bounded by $O(r_{\max}^2)$ according to Assumption 5. On the other hand, we can derive an upper bound for $\mathcal{L}(0)$, which corresponds to assigning zero weights to all historical data. In this case, the objective reduces to the variance from historical data and the squared bias(35):

$$\mathcal{L}(0) = \frac{1}{|\mathcal{D}^{(h)}|} \left( \mathrm{Var}(\mu(S^{(h)}) r^{(h)}(0, S^{(h)})) + \mathbb{E}[\mu(S^{(h)})^2] \cdot (\sigma^{(h)})^2 + \left( \mathbb{E}[b(S^{(e)})] \right)^2 \right.$$

$$= \frac{1}{|\mathcal{D}^{(h)}|} \left( \mathbb{E}[\mu(S^{(h)}) r^{(h)}(0, S^{(h)})]^2 - (\mathbb{E}[\mu(S^{(h)}) r^{(h)}(0, S^{(h)})])^2 \right)$$

$$+ \frac{1}{|\mathcal{D}^{(h)}|} \mathbb{E}[\mu(S^{(h)})^2](\sigma^{(h)})^2 + \left( \mathbb{E}[b(S^{(e)})] \right)^2 .$$

$$\leq \frac{1}{|\mathcal{D}^{(h)}|} \left( \mathbb{E}[\mu(S^{(h)}) r^{(h)}(0, S^{(h)})]^2 + \mathbb{E}[\mu(S^{(h)})^2] \cdot (\sigma^{(h)})^2 \right) + O(r_{max}^2)$$

$$\leq \frac{1}{|\mathcal{D}^{(h)}|} \left( O\frac{(r_{max}^2)}{\epsilon} + \mathbb{E}[\mu(S^{(h)})^2] \cdot (\sigma^{(h)})^2 \right) + O(r_{max}^2)$$

The reasoning behind this conclusion is as follows. The squared bias term $(\mathbb{E}[b(S^{(e)})])^2$ can be bounded by $O(r_{\max}^2)$. Similarly, the expectation term $\mathbb{E}[\mu(S^{(h)}) r^{(h)}(S^{(h)})]$ is bounded by $O(r_{\max})$, which implies that its square is of order $O(r_{\max}^2)$ as well.

Direct calculations lead to

$$\mathcal{L}(w^*) - \mathcal{L}(0) \geq \frac{1}{|\mathcal{D}^{(e)}|} \sum_{S^{(e)} \in \mathcal{S}_2} \frac{w(S^{(e)})^2 (\sigma_0^{(e)})^2}{\pi(0 \mid S^{(e)})} p_e(S^{(e)}) - O(r_{\max}^2)$$

$$- \frac{1}{|\mathcal{D}^{(h)}|} \mathbb{E}[(1 - (1 - w(S^{(h)}))^2) \mu(S^{(h)})^2](\sigma^{(h)})^2$$

$$= \frac{1}{|\mathcal{D}^{(e)}|} \sum_{S^{(e)} \in \mathcal{S}_2} \frac{w(S^{(e)})^2 (\sigma_0^{(e)})^2}{\pi(0 \mid S^{(e)})} p_e(S) - O(r_{\max}^2)$$

$$- \frac{1}{|\mathcal{D}^{(h)}|} \sum_{S^{(h)} \in \mathcal{S}_2} \left[ 1 - (1 - w(S^{(h)}))^2 \right] \frac{(p_e(S^{(e)}))^2}{p_h(S^{(h)})} \cdot (\sigma^{(h)})^2$$

$$= \sum_{S^{(e)} \in \mathcal{S}_2} \frac{w(S)^2 p_e(S^{(e)})}{|\mathcal{D}^{(e)}| \pi(0 \mid S^{(e)})} \cdot$$

$$\left( (\sigma_0^{(e)})^2 - \frac{(1 - (1 - w(S^{(e)}))^2) p_e(S^{(e)})}{w(S^{(h)})^2 p_h(S^{(h)}) \delta} \pi(0 \mid S^{(e)})(\sigma^{(h)})^2 \right) - O(r_{\max}^2),$$

where the first equality directly follows from the calculation of expectation, and the second equality uses $\mu(S) = \frac{p_e(S^{(e)})}{p_h(S^{(h)})}$ and $\delta = |\mathcal{D}^{(h)}|/|\mathcal{D}^{(e)}|$.

According to the assumption

$$\sigma^{(e)} \gg \epsilon^{-1/2} \left( \frac{\sigma^{(h)}}{\sqrt{\delta}} + \sqrt{|\mathcal{D}^{(e)}|} \cdot r_{\max} \right),$$

the order of the first term in $\mathcal{L}(w) - \mathcal{L}(0)$ dominates the term of order $\mathrm{O}(r_{\max}^2)$, leading to

$$\mathcal{L}(w) - \mathcal{L}(0) > 0$$

for any $w$. Hence, the lower bound of $\mathcal{L}(w)$ strictly exceeds the upper bound of $\mathcal{L}(0)$. This contradicts the optimality of $w^*$, since $w^* = \arg\min_w \mathcal{L}(w)$. Therefore, the assumption must be false, and it follows that $w^* \to 0$ for all $S$. In the following, we discuss additional conditions under which the gap between the MSE of our proposed method and that of the HDB estimator becomes negligible relative to the MSE of the HDB estimator itself.

$\square$

*Optimality of HDB*. At first,We give the additional assumption:

**Assumption 8.** *The variance* $\sigma_1^{(e)}$ *satisfies:*

$$\sigma_1^{(e)} \gg \max\left\{\log^{1/4}|\mathcal{D}^{(e)}||\mathcal{D}^{(e)}|^{1/4}(\sigma_0^{(e)} + r_{\max}),\ \log^{1/4}|\mathcal{D}^{(h)}||\mathcal{D}^{(h)}|^{1/4}\sqrt{\frac{|\mathcal{D}^{(e)}|}{|\mathcal{D}^{(h)}|}}(\sigma^{(h)} + r_{\max})\right\}.$$

We begin by recalling the asymptotic order of the mean squared error (MSE) of the HDB estimator given in (34):

$$
\begin{aligned}
\mathrm{MSE}(\mathrm{HDB}) &= \mathrm{bias}^2(0) + \frac{\mathrm{Var}\left(\psi_1^{(e)}(O^{(e)})\right)}{|\mathcal{D}^{(e)}|} + \frac{\mathrm{Var}\left(\psi_0^{(h)}(O^{(h)})\right)}{|\mathcal{D}^{(h)}|} \\
&= \mathrm{bias}^2(0) + \Omega\left(\frac{(\sigma_1^{(e)})^2}{|\mathcal{D}^{(e)}|}\right) + \Omega\left(\frac{(\sigma^{(h)})^2}{|\mathcal{D}^{(h)}|}\right).
\end{aligned}
\tag{45}
$$

The difference between our proposed method and the MSE of the HDB estimator can be decomposed according to Theorem 1,

$$\mathrm{MSE}(\widehat{w}) - \mathrm{MSE}(\mathrm{HDB}) \leq \mathbb{E}\left[\widehat{\mathrm{bias}}_U^2(0) - \mathrm{bias}^2(0)\right] + \mathbb{E}\left[\widehat{\mathrm{Var}}_U(\mathrm{HDB}) - \mathrm{Var}(\mathrm{HDB})\right] + O(\alpha B^2).$$

For the variance difference component, we have:

$$
\begin{aligned}
\mathbb{E}\left[\widehat{\mathrm{Var}}_U(\mathrm{HDB}) - \mathrm{Var}(\mathrm{HDB})\right] &= \frac{1}{|\mathcal{D}^{(e)}|} \cdot \mathbb{E}\left[\widehat{\mathrm{Var}}\left(\psi_1^{(e)}(O^{(e)})\right) - \mathrm{Var}\left(\psi_1^{(e)}(O^{(e)})\right)\right] \\
&\quad + \frac{1}{|\mathcal{D}^{(h)}|} \cdot \mathbb{E}\left[\widehat{\mathrm{Var}}_U\left(\psi_0^{(h)}(O^{(h)})\right) - \mathrm{Var}\left(\psi_0^{(h)}(O^{(h)})\right)\right].
\end{aligned}
$$

Similar to the derivation of (41), according to the fourth-moment version of Markov's inequality(17) ,(18),and assumption8, we can take:

$$U_1 = O\left(\frac{\log^{1/4}|\mathcal{D}^{(e)}| \cdot \mathrm{Var}^{1/2}\left((\psi_1^{(e)}(O^{(e)}) - \mu_1)^2\right)}{|\mathcal{D}^{(e)}|^{1/4}}\right) \quad \alpha_1 = O\left(|\mathcal{D}^{(e)}|^{-1}\log^{-1}|\mathcal{D}^{(e)}|\right),$$

$$U_2 = O\left(\frac{\log^{1/4}|\mathcal{D}^{(h)}| \cdot \mathrm{Var}^{1/2}\left((\psi_0^{(h)}(O^{(h)}) - \mu_2)^2\right)}{|\mathcal{D}^{(h)}|^{1/4}}\right) \quad \alpha_2 = O\left(|\mathcal{D}^{(h)}|^{-1}\log^{-1}|\mathcal{D}^{(h)}|\right),$$

where $\mu_1 = \mathbb{E}\left[\psi_1^{(e)}(O^{(e)})\right]$ and $\mu_2 = \mathbb{E}\left[\psi_0^{(h)}(O^{(h)})\right]$ denote the corresponding population means.

Furthermore, the order of $\mathbb{E}\left[\widehat{\mathrm{Var}}_U(\mathrm{HDB}) - \mathrm{Var}(\mathrm{HDB})\right]$ can be derived as follows:

$$
\begin{aligned}
\mathbb{E}\left[\widehat{\mathrm{Var}}_U(\mathrm{HDB}) - \mathrm{Var}(\mathrm{HDB})\right] &= O\left(\frac{\log^{1/4}|\mathcal{D}^{(e)}| \cdot \mathrm{Var}^{1/2}\left((\psi_1^{(e)}(O^{(e)}) - \mu_1)^2\right)}{|\mathcal{D}^{(e)}|^{5/4}}\right) \\
&\quad + O\left(\frac{\log^{1/4}|\mathcal{D}^{(h)}| \cdot \mathrm{Var}^{1/2}\left((\psi_0^{(h)}(O^{(h)}) - \mu_2)^2\right)}{|\mathcal{D}^{(h)}|^{5/4}}\right).
\end{aligned}
\tag{46}
$$

For the bias difference component, we directly compute the decomposition as:

$$\mathbb{E}\left[\widehat{\text{bias}}_U^2(0) - \text{bias}^2(0)\right] = \mathbb{E}\left[\left(\widehat{\text{bias}}_U(0) - \text{bias}(0)\right)^2\right] + 2 \cdot \text{bias}(0) \cdot \mathbb{E}\left[\widehat{\text{bias}}_U(0) - \text{bias}(0)\right].$$

More precisely, the bias estimation error can be decomposed into two components,

$$\frac{1}{|\mathcal{D}^{(h)}|}\sum_{i=1}^{|\mathcal{D}^{(h)}|}\psi_{0,i}^{(h)}(O^{(h)}) - \mathbb{E}[\psi_0^{(h)}(O^{(h)})] \quad \frac{1}{|\mathcal{D}^{(e)}|}\sum_{j=1}^{|\mathcal{D}^{(e)}|}\psi_{0,j}^{(e)}(O^{(e)}) - \mathbb{E}[\psi_0^{(e)}(O^{(e)})],$$

each of which can be controlled via (40):

$$U_{b1} = O\left(\frac{\log^{1/4}|\mathcal{D}^{(e)}| \cdot \text{Var}^{1/2}\left(\psi_0^{(e)}(O^{(e)})\right)}{|\mathcal{D}^{(e)}|^{1/4}}\right) \quad \alpha_3 = O\left(|\mathcal{D}^{(e)}|^{-1}\log^{-1}|\mathcal{D}^{(e)}|\right)$$

$$U_{b2} = O\left(\frac{\log^{1/4}|\mathcal{D}^{(h)}| \cdot \text{Var}^{1/2}\left(\psi_0^{(h)}(O^{(h)})\right)}{|\mathcal{D}^{(h)}|^{1/4}}\right) \quad \alpha_4 = O\left(|\mathcal{D}^{(h)}|^{-1}\log^{-1}|\mathcal{D}^{(h)}|\right).$$

Denote $U_b = U_{b1} + U_{b2}$, it is easy to deduce that $\text{bias}(0) \ll U_b$. Therefore, we obtain:

$$U_b = O\left(\frac{\log^{1/4}|\mathcal{D}^{(e)}|\,\text{Var}^{1/2}(\psi_0^{(e)}(O^{(e)}))}{|\mathcal{D}^{(e)}|^{1/4}}\right) + O\left(\frac{\log^{1/4}|\mathcal{D}^{(h)}|\,\text{Var}^{1/2}(\psi_0^{(h)}(O^{(h)}))}{|\mathcal{D}^{(h)}|^{1/4}}\right).$$

Next, we can get:

$$\mathbb{E}\left[\widehat{\text{bias}}_U^2(0) - \text{bias}^2(0)\right] = O(U_b^2) + O\big(\text{bias}(0) \cdot U_b\big) = O(U_b^2)$$

$$= O\left(\frac{\log^{1/2}|\mathcal{D}^{(e)}|\,\text{Var}(\psi_0^{(e)}(O^{(e)}))}{|\mathcal{D}^{(e)}|^{1/2}}\right) + O\left(\frac{\log^{1/2}|\mathcal{D}^{(h)}|\,\text{Var}(\psi_0^{(h)}(O^{(h)}))}{|\mathcal{D}^{(h)}|^{1/2}}\right). \quad (47)$$

By summing all $\alpha_i$'s, we obtain

$$\alpha = \sum_{i=1}^{4}\alpha_i = O\left(|\mathcal{D}^{(e)}|^{-1}\log^{-1}|\mathcal{D}^{(e)}| + |\mathcal{D}^{(h)}|^{-1}\log^{-1}|\mathcal{D}^{(h)}|\right). \quad (48)$$

If Assumption 8 holds, namely,

$$\sigma_1^{(e)} \gg \max\left\{\log^{1/4}|\mathcal{D}^{(e)}| \cdot |\mathcal{D}^{(e)}|^{1/4}(\sigma_0^{(e)} + r_{\max})\epsilon^{-1/2},\right.$$

$$\left.\log^{1/4}|\mathcal{D}^{(h)}| \cdot |\mathcal{D}^{(h)}|^{1/4}\sqrt{\frac{|\mathcal{D}^{(e)}|}{|\mathcal{D}^{(h)}|}}(\sigma^{(h)} + r_{\max})\epsilon^{-1/2}\right\}. \quad (49)$$

This condition is equivalent to:

$$\frac{(\sigma_1^{(e)})^2}{|\mathcal{D}^{(e)}|\epsilon} \gg \max\left\{\frac{\log^{1/2}|\mathcal{D}^{(e)}|}{|\mathcal{D}^{(e)}|^{1/2}}(\sigma_0^{(e)} + r_{\max})^2\epsilon^{-1}, \frac{\log^{1/2}|\mathcal{D}^{(h)}|}{|\mathcal{D}^{(h)}|^{1/2}} \cdot (\sigma_0^{(h)} + r_{\max})^2\epsilon^{-1}\right\}.$$

Furthermore, we know that:

$$\text{Var}\left(\psi_0^{(e)}(O^{(e)})\right) \le O(\frac{(\sigma_0^{(e)} + r_{\max})^2}{\epsilon}), \text{Var}\left(\psi_0^{(h)}(O^{(h)})\right) \le O(\frac{(\sigma^{(h)} + r_{\max})^2}{\epsilon}). \quad (50)$$

By combining the bounds in (45), (47), (48), and (50), under Assumption 8, and comparing it with the order in (45), we have:

$$\text{MSE}(\widehat{w}) - \text{MSE}(0) \ll O\left(\frac{\log^{1/4}|\mathcal{D}^{(e)}|}{|\mathcal{D}^{(e)}|^{5/4}} \cdot \frac{(r_{\max} + \sigma_1^{(e)})^2}{\epsilon} + \frac{\log^{1/4}|\mathcal{D}^{(h)}|}{|\mathcal{D}^{(h)}|^{5/4}} \cdot \frac{(r_{\max} + \sigma^{(h)})^2}{\epsilon}\right.$$

$$\left. + \left(\frac{B^2}{\log|\mathcal{D}^{(e)}||\mathcal{D}^{(e)}|} + \frac{B^2}{\log|\mathcal{D}^{(h)}||\mathcal{D}^{(h)}|}\right) + (47)\right),$$

which is much smaller than $\mathrm{MSE}(0)$ such that

$$\frac{\mathrm{MSE}(\widehat{w}) - \mathrm{MSE}(0)}{\mathrm{MSE}(0)} \to 0$$

This completes the proof. $\qquad\square$

## C.6 Proof of Corollary 3

***Oracle property***. Recall that in (27),

$$\mathrm{MSE}(w) = \frac{\mathrm{Var}(Z^{(e)}(w))}{|\mathcal{D}^{(e)}|} + \frac{\mathrm{Var}(Z^{(h)}(w))}{|\mathcal{D}^{(h)}|} + \left( \mathbb{E}\left[ (1 - w(S^{(e)})b(S^{(e)})] \right) \right)^2.$$

Since Minimum Variance Estimator (MVE) is designed to minimize total variance, it suffices to show that the squared bias (the third term) is negligible compared to the variance terms (the first two). In this case, the bias contributes asymptotically little to the overall error.

Therefore, our analysis focuses on deriving a nontrivial lower bound for the variance component. For the variance contribution from the experimental data, we have the lower bound from Lemma 2, and Assumptions 3 and 5, one can derive that

$$\frac{\mathrm{Var}(Z^{(e)}(w))}{|\mathcal{D}^{(e)}|} \geq \frac{1}{|\mathcal{D}^{(e)}|} \cdot \mathbb{E}\left[ \frac{w(S^{(e)})^2}{\pi(0 \mid S^{(e)})} \right] \cdot (\sigma_0^{(e)})^2.$$

This inequality holds by regrouping the terms involving $r^{(e)}(0, S^{(e)})$ and $r^{(e)}(1, S^{(e)})$, and applying a basic inequality that ensures the non-negativity of the remaining components. Similarly, for the variance contribution from the historical data, we obtain the following lower bound from Lemma 3 and Assumptions 3 and 5,

$$\frac{\mathrm{Var}(Z^{(h)}(w))}{|\mathcal{D}^{(h)}|} \geq \frac{1}{|\mathcal{D}^{(h)}|} \cdot \mathbb{E}\left[ (1 - w(S^{(h)}))^2 \mu(S^{(h)})^2 \right] \cdot (\sigma^{(h)})^2.$$

Note that both $\mathbb{E}\left[ \frac{w(S^{(e)})^2}{\pi(0|S^{(e)})} \right]$ and $\mathbb{E}\left[ (1 - w(S^{(h)}))^2 \mu(S^{(h)})^2 \right]$ can not be simultaneously zero for any nontrivial weight function w. Therefore, the total variance is lower bounded by:

$$\frac{\mathrm{Var}(Z^{(e)}(w))}{|\mathcal{D}^{(e)}|} + \frac{\mathrm{Var}(Z^{(h)}(w))}{|\mathcal{D}^{(h)}|} \geq \min \left\{ \frac{(\sigma^{(e)})^2}{|\mathcal{D}^{(e)}|}, \frac{(\sigma^{(h)})^2}{|\mathcal{D}^{(h)}|} \right\}, \quad \text{for any } w.$$

For the bias term, the condition

$$|b(S^{(e)})| \ll \min \left( \frac{\sigma^{(e)}}{\sqrt{|\mathcal{D}^{(e)}|}}, \frac{\sigma^{(h)}}{\sqrt{|\mathcal{D}^{(h)}|}} \right),$$

implies

$$\left( \mathbb{E}\left[ (1 - w(S^{(e)}))b(S^{(e)}) \right] \right)^2 \ll \min \left\{ \frac{(\sigma^{(e)})^2}{|\mathcal{D}^{(e)}|}, \frac{(\sigma^{(h)})^2}{|\mathcal{D}^{(h)}|} \right\}, \quad \text{for any } w.$$

In this scenario, minimizing MSE is equivalent to minimizing the total variance. Therefore, the optimal weight $w^*$ is given by:

$$w^* = \arg\min_w \left( \frac{\mathrm{Var}(Z^{(e)}(w))}{|\mathcal{D}^{(e)}|} + \frac{\mathrm{Var}(Z^{(h)}(w))}{|\mathcal{D}^{(h)}|} \right), \tag{51}$$

which corresponds exactly to the MVE formulation. This completes the proof of the corollary. $\quad\square$

***Optimality of MVE***. The conditions required here are identical to those stated in Assumption 8. In this setting, the order of the MSE of the MVE estimator is given by

$$\mathrm{MSE}(\mathrm{MVE}) = \mathrm{MSE}(\mathrm{w}^*) = \Omega \left( \frac{\sigma_0^{(e)} + \sigma_1^{(e)})^2}{|\mathcal{D}^{(e)}|} \right) + \Omega \left( \frac{(\sigma^{(h)})^2}{|\mathcal{D}^{(h)}|} \right). \tag{52}$$

We can derive the following MSE gap by plugging in $w^*$ in (51) in Theorem 1,

$$\text{MSE}(\widehat{w}) - \text{MSE}(w^*) \leq \mathbb{E}\left[\widehat{\text{bias}}_U^2(w^*) - \text{bias}^2(w^*)\right] + \mathbb{E}\left[\widehat{\text{Var}}_U(w^*) - \text{Var}(w^*)\right] + O(\alpha B^2).$$

For the bias component, we further obtain:

$$\mathbb{E}\left[\widehat{\text{bias}}_U^2(w^*) - \text{bias}^2(w^*)\right] = O(U_b^2) + O\left(\text{bias}(w^*) \cdot U_b\right)$$

by using the 4-th moment Markov equality similarly to derivations in Sections C.4-C.5, where

$$U_b = O\left(\frac{\log^{1/4}|\mathcal{D}^{(e)}|}{|\mathcal{D}^{(e)}|^{1/4}}\sqrt{\left(\frac{r_{\max}^2 + (\sigma_0^{(e)})^2}{\epsilon}\right)} + \frac{\log^{1/4}|\mathcal{D}^{(h)}|}{|\mathcal{D}^{(h)}|^{1/4}}\sqrt{\left(\frac{r_{\max}^2 + (\sigma^{(h)})^2}{\epsilon}\right)}\right), \quad (53)$$

and $\alpha_1 = O\left(|\mathcal{D}^{(e)}|^{-1}\log^{-1}|\mathcal{D}^{(e)}| + |\mathcal{D}^{(h)}|^{-1}\log^{-1}|\mathcal{D}^{(h)}|\right)$.
Under the small shift assumption, the bias term satisfies $|\text{bias}(w^*)| \ll U_b$. Therefore, we obtain:

$$\mathbb{E}\left[\widehat{\text{bias}}_U^2(w^*) - \text{bias}^2(w^*)\right] = O\left(\frac{\log^{1/2}|\mathcal{D}^{(e)}|}{|\mathcal{D}^{(e)}|^{1/2}}\left(\frac{r_{\max}^2 + (\sigma_0^{(e)})^2}{\epsilon}\right)\right.$$
$$\left. + \frac{\log^{1/2}|\mathcal{D}^{(h)}|}{|\mathcal{D}^{(h)}|^{1/2}}\left(\frac{r_{\max}^2 + (\sigma^{(h)})^2}{\epsilon}\right)\right) \quad (54)$$

For the second term in MSE gap, similar to the derivation for (41), it is easy to deduce that

$$\mathbb{E}\left[\widehat{\text{Var}}_U(w^*) - \text{Var}(w^*)\right] \quad (55)$$
$$= O(U) = \left(\frac{(r_{\max} + \sigma_0^{(e)} + \sigma_1^{(e)})^2}{\epsilon \cdot |\mathcal{D}^{(e)}|^{5/4}} \cdot \log^{1/4}|\mathcal{D}^{(e)}| + \frac{(r_{\max} + \sigma^{(h)})^2}{\epsilon \cdot |\mathcal{D}^{(h)}|^{5/4}} \cdot \log^{1/4}|\mathcal{D}^{(h)}|\right),$$

$\alpha_2 = O\left(|\mathcal{D}^{(e)}|^{-1}\log^{-1}|\mathcal{D}^{(e)}| + |\mathcal{D}^{(h)}|^{-1}\log^{-1}|\mathcal{D}^{(h)}|\right)$. Simple calculations show that the decay rate of $\mathbb{E}\left[\widehat{\text{Var}}_U(w^*) - \text{Var}(w^*)\right]$ is faster than that in $\text{MSE}(w^*)$ in (52). The term

$$\alpha B^2 = (\alpha_1 + \alpha_2)B^2 = O\left(B^2|\mathcal{D}^{(e)}|^{-1}\log^{-1}|\mathcal{D}^{(e)}| + B^2|\mathcal{D}^{(h)}|^{-1}\log^{-1}|\mathcal{D}^{(h)}|\right) \quad (56)$$

which also vanishes faster than the leading MSE terms in (52).
Given the additional assumptions that

$$\frac{(\sigma_1^{(e)})^2}{|\mathcal{D}^{(e)}|\epsilon} \gg \max\left\{\frac{\log^{1/2}|\mathcal{D}^{(e)}|}{|\mathcal{D}^{(e)}|^{1/2}}(\sigma_0^{(e)} + r_{\max})^2\epsilon^{-1}, \frac{\log^{1/2}|\mathcal{D}^{(h)}|}{|\mathcal{D}^{(h)}|} \cdot (\sigma^{(h)} + r_{\max})^2\epsilon^{-1}\right\},$$

which is asymptotically smaller than
Furthermore, by summing all three gap terms in (54)-(56),

$$\text{MSE}(\widehat{w}) - \text{MSE}(w^*) = O\left(\frac{\log^{1/4}|\mathcal{D}^{(e)}|}{|\mathcal{D}^{(e)}|^{5/4}}\frac{(r_{\max} + \sigma_0^{(e)} + \sigma_1^{(e)})^2}{\epsilon} + \right.$$

$$\left.\frac{\log^{1/4}|\mathcal{D}^{(h)}|}{|\mathcal{D}^{(h)}|^{5/4}}\frac{(r_{\max} + \sigma^{(h)})^2}{\epsilon} + \left(\frac{B^2}{\log|\mathcal{D}^{(e)}||\mathcal{D}^{(e)}|} + \frac{B^2}{\log|\mathcal{D}^{(h)}||\mathcal{D}^{(h)}|}\right) + (54)\right).$$

It follows directly that

$$\frac{\text{MSE}(\widehat{w}) - \text{MSE}(\text{MVE})}{\text{MSE}(\text{MVE})} \to 0$$

as the sample size goes to infinity. This completes the proof. $\qquad \square$

## C.7 Proof of Corollary 4

***Oracle property***. We prove the corollary by contradiction, considering both the **moderate shift** and **large shift** cases.

(1)**Moderate shift.** We define the set $\mathcal{S}_3 := \{S : 1 - w(S) \geq \frac{\Delta}{\epsilon}, \Delta > 0\}$. We consider the experiment-related component for the objective function in (35),

According to the condition $|r^{(e)}(0, S^{(e)})| \leq r_{\max}$, $w(S^{(e)}) \in [0, 1]$, $\pi(0 \mid S^{(e)}) \geq \epsilon$ for any $S^{(e)}$ and triangle inequality, we obtain the upper bound

$$|\mathcal{L}_{\exp}(w)| \leq \frac{1}{|\mathcal{D}^{(e)}|} O\left(\frac{r_{\max}^2 + (\sigma_0^{(e)})^2}{\epsilon}\right).$$

Similarly, the historical-related component can be bounded by

$$\begin{aligned}
\mathcal{L}_{\text{his}}(w) &= \frac{1}{|\mathcal{D}^{(h)}|}\left(\mathcal{L}_4 + \mathcal{L}_5\right) \\
&= \frac{1}{|\mathcal{D}^{(h)}|} \operatorname{Var}\left((1 - w(S^{(h)}))\, \mu(S^{(h)})\, r^{(h)}(0, S^{(h)})\right) \\
&\quad + \frac{1}{|\mathcal{D}^{(h)}|} \mathbb{E}\big[(1 - w(S^{(h)}))^2\, \mu(S^{(h)})^2\big]\, (\sigma^{(h)})^2 \leq \frac{1}{|\mathcal{D}^{(h)}|} O\left(\frac{r_{\max}^2 + (\sigma^{(h)})^2}{\epsilon}\right).
\end{aligned}$$

For the bias item:

$$\mathcal{L}_6 = \left|\mathbb{E}\left[(1 - w(S^{(e)}))b(S^{(e)})\right]\right|^2 \geq \Big(\sum_{S^{(e)} \in \mathcal{S}_3} p_e(S^{(e)})\frac{\Delta}{\epsilon} \cdot |b(S^{(e)})|\Big)^2 \geq \Big(\sum_{S^{(e)} \in \mathcal{S}_3} \Delta \cdot |b(S^{(e)})|\Big)^2,$$

where the inequality follows from the fact that $b(S^{(e)})$ is sign-consistent over $\mathcal{S}$, either non-negative or non-positive for all $S^{(e)} \in \mathcal{S}$, and the coefficient of $|b(S^{(e)})|$ is strictly positive in $\mathcal{S}_3$ .

As for $\mathcal{L}(1)$, it can be upper bounded as:

$$\mathcal{L}(1) \leq O\left(\frac{r_{\max}^2 + (\sigma_1^{(e)})^2}{\epsilon|\mathcal{D}^{(e)}|}\right).$$

In the moderate case, for any $S^{(e)}$, we have $|b(S^{(e)})| \gg \frac{1}{\sqrt{\epsilon}}\left(\frac{\sigma^{(e)}+r_{\max}}{\sqrt{|\mathcal{D}^{(e)}|}} + \frac{\sigma^{(h)}+r_{\max}}{\sqrt{|\mathcal{D}^{(h)}|}}\right)$, so

$$\mathcal{L}_6 \geq \Big(\sum_{S^{(e)} \in \mathcal{S}_3} \Delta \cdot |b(S^{(e)})|\Big)^2 \gg \frac{1}{|\mathcal{D}^{(e)}|} \cdot \left(\frac{r_{\max}^2 + (\sigma_0^{(e)})^2}{\epsilon}\right) + \frac{1}{|\mathcal{D}^{(h)}|} \cdot \left(\frac{r_{\max}^2 + (\sigma^{(h)})^2}{\epsilon}\right),$$

then the bias dominates the total MSE. Compare the order of $\mathcal{L}(w)$ with $\mathcal{L}(1)$, we have

$$\mathcal{L}(w) = \mathcal{L}_{\exp}(w) + \mathcal{L}_{\text{his}}(w) + \mathcal{L}_6 \gg \mathcal{L}(1),$$

for any $w \neq 1$, which leads to a contradiction. Hence $\operatorname{MSE}(w)$ in $w$, the optimal weight function $w^* = \arg\min_w \operatorname{MSE}(w)$ satisfies

$$w \to 1 \quad \text{for all } S.$$

(2) **Large shift.** The proof follows the same reasoning as in the moderate shift case by using the condition

$$\Big(\sum_{S^{(e)} \in \mathcal{S}_3} \Delta \cdot |b(S^{(e)})|\Big)^2 \geq \frac{\log|\mathcal{D}^{(e)}|}{|\mathcal{D}^{(e)}|}\left(\frac{r_{\max}^2 + (\sigma_0^{(e)})^2}{\epsilon}\right) + \frac{|\log \mathcal{D}^{(h)}|}{|\mathcal{D}^{(e)}|}\left(\frac{r_{\max}^2 + (\sigma^{(h)})^2}{\epsilon}\right).$$

In this case, where the EDO method remains optimal, the gap between the MSE of our proposed estimator and that of EDO becomes asymptotically negligible. This completes the proof. □

***Optimality of EDO***. **Conditions for small MSE difference** Since the assumption and proof strategy are the same as those used in C.4 , we omit the detailed proof of this result for brevity. □

# D  Limitation

The current paper considers settings without carryover effects where each action affects only its immediate reward and does not influence future outcomes. However, in many real-world applications, treatments are sequentially assigned over time, and such carryover effects can arise [104, 131]. This represents a potential limitation of our work, as it does not account for such effects. In scenarios with carryover effects, the weight function for data integration may depend not only on contextual variables but also vary over time. Determining an optimal time-dependent weight function remains an open question, which we leave for future research.

