# OpenReview forum: "Pessimistic Data Integration for Policy Evaluation"
_NeurIPS.cc/2025/Conference — NeurIPS 2025 poster_

### Official Review · Reviewer_b6bo · 2025-06-24

**Clarity:** 3
**Significance:** 2
**Originality:** 2
**Rating:** 5
**Confidence:** 2

**Summary:**

The paper showcases a reweighing procedure to include historical data (not randomized) together with randomized data to compute an average treatment effect.

Apart from the standard assumptions on historical data they add a possible shift on the reward function between the experiments and the historical data.

EDO is the ATE computed only data that has been randomized and then a proxy for the ATE using historical data, where the remaining expected value of the reward is computed with the experimental data.

The final estimator is a convex combination between the ATE and the estimator computed using the historical data

**Questions:**

No questions.

**Ethical Concerns:**

["NO or VERY MINOR ethics concerns only"]

**Final Justification:**

It is a well executed paper although heavily based on Li et al. [3]. However because it is an interesting and well presented result I decided to maintain my score.

**Limitations:**

yes, albeit in the appendix.

**Paper Formatting Concerns:**

No paper formatting concerns.

**Quality:**

3

**Strengths And Weaknesses:**

Strengths
- The assumptions are not too crazy and the paper presents a bound on the best possible MSE attained.
- They give sufficient conditions to guarantee optimality of the EDO
- Good numerical simulations

Weaknesses

- I do not understand why the new estimator is an improvement over the ATE. It is clearly biased and only in a special case will recover the ATE.
- They do not perform a characterization for when the EDO is optimal

---

> ### Author Rebuttal · Authors · 2025-07-30
>
> Thank you for your thoughtful and critical assessment. Many of your comments will help us produce a more readable and self-contained version of the paper. Below, we address each of your specific concerns in turn.
>
> * **Improvement over the ATE.**
> Thank you for the insightful comment. We respectfully clarify that the proposed estimator is designed to improve the mean squared error (MSE) of the standard ATE estimator by integrating experimental data with potentially large-sample historical data. The core idea of our method is to adaptively combine potentially biased but low-variance historical estimators with unbiased experimental estimators—striking a principled balance between bias and variance in finite-sample settings. This leads to consistently improved performance across a broad range of practical scenarios.
>
>
>   - When the bias introduced by the historical data is small or negligible, the proposed estimator effectively leverages both datasets, benefiting from the larger sample size and achieving a smaller MSE compared to the experimental-data-only (EDO) estimator.
>
>   - When the bias is moderate, our method balances bias and variance in a way that reduces overall MSE. Minimizing MSE—reflecting the fundamental trade-off between bias and variance—is widely recognized as a more practical and robust objective in real-world applications. While the proposed estimator may introduce some bias, it is carefully calibrated to ensure that the reduction in variance outweighs the increase in bias, particularly in settings with distributional heterogeneity between datasets.
>
>   - When the bias is substantial, the learned weight on historical data approaches zero, causing the proposed estimator to behave similarly to the EDO estimator. This adaptive behavior ensures robustness even under significant posterior shifts.
>
>   - As demonstrated in our toy example (Table 2 on Page 5), even a slightly biased estimator can outperform the standard ATE estimator significantly in terms of MSE. Our theoretical analysis rigorously characterizes this trade-off, and our empirical evaluations across diverse scenarios consistently show that the proposed method achieves near-oracle performance by effectively managing the bias–variance balance.
>
> * **Perform a characterization for when the EDO is optimal.**
>   We thank the reviewer for the helpful comment. Below, we provide a more detailed explanation of the conditions under which the EDO estimator tends to perform best.
>
>   In Section 4 of the paper, we analyze five representative scenarios and formally characterize the settings where the experimental-data-only (EDO) estimator achieves the lowest mean squared error (MSE). These conditions are explicitly stated in:
>   - Corollary 1 (Page 7): In the presence of heavy-tailed historical rewards, EDO becomes optimal because the high variability in historical data outweighs its variance-reducing benefit. Under mild regularity conditions, we show that the optimal weight function converges to 1, effectively defaulting to EDO.
>   - Corollary 4 (Page 7): When there is moderate or large posterior shift between historical and experimental data (i.e., substantial discrepancies in conditional outcome distributions), our analysis demonstrates that the optimal weight again converges to 1. In such settings, incorporating biased historical data would result in significant estimation error, making EDO the preferred choice.
>
>   In both cases, we derive upper bounds for the excess MSE of our proposed estimator relative to EDO, and prove that this excess is negligible under the specified conditions. These results demonstrate that the proposed method is adaptive: it defaults to EDO when EDO is indeed optimal.
>
> Thank you again for your thoughtful review. We look forward to further discussion.

---

> > ### Comment · Reviewer_b6bo · 2025-08-08
> >
> > Thanks for the clarifications, I decided to maintain my score.

---

> > > ### Author Response · Authors · 2025-08-08
> > >
> > > Thank you very much for your positive and constructive feedback. We truly appreciate the time and effort you dedicated to reviewing our work.

---

### Official Review · Reviewer_XVXr · 2025-06-28

**Clarity:** 3
**Significance:** 3
**Originality:** 3
**Rating:** 3
**Confidence:** 4

**Summary:**

This paper studies the optimal integration of experimental data (with both treated and control units) and historical data (only control units, subject to covariate, policy, and posterior shift compared with the experimental data). The authors propose to use a weighted combination of the dataset, where the weights are conceptualized as a "policy" that determines the importance of each data point. Then, the policy is learned by minimizing an estimated combination of MSE. The resulting estimator is analyzed in several scenarios, and experiments show the superior performance of the resulting estimator.

**Questions:**

1. It's a bit confusing to use $\hat{\mathbb{E}}_n$ for both average in experimental data and average in historical data. It might be helpful to use different symbols for the two sample averages.

2. Theorem 1 appears incorrect to me. The issue is that on Assumption 1, one doesn't have the claimed equation $$bias(\hat{w}) + Var(\hat{w}) \leq \hat{bia}(\hat{w}) + \hat{Var}(\hat{w})$$ in the proof, since Assumption 1 only concerns a single, fixed $w$, whereas $\hat{w}$ is data-dependent. Indeed, in the literature of pessimism one typically needs uniform concentration for the estimated upper bound (i.e., uniformly valid for all $w$ with high probability). Also in the proof of Theorem 1, it's unclear what $MSE(\hat{w})$ means -- is the MSE marginalized over everything?

3. Apart from MSE, inference is also important for A/B testing applications. Could you comment on inference with the proposed estimators?

**Ethical Concerns:**

["NO or VERY MINOR ethics concerns only"]

**Final Justification:**

authors have provided responses and revisions to my questions.

**Limitations:**

Yes.

**Quality:**

2

**Strengths And Weaknesses:**

Strength:

1. The problem is important and the solutions is interesting
2. The experiments show superior performance
3. The paper is clearly written and easy to follow

Weakness:

1. The theory is incorrect (see the Questions section), which is my biggest concern.

---

> ### Author Rebuttal · Authors · 2025-07-30
>
> Thank you for your thoughtful and critical assessment. Many of your comments will help us produce a more readable and self-contained version of the paper. Below, we address each of your specific concerns in turn.
>
> * **Notation for $\mathbb{E}\_{n}$** . We use $\\mathbb{E}\_{n}$ to denote the empirical average computed over a dataset. To distinguish between different data sources, we let $\\mathbb{E}\_{n_e}$ and $\mathbb{E}\_{n_h}$ represent the empirical averages over the experimental and historical data, respectively.
>
> * **Theory concerns**.
> We sincerely thank the reviewer for the insightful comments regarding the theoretical analysis. Upon revisiting the proof, we agree with the observation and have strengthened Assumption 1 to ensure uniform concentration guarantees. We also clarified the definition of $\text{MSE}(\hat{w})$ by explicitly treating it as a conditional expectation given $\hat{w}$. Based on these revisions, we provide a more rigorous proof of Theorem 1.
>
>   - **Revised Assumption1**. Define the following events:
>     $$
>      \\begin{aligned}
>      A &:= \\left\\{ \\widehat{\\text{bias}}_U(w) \\geq |\\text{bias}(w)| \\text{ for all } w \\in \\mathcal{W} \\right\\}, \\\\
>      B &:= \\left\\{ \\widehat{\\text{Var}}_U(w) \\geq \\text{Var}(w) \\text{ for all } w \\in \\mathcal{W} \\right\\}, \\\\
>      \\end{aligned}
>   $$
>      We assume, for some $\alpha \in (0,1)$, that $\mathbb{P}(A) \geq1 - \alpha$, $\mathbb{P}(B) \geq 1 - \alpha$.
>
>   - **Clarify the definition of $\\text{MSE}(\\hat{w}) .$** We define $\text{MSE}(\hat{w})$ as a conditional expectation—once $\hat{w}$ is fixed, the expectation is taken over the remaining randomness. Thus, $\text{MSE}(\hat{w})$ is itself a random variable. The overall estimator's MSE is: $ \text{MSE}(\widehat{\text{ATE}}(\hat{w})) = \mathbb{E}\left[\text{MSE}(\hat{w})\right], $where the expectation is taken over the randomness of  $ \hat{w} $.
>
>    - **Proof of  Theorem1**. We decompose the MSE difference into two parts based on whether the event $   C := A \\cap B $ holds:
>       $$ \text{MSE}(\widehat{\text{ATE}}(\hat{w})) - \text{MSE}(\widehat{\text{ATE}}(w)) = M_1 + M_2, $$
>
>       where
>       $ M_1 = \mathbb{E} \left[(\text{MSE}(\hat{w}) - \text{MSE}(w)) \cdot \mathbf{1}_{C } \right], $
>
>       and
>      $ M_2 = \mathbb{E}\left[(\text{MSE}(\hat{w}) - \text{MSE}(w)) \cdot \mathbf{1}_{C ^c} \right]. $
>
>      For $M_1$, since $\hat{w}$ minimizes the pessimistic upper bound within event $C$, we have:
>      $$ \mathrm{bias}^2(\hat{w}) + \mathrm{Var}(\hat{w})  \leq  \widehat{\mathrm{bias}}_U^2(\hat{w}) + \widehat{\mathrm{Var}}_U(\hat{w})  \leq  \widehat{\mathrm{bias}}_U^2(w) + \widehat{\mathrm{Var}}_U(w). $$
>
>      Thus,
>      $$ M_1 \leq \mathbb{E} \left[ \left( \widehat{\text{bias}}_U^2(w) - \text{bias}^2(w) + \widehat{\text{Var}}_U(w) - \text{Var}(w) \right) \cdot \mathbf{1}_C \right] \leq \mathbb{E} \left[ \widehat{\text{bias}}_U^2(w) - \text{bias}^2(w) \right] + \mathbb{E} \left[ \widehat{\text{Var}}_U(w) - \text{Var}(w) \right]. $$
>
>      For $M_2$, we apply the triangle inequality and Assumption 2 to get
>      $$ \text{MSE}(\hat{w}) = \mathbb{E}[(\widehat{\text{ATE}}(\hat{w}) - \text{ATE})^2] \leq \mathbb{E}[(|\widehat{\text{ATE}}(\hat{w})| + |\text{ATE}|)^2] \leq \mathcal{O}(B^2). $$
>
>      Similarly, we have $\text{MSE}(w) \leq \mathcal{O}(B^2)$ for any $w \in \mathcal{W}$.
>
>      By the union bound on probabilities
>      $\mathbb{P}(C^{c}) = \mathbb{P}(A^{c} \cup B^{c} ) \leq \mathbb{P}(A^{c}) + \mathbb{P}(B^{c}) \leq 2\alpha.$ Hence, we bound the term $M_2$ as:
>      $$ M_2 = \mathbb{E}\left[(\text{MSE}(\hat{w}) - \text{MSE}(w)) \cdot \mathbf{1}_{C^c} \right] \leq 2B^2 \cdot \mathbb{P}(C^c) \leq \mathcal{O}(\alpha B^2). $$
>
>      Combining both parts yields the final bound
>      $$ \text{MSE}(\widehat{\text{ATE}}(\hat{w})) - \text{MSE}(\widehat{\text{ATE}}(w)) \leq \mathbb{E} \left[ \widehat{\text{bias}}_U^2(w) - \text{bias}^2(w) \right] + \mathbb{E} \left[ \widehat{\text{Var}}_U(w) - \text{Var}(w) \right] + \mathcal{O}(\alpha B^2). $$
>
> * **Statistical inference**. This is an excellent comment, and we fully agree on the importance of conducting valid inference in A/B testing, as A/B testing is essentially a statistical inference problem. Fortunately, when employing DR -- used in our numerical studies -- for ATE estimation, valid p-values can be readily obtained. We have conducted both theoretical analysis and empirical evaluations during the rebuttal to demonstrate this.
>
>    - Specifically, when combined with sample-splitting -- using one half of the data to estimate the weight function $\widehat{w}$ and nuisance functions (the reward and density ratio), and the other half to construct the ATE estimator -- following Chernozhukov et al. (2018), one can show that the resulting ATE estimator is asymptotically normal under suitable regularity conditions (i.e., the reward and density ratio functions converge at certain rates).
>
>   - As a result, standard $z$-tests with normal approximation can be used to obtain valid p-values. The only issue that may invalidate the p-value is the bias introduced through the incorporation of historical data. However, our proposed data integration procedure is explicitly designed to minimize MSE and control this bias. We elaborate on this below.
>
>   - For a given $\widehat{w}$, the bias of the ATE estimator is given by $E(1-\widehat{w}(S^{(e)}))b(S^{(e)})$ where the expectation is taken over $S^{(e)}$. In the first three scenarios considered in our paper, the variance of the estimator dominates $b^2(s)$, so the bias $E(1-\widehat{w}(S^{(e)}))b(S^{(e)})$ is also negligible compared to the variance. This ensures the validity of p-values in these three scenarios. In the last two scenarios, $b^2(s)$ might dominate the variance. However, since our method is designed to minimize the MSE, the estimated weight function $\widehat{w}$ will converge to 1 to ensure the bias $E(1-\widehat{w}(S^{(e)}))b(S^{(e)})$ remains small. Our numerical studies presented below confirm this behavior: the bias is negligible in magnitude, and the resulting p-values remain valid across all scenarios.
>
> * **Additional simulation studies**. For completeness, we conducted additional simulations to evaluate the finite-sample performance of hypothesis testing using the proposed estimator. Specifically, we consider the following hypothesis testing problem:
> $H_0: \text{ATE} = 0 \quad \text{vs.} \quad H_1: \text{ATE} \ne 0$.
>
>   - To assess the performance of the EDO, Pessi, and Proposed methods, we first set the true ATE to 0 and perform 1,000 Monte Carlo simulations to estimate the Type I error rate of each method at the 5\% significance level. Next, to evaluate the empirical power, we gradually increase the signal strength by setting the ATE to values in the set ${0.5,\ 1.0,\ 1.5,\ 2.0,\ 2.5,\ 3.0}$, and again conduct 1,000 simulations for each setting to compute the probability of correctly rejecting the null hypothesis under the alternative. We also examine three levels of bias: small, moderate, and large.
>
>   - The tables below report the empirical rejection rates. Under the null hypothesis, all three methods maintain Type I error rates close to the nominal 5% level, albeit slightly conservative. Under the alternative hypothesis, the proposed method consistently achieves higher power than the alternatives, with power approaching 1 as the signal strength increases.
>
>
>    | ATE | Metric | EDO (Small) | CWE (Small) | Proposed (Small) | EDO (Moderate) | CWE (Moderate) | Proposed (Moderate) | EDO (Large) | CWE (Large) | Proposed (Large) |
>    |:---:|:------:|:-----------:|:-----------:|:----------------:|:--------------:|:---------------:|:-------------------:|:-----------:|:-----------:|:----------------:|
>    | 0.0 | Type-I Error |    0.030    |    0.032    |      **0.024**    |     0.030      |      0.035      |      **0.029**       |    0.030    |    0.030    |     **0.027**     |
>    | 0.5 | Power  |    0.081    |    **0.097**    |       0.093   |     0.081      |      0.103      |      **0.114**       |    0.081    |    0.084    |     **0.088**     |
>    | 1.0 | Power  |    0.267    |    0.310    |      **0.320**    |     0.267      |      0.311      |      **0.335**       |    0.267    |    0.276    |     **0.286**     |
>    | 1.5 | Power  |    0.546    |    0.592    |      **0.611**    |     0.546      |      0.586      |      **0.613**       |    0.546    |    0.558    |     **0.572**     |
>    | 2.0 | Power  |    0.810    |    0.847    |      **0.867**    |     0.810      |      0.835      |      **0.863**       |    0.810    |    0.815    |     **0.825**     |
>    | 2.5 | Power  |    0.955    |    0.967    |      **0.974**    |     0.955      |      0.965      |      **0.970**       |    0.955    |    0.960    |     **0.962**     |
>    | 3.0 | Power  |    0.994    |    0.996    |      **0.997**    |     0.994      |      0.996      |      **0.997**       |    0.994    |    0.994    |     **0.994**     |
> To conclude, we sincerely thank you for your detailed and constructive review. Your comments on the theoretical aspects have guided us in refining the arguments and completing a more rigorous proof framework. We also found your suggestion regarding the potential for statistical inference based on our method to be highly insightful. In response, we have revised the manuscript to incorporate your suggestions and would welcome any further discussion. Thank you again for your thoughtful review.

---

> > ### Comment · Reviewer_XVXr · 2025-08-02
> >
> > Thank you for the great efforts in replying to my comments! For the theory part, how could you *assume* that the uniform bound holds? Are these assumptions provable? Otherwise the theory still cannot go through.

---

> > > ### Author Response · Authors · 2025-08-03
> > >
> > > Many thanks for the follow-up and raising this question. In response, we discuss how to construct the uniform bound below. Since the constructions of the uniform bounds on the bias and variance are very similar, we focus on the derivation for the **bias** below; the bound for the variance can be obtained in a very similar manner.
> > >
> > > Our response is structured as follows: We first present our estimator for the bias. We then describe two approaches for constructing the uniform bound: one based on empirical process theory, which yields a non-asymptotic error bound; and the other based on asymptotic control via a bootstrap procedure proposed by Chernozhukov et al. (2013, 2014).
> > >
> > > 1. **Estimator for the Bias**
> > >
> > >    Recall that the bias takes the form:
> > >
> > >    $$\text{bias}(w) = \mathbb{E}[(1-w(s))b(s)],$$
> > >
> > >    we construct an unbiased estimator:
> > >
> > >     $$
> > >     \widehat{\text{bias}}(w) = \frac{1}{|\mathcal{D}^{(e)}|} \sum_{i=1}^{|\mathcal{D}^{(e)}|} (1-w(S^{(e)}))\psi^{(e)}_0(O^{(e)}\_i) - \frac{1}{|\mathcal{D}^{(h)}|} \sum\_{j=1}^{|\mathcal{D}^{(h)}|} (1-w(S^{(h)}))\psi^{(h)}(O^{(h)}\_j),
> > >     $$
> > >
> > >    where:
> > >
> > >    - $\psi_{0}^{(e)}(O^{(e)}) = \mathbb{I}(A^{(e)}=0) R^{(e)} / \pi(0|S^{(e)})$ is the importance sampling term based on the experimental data.
> > >    - $\psi^{(h)}(O^{(h)}) = \mu(S^{(h)}) R^{(h)}$ is the weighted outcome in historical data, with $\mu(\cdot)$ being the density ratio between $S^{(e)}$ and $S^{(h)}$.
> > >
> > >    Based on this, we aim to construct $\widehat{\text{bias}}_U(w)$ such that:
> > >
> > >    $$
> > >    \mathbb{P}(|\text{bias}(w)| \le \widehat{\text{bias}}_U(w)\ \text{for all } w \in \mathcal{W}) \ge 1 - \alpha,
> > >    $$
> > >
> > >    for some $\alpha > 0$.
> > >
> > > 2. **Empirical Process-Based Bound**
> > >
> > >    Define the following function classes:
> > >
> > >    $$
> > >    \mathcal{F}^{(e)} = \{f_w^{(e)}(o) := (1-w(s))\psi^{(e)}(o): w \in \mathcal{W}\}, \quad
> > >    \mathcal{F}^{(h)} = \{f_w^{(h)}(o) := (1-w(s))\psi^{(h)}(o): w \in \mathcal{W}\}.
> > >    $$
> > >
> > >    Then, $\widehat{\text{bias}}(w) - \text{bias}(w)$ is the difference between two empirical processes indexed by $w$. If the function class $\mathcal{W}$ is of bounded complexity (e.g., VC class), one can apply empirical process theory (van der Vaart and Wellner, 1996) to obtain:
> > >
> > >    $$
> > >    \mathbb{P}\left( \sup_{w \in \mathcal{W}} |\widehat{\text{bias}}(w) - \text{bias}(w)| \le U \right) \ge 1 - \alpha, \tag{1}
> > >    $$
> > >
> > >    or in normalized form:
> > >
> > >    $$
> > >    \mathbb{P}\left( \sup_{w \in \mathcal{W}} \frac{|\widehat{\text{bias}}(w) - \text{bias}(w)|}{\widehat{\sigma}(w)} \le U \right) \ge 1 - \alpha, \tag{2}
> > >    $$
> > >
> > >    where $\widehat{\sigma}(w)$ is a consistent estimator of the asymptotic variance of $\widehat{\text{bias}}(w)$.
> > >
> > >    Accordingly, we can set:
> > >
> > >    - $\widehat{\text{bias}}_U(w) = U + |\widehat{\text{bias}}(w)|$ in the unnormalized case
> > >    - $\widehat{\text{bias}}_U(w) = \widehat{\sigma}(w) U + |\widehat{\text{bias}}(w)|$ in the normalized case
> > >
> > >    Specifically, under sub-Gaussian error terms $\varepsilon^{(e)}$, $\varepsilon^{(h)}$ and VC dimension $v$ for $\mathcal{W}$, we may choose:
> > >
> > >    $$
> > >    U = \frac{c r_{\max}}{\epsilon} \sqrt{\frac{v \log n_{\max} + \log(1/\alpha)}{n_{\min}}},
> > >    $$
> > >
> > >    where $c > 0$ is a constant, $n_{\max} = \max(|\mathcal{D}^{(e)}|, |\mathcal{D}^{(h)}|)$ and $n_{\min} = \min(|\mathcal{D}^{(e)}|, |\mathcal{D}^{(h)}|)$. This can also be extended to heavy-tailed errors (see Chernozhukov et al., 2014, Section 5).
> > >
> > > 3. **Multiplier Bootstrap-Based Bound**
> > >
> > >    Alternatively, one can use the high-dimensional multiplier bootstrap (Chernozhukov et al., 2013, 2014). Under regularity conditions, the distribution of the supremum in equation (2) converges to that of a Gaussian process (Chernozhukov et al., 2014, Theorem 2.1). Thus, $U$ can be set as the $\alpha$-quantile of the supremum, which can be approximated via:
> > >
> > >    $$ \sup\_{w \in \mathcal{W}} \frac{1}{\widehat{\sigma}(w)} \left[
> > >    \frac{1}{|\mathcal{D}^{(e)}|} \sum\_{i=1}^{|\mathcal{D}^{(e)}|} (1 - w(S^{(e)}))\psi\_0^{(e)}(O\_i^{(e)}) g\_i
> > >    -\frac{1}{|\mathcal{D}^{(h)}|} \sum\_{j=1}^{|\mathcal{D}^{(h)}|} (1 - w(S^{(h)}))\psi^{(h)}(O\_j^{(h)}) g\_{|\mathcal{D}^{(e)}| + j}
> > >    \right],   $$
> > >
> > >    where $\{g_i\}$ are i.i.d. standard Gaussian variables.
> > >
> > >    Repeating the process over multiple bootstrap samples provides an empirical estimate of the quantile. Finally, as before, the uniform bound can be defined as:
> > >
> > >    $$\widehat{\text{bias}}_U(w) = \widehat{\sigma}(w) U + |\widehat{\text{bias}}(w)|.$$
> > >
> > > ---
> > >
> > > We hope our responses have adequately addressed your concerns. Please don’t hesitate to reach out if you have any further questions. Finally, we would sincerely appreciate it if you would consider adjusting your score in light of our clarifications.

---

> > > > ### Comment · Reviewer_XVXr · 2025-08-08
> > > >
> > > > Thank you for the follow-up response! While the bounds are asymptotic in nature, I think adding these points would improve the credibility of the theoretical framework. I've raised the score accordingly.

---

> > > > > ### Author Response · Authors · 2025-08-08
> > > > >
> > > > > Thank you for the kind feedback and for raising the score.We’re glad the additional points helped clarify the theory, and we’ll make sure to highlight them in the final version.

---

### Official Review · Reviewer_TPvk · 2025-07-02

**Clarity:** 2
**Significance:** 2
**Originality:** 2
**Rating:** 4
**Confidence:** 3

**Summary:**

This paper investigates the problem of Average Treatment Effect (ATE) estimation in an off-policy evaluation setting by leveraging both historical (observational) and experimental data. The authors propose an estimator that combines estimates derived from each data source via a learned weight parameter. A key aspect of their method is the use of a contextual bandit algorithm to adaptively optimize this mixing weight with the goal of minimizing the Mean Squared Error (MSE) of the final ATE estimate. The performance of the proposed estimator is evaluated on synthetic and semi-synthetic datasets, where it is shown to achieve a low MSE empirically.

**Questions:**

- Could you elaborate on the use of a Doubly Robust (DR) estimator in the experiments when the theoretical analysis is provided for an Importance Sampling (IS) estimator? Do the theoretical guarantees extend to the DR setting?
- I understand that some datasets are proprietary. However, have you considered validating your findings on a publicly available benchmark dataset for policy evaluation?

**Ethical Concerns:**

["NO or VERY MINOR ethics concerns only"]

**Final Justification:**

The rebuttal addressed most of my concerns

I appreciate the authors adding new experiments, however, I believe the claim of "consistently strong performance" is overstated. The new results suggest the method's primary strength lies in the low-to-moderate posterior shift regime, as its advantage diminishes for larger shifts and it is not uniformly superior to all baselines.

**Limitations:**

yes

**Quality:**

2

**Strengths And Weaknesses:**

Strengths:
- Simplicity and Intuition: The proposed technique is simple and intuitive. The concept of a weighted mixture of estimators is straightforward to understand and appears relatively simple to implement.
- Novel Optimization Method: The application of a contextual bandit to learn the optimal mixing weight is a creative and interesting contribution.
- Strong Empirical Results: The experiments, though on synthetic data, demonstrate that the proposed estimator is effective. The reported low MSE suggests that the method holds significant promise.

Weaknesses:
- Incremental noveltry: The general idea of combining estimators from observational and experimental data is well-established in the literature. While the use of a bandit is novel, the paper could benefit from a clearer positioning of its contribution relative to this existing body of work.
- Reproducibility concerns: The semi-synthetic experiments are based on a closed-source dataset, which hinders the reproducibility of those results.
- Gap between theory and practice: There appears to be a disconnect between the theoretical analysis and the empirical implementation. The theory is developed for Importance Sampling (IS) based estimators, while the experiments are conducted using a Doubly Robust (DR) estimator.
- Experimental setting: The evaluation relies only on synthetic and semi-synthetic data. The absence of experiments on a real-world, public benchmark makes it challenging to assess how the method would perform in practical scenarios.

---

> ### Author Rebuttal · Authors · 2025-07-30
>
> Thank you for your thoughtful and critical assessment. Many of your comments will help us produce a more readable and self-contained version of the paper. Below, we address each of your specific concerns in turn.
> * **Contribution relative to existing work on data combination.**
> We thank the reviewer for raising the connection to prior work on combining observational and experimental data—a well-studied topic.
>
>   - Existing methods largely fall into two categories:
>   (1) those addressing **unobserved confounding** in observational data using experimental data (e.g., Kallus et al., 2018; Athey et al., 2020; Bartolomeis et al., 2024; Imbens et al., 2025), and
>   (2) those improving **efficiency under distributional shifts** by combining datasets (e.g., Degtiar & Rose, 2023; Shi et al., 2023; Colnet et al., 2024).
>    - Our method belongs to the latter and focuses on mitigating distribution shift to enhance estimation accuracy.Within this group, existing methods can be further divided into:
>
>      - **Category 1**: Assume **no posterior shift**, and only address covariate and policy shifts (e.g., Li & Luedtke, 2023; Yang et al., 2023; Gao et al., 2025).
>       - **Category 2**: Handle **posterior, covariate, and policy shifts**, often via $\ell_1$-penalized selection (e.g., Xiong et al., 2023; Sheng et al., 2024; Wu et al., 2025; Cheng & Cai, 2021; Han et al., 2023, 2025).
>
>   - Our approach differs from both in three key aspects:
>
>     - **Posterior shift**: Unlike Category 1, we explicitly model and account for it.
>
>     - **Robustness**: Unlike Category 2, which may rely on large posterior shifts, our method is robust to **moderate shifts** and **heavy-tailed data**.
>
>     - **Estimation principle**: We go beyond naive pooling or penalized influence functions by directly **minimizing estimated MSE** via **covariate-adaptive, pessimistic weighting**.
>     - While the idea of data combination is well-established, our contribution lies in its **robust handling of posterior shift**, **adaptive sample-wise weighting**, and a **bandit-inspired framework** suited to real-world heterogeneous settings.
>
> * **Gap Between Theory and Practice**
>
>   First, we emphasize that **Theorem 1 is generic**. As noted (Page 6, Line 234), it applies not only to IS but also to the DR estimator used in our experiments. Details for DR construction are provided in Appendix B.3 to save space. Thus, our theory aligns with our practical implementation.
>   Although the corollaries focus on IS for clarity, our theoretical analysis shows that **all results can be extended to DR**. Below, we outline the necessary assumptions and provide sketches of the main results.
>
>    - **Additional Assumptions for DR**
>
>        - **Assumption 6 (Doubly Robust Specification)**: Either the reward function or the density ratio is correctly specified.
>         - **Assumption 7 (Boundedness of Nuisance Functions)**:
>              - (i) Reward functions are bounded by $r_{\max}$.
>              - (ii) Density ratios are bounded between $\epsilon$ and $\epsilon^{-1}$.
>
>     Assumption 6 ensures consistency of DR estimators and can be replaced by requiring both nuisance functions to converge to oracle targets. Under Assumptions 3 and 5, the oracle functions satisfy Assumption 7, making it reasonable to impose boundedness on their estimates.
>
>     - **Corollaries for DR**
>
>        Let $\delta$ represent the posterior shift magnitude and $\sigma^{(e)}, \sigma^{(h)}$ be reward variances in experimental and historical data.
>
>      - **Corollary 1 (Heavy-tailed Historical Rewards)**
>   DR-EDO remains optimal if:   $ \sigma^{(h)} \gg \epsilon^{-2} \sqrt{\delta} [\sigma^{(e)} + r_{\max}] $  and the MSE difference between DR and EDO is negligible.
>
>     -  **Corollary 2 (Heavy-tailed Experimental Rewards)**
>   DR-HDB remains optimal if:  $\sigma^{(e)} \gg \epsilon^{-2} \sqrt{\delta} [\sigma^{(h)} \delta^{-1/2} + \sqrt{n^{(e)}} r_{\max}] $
>
>     - **Corollary 3 (Small Posterior Shift)**  DR-MVE remains optimal under the same condition as the main text.
>
>     - **Corollary 4 (Moderate/Large Posterior Shift)**
>   DR-EDO remains optimal if:  $ |b(s)| \gg \epsilon^{-2} \left[ |D^{(e)}|^{-1/2} (\sigma^{(e)} + r_{\max}) + |D^{(h)}|^{-1/2} (\sigma^{(h)} + r_{\max}) \right] $
>
>      Compared to IS-based results, the main difference is the dependence on $\epsilon^{-2}$ due to potential density ratio misspecification.
>
>     - **Sketch of Proof**
>
>        The proof follows that of IS but differs in bias and variance control:
>
>    - **Bias**: In IS, correct density ratios are needed. In DR, the bias still equals $\mathbb{E}[b(S^{(e)})]$ under Assumption 6 due to its doubly robust nature (see Chernozhukov et al., 2018).
>
>    - **Variance**: IS is a special case of DR with zero reward function. For general DR, variance bounds still hold under Assumption 7. However, misspecified density ratios yield heavier dependence on $\epsilon^{-1}$, explaining the stronger conditions in DR corollaries.
>
> * **Additional Experiments**.
> We add  experiments on a public real-world dataset.We evaluate our method on the ACTG175 dataset with 2,139 HIV-positive individuals randomized to four treatments. We focus on comparing ZDV+ddI ($n=522$) and ZDV+zal ($n=524$), using rescaled CD4 count as the outcome and three covariates: age ($S_1$, rescaled by 10), homosexual activity ($S_2$), and hemophilia ($S_3$). The outcome model is:
> $$
> R = f(S)=\beta_0 + \beta_1 S_1^2 + \beta_2 S_1 + \beta_3 S_2 + \beta_4 S_3 + \gamma \cdot  A,
> $$ Using fitted model parameters, we generate real-data-based synthetic outcomes:
> $$
> R_e = f(S') +  0.8 \delta \epsilon_e, \quad
> R_h = f(S') + \mu_d + 0.05\mu_d S_1' + 0.8 \delta \epsilon_h,
> $$
> with $\epsilon_e \sim \mathcal{N}(0,1)$ and $\epsilon_h \sim \mathcal{N}(0,1)$ or $t_6$. In experiments, actions are randomized ($A \sim \text{Bernoulli}(0.5)$) in experimental data, while fixed at $A=0$ in historical data. Both datasets have 48 samples. We vary $\boldsymbol{\mu}_d\in [0,5]$ (25 points) and report the $ \textbf{empirical average MSE}$ over 100 simulations. Results $\textbf{stabilize }$for large $\boldsymbol{\mu}_d\$ and are omitted $\textbf{for brevity}$.
>
>    -  **EDO  0.3277 N-N and N-t**
>       | CWE (N-N) | NonPessi | Proposed | LASSO  | CWE (N-t) | NonPessi | Proposed | LASSO  |
>       |:---------:|:--------:|:--------:|:------:|:---------:|:--------:|:--------:|:------:|
>       | 0.3154    | **0.2868** | 0.2912 | 0.3162 | 0.3189    | 0.3014   | **0.3013** | 0.3232 |
>       | 0.3153    | **0.2919** | 0.2927 | 0.3205 | 0.3186    | 0.3013   | **0.3007** | 0.3223 |
>       | 0.3172    | 0.3032   | **0.2997** | 0.3262 | 0.3198    | 0.3056   | **0.3041** | 0.3274 |
>       | 0.3203    | 0.3152   | **0.3088** | 0.3306 | 0.3220    | 0.3147   | **0.3098** | 0.3355 |
>       | 0.3231    | 0.3225   | **0.3164** | 0.3341 | 0.3234    | 0.3196   | **0.3140** | 0.3304 |
>       | 0.3245    | 0.3259   | **0.3210** | 0.3290 | 0.3241    | 0.3213   | **0.3165** | 0.3278 |
>       | 0.3258    | 0.3285   | **0.3246** | 0.3302 | 0.3245    | 0.3231   | **0.3178** | 0.3264 |
>       | 0.3263    | 0.3293   | **0.3252** | 0.3296 | 0.3251    | 0.3236   | **0.3199** | 0.3277 |
>       | 0.3264    | 0.3275   | **0.3248** | 0.3277 | 0.3256    | 0.3243   | **0.3212** | 0.3290 |
>       | 0.3264    | 0.3266   | **0.3241** | 0.3277 | 0.3257    | 0.3241   | **0.3216** | 0.3277 |
>
>
>    We further consider a $t_9$ target distribution with sources either standard normal or $t_9$, increasing $\delta$ to 2.5 for stronger signals.
>
>    - **EDO 1.42 t-N and t-t**
>       | CWE(t-N) | NonPessi | Proposed | LASSO | CWE(t-t | NonPessi | Proposed | LASSO |
>       |:---:|:--------:|:--------:|:-----:|:---:|:--------:|:--------:|:-----:|
>       | 1.38 | 1.29 | **1.28** | 1.41 | 1.39 | 1.32 | **1.30** | 1.41 |
>       | 1.38 | 1.29 | **1.28** | 1.41 | 1.39 | 1.33 | **1.30** | 1.42 |
>       | 1.38 | 1.30 | **1.29** | 1.41 | 1.39 | 1.34 | **1.31** | 1.44 |
>       | 1.39 | 1.31 | **1.30** | 1.41 | 1.39 | 1.33 | **1.32** | 1.43 |
>       | 1.39 | 1.33 | **1.32** | 1.41 | 1.40 | 1.34 | **1.33** | 1.42 |
>       | 1.40 | 1.36 | **1.34** | 1.44 | 1.40 | 1.36 | **1.34** | 1.43 |
>       | 1.41 | 1.38 | **1.37** | 1.45 | 1.41 | 1.38 | **1.36** | 1.43 |
>       | 1.41 | 1.40 | **1.38** | 1.43 | 1.41 | 1.40 | **1.37** | 1.43 |
>       | 1.42 | 1.41 | **1.39** | 1.44 | 1.41 | 1.41 | **1.38** | 1.42 |
>       | 1.42 | 1.42 | **1.40** | 1.44 | 1.42 | 1.42 | **1.40** | 1.42 |
>       | 1.42 | 1.44 | **1.41** | 1.44 | 1.42 | 1.43 | **1.40** | 1.43 |
>       | 1.43 | 1.44 | **1.42** | 1.43 | 1.42 | 1.43 | **1.41** | 1.44 |
> Across all settings and $\mu_d$ values, our method consistently achieves strong performance.
>
> Thank you again for your thoughtful review. We look forward to further discussion.
> * **References**.
> - Athey et al. (2020). Combining experimental and observational data. arXiv.
> - Cheng & Cai (2021). Adaptive combination of RCT and obs. data. arXiv.
> - Colnet et al. (2024). Causal inference combining trials and observational studies. Stat. Sci.
> - De Bartolomeis et al. (2024). Lower bound for confounding strength. AISTATS.
> - Degtiar & Rose (2023). Generalizability and transportability. Annu. Rev. Stat. Appl.
> - Gao et al. (2025). Data-adaptive borrowing in RCTs. Biometrika.
> - Han et al. (2025). Federated causal estimation (FACE). JASA.
> - Han et al. (2023). Multiply robust federated estimation. NeurIPS.
> - Imbens et al. (2025). Long-term causal inference under confounding. JRSSB.
> - Kallus et al. (2018). Removing hidden confounding. NeurIPS.
> - Li & Luedtke (2023). Efficient estimation under data fusion. Biometrika.
> - Sheng et al. (2024). Sequential data integration under shift. Technometrics.
> - Shi et al. (2023). Data integration in causal inference. Wiley Interdiscip. Rev. Comput. Stat.
> - Wu et al. (2025). Estimating ATE via data combination. JASA.
> - Xiong et al. (2023). Federated causal inference in heterogeneous data. Stat. Med.
> - Yang et al. (2023). Elastic integrative analysis. JRSSB.

---

> > ### Comment · Reviewer_TPvk · 2025-08-06
> >
> > Thank you for your comprehensive rebuttal.
> >
> > Your response has resolved most of my concerns
> >
> > I appreciate you adding new experiments, however, I believe the claim of "consistently strong performance" is overstated. The new results suggest the method's primary strength lies in the low-to-moderate posterior shift regime, as its advantage diminishes for larger shifts and it is not uniformly superior to all baselines.
> >
> > The rebuttal has been very effective, and I am raising my score.

---

> > > ### Author Response · Authors · 2025-08-07
> > >
> > > Thank you for your thoughtful follow-up and for raising your score. We sincerely thank you for acknowledging our efforts in addressing your concerns.
> > >
> > > Regarding the claim of "consistently strong performance," we agree that it is important to be precise and avoid overstatement. As you rightly noted, our method demonstrates particular strength in the low-to-moderate posterior shift regime. In the final version, we will revise our wording to better reflect this nuance, clarifying that while our method achieves competitive results across a range of scenarios, its advantage is especially notable in settings with mild to moderate distribution shifts.
> > >
> > > Thank you again for your constructive feedback—it has helped us significantly improve the clarity and rigor of our paper.

---

### Official Review · Reviewer_KjHs · 2025-07-03

**Clarity:** 3
**Significance:** 2
**Originality:** 2
**Rating:** 3
**Confidence:** 2

**Summary:**

The authors study the problem of ATE estimation by combining experimental data (in which a binary treatment is randomized) and observational data observed under the control treatment only. The contribution is a new estimation method that linearly combines experimental and observational estimators with data-dependent weights that are optimized for minimum MSE.

**Questions:**

- Could extensions to the sequential decision-making problem not be done similarly to Li et al. [3]?
- The presentation, distribution shift scenarios, related work, experiments, and theory follow quite closely the work of Li et al. [3]. I understand that the authors define their weights as functions of covariates rather than constant values, is there a theoretical argument to favour one method over the other? Are there scenarios, e.g. low data regimes, in which we should expect Li et al. [3] to outperform?
- How exhaustive are the distribution shift scenarios considered?
- Are there any differences between your Theorem 1 and the Theorem 1 of Li et al. [3]?

**Ethical Concerns:**

["NO or VERY MINOR ethics concerns only"]

**Final Justification:**

I didn't have any particular concerns with the work and most of my questions were addressed during the rebuttal period. I still believe that much of the presentation and results follow closely from Li et al. [3] and it is not clear to me that the contribution significantly improves on previous work, especially since the idea of estimating weights using covariate information is quite well-established and seems like a small improvement upon previously proposed constant weights. That being said, I am not very familiar with the literature and thus my confidence level is low.

**Limitations:**

- There is a disconnect between the presentation of the method and theory with the experimental results, the former developing an importance sampling estimator while the latter implementing a doubly robust estimator. This is a little bit unsatisfying.
- I find that most of the presentation, distribution shift scenarios, related work, experiments, and theory to be very similar to that of Li et al. [3] which in my view detracts from the novelty and originality of the paper.
- There is no conclusion.
- The historical data is limited to observations of control outcomes.

**Paper Formatting Concerns:**

No concerns.

**Quality:**

3

**Strengths And Weaknesses:**

Strengths.
- The proposed method genuinely solves an important problem, it is shown to have oracle properties in many different settings and performs well experimentally.

Weaknesses.
- The conceptual and methodological advancement is quite limited, especially in relation to related work such as Li et al. [3] that already include many of the elements of the proposed recipe.

---

> ### Author Rebuttal · Authors · 2025-07-30
>
> Thank you for your thoughtful and critical assessment. Many of your comments will help us produce a more readable and self-contained version of the paper. Below, we address each of your specific concerns in turn.
> * **Extension to Sequential Setting**
>   Our method naturally extends to sequential decision-making, differing significantly from Li et al. [3]. Given a time horizon $T$, we split the process into $T$ sub-sequences of lengths $1, 2, \ldots, T$, where only the reward at the final step $t$ is retained, preserving the original reward structure.
>    - For each sub-sequence, we build a state-varying double robust estimator using both experimental and historical data. At each time $t$, we compute pessimistic estimates of bias, variance, and account for cross-time correlations, aiming to minimize the sum of pessimistic MSEs over all $T$ steps.
>   - To illustrate the key difference between our method and that of Li et al.[3], we present a simplified mathematical formulation.
> Suppose we have value function estimates at each of the $T$ time steps: $\phi_t(S^{(e)})$ based on experimental data and $\phi_t(S^{(h)})$ based on historical data.
>
>     Our weighted estimators:
>        - For experimental unit data: $\sum_{t=1}^{T} w_t(S^{(e)}) \cdot \phi_t(S^{(e)})$
>        - For historical  unit data: $\sum_{t=1}^{T} \left(1 - w_t(S^{(h)}) \right) \cdot \phi_t(S^{(h)})$
>
>     Li et al.'s fixed-weight estimators:
>        - For experimental unit  data: $w \sum_{t=1}^{T}  \cdot \phi_t(S^{(e)})$
>        - For historical unit data: $(1-w) \sum_{t=1}^{T}  \cdot \phi_t(S^{(h)})$
>
>   - Advantages over Li et al. [3]:
>     - **State-adaptivity**:  Sample weights vary with local state information, reducing MSE.
>
>     - **Time-adaptivity**:   Unlike Li et al. [3], who use fixed weights across time, our method allows time-varying weights. It reduces the MSE by leveraging information from specific moments in the dataset. Even if an estimator has a large overall bias across $T$ moments, certain moments may exhibit smaller biases.
>
>     - **Computational challenges**: Estimating time- and state-varying weights involves handling multi-dimensional state spaces and complex dependencies, while accounting for cross-time correlations adds further computational burden.
>
>
> * **Differences with Li et al. [3]**. We appreciate the reviewer’s observation and clarify key distinctions highlighting our novel contributions beyond Li et al. [3].
>
>   - **Covariate-Adaptive vs. Constant Weights**. A central distinction lies in how the integration weight is defined and optimized.
>     - Li et al. [3] propose a constant weight that linearly combines historical and experimental estimators by minimizing a pessimistic bound on the MSE. This approach performs well in cases of large or zero reward shift, but may not adapt well in intermediate regimes.
>     - Our method extends this framework by allowing the weight to be a function of the covariates (i.e., covariate-adaptive), treating it as a policy in a contextual bandit framework. This design enables local adaptation to distributional heterogeneity, leading to more refined and robust integration.
>
>   - **Theoretical Justification**. Our theoretical analysis (see Theorem 1 and Corollaries 1–4 in our paper) provides rigorous justification for the superiority of covariate-adaptive weighting in several challenging settings.
>
>     - In moderate posterior shift scenarios (which are typically the hardest to detect and address), our covariate-adaptive weighting achieves near-oracle performance, while the constant-weighted estimator may suffer from substantial bias if the fixed weight is not well-tuned.
>
>     - The proposed estimator is adaptive: it recovers the optimal performance of the scenario-specific best estimator without knowing the shift regime a priori. This is not achievable by constant-weighted methods, including Li et al. [3].
>
>     - We show that our estimator retains robustness even when the historical or experimental data are heavy-tailed, or when the posterior shift is moderate and non-identifiable from the data.
>
>   - **Low-data regimes**. We agree that in very low-data regimes, a simple constant-weight approach (e.g., Li et al. [3]) may exhibit lower variance due to reduced model complexity. However, we highlight that
>
>     - Our estimator includes constant weighting as a special case, meaning it can recover the behavior of Li et al. [3] when that is optimal.
>
>     - Moreover, the pessimistic principle we adopt prevents overfitting even in small samples by minimizing an upper bound of the estimated MSE, providing a safeguard in low-data settings.
>
>     - In our experiments, our estimator consistently outperforms or matches the best baseline across all data regimes, including low-sample scenarios.
>
>
>
>
> * **Exhaustiveness of distribution shift scenarios**. This is an excellent comment. As summarized in Section 3.1 of the main text, all potential distributional shifts can be broadly categorized into three types: (i) covariate shift, (ii) policy shift, and (iii) posterior shift. The first two types can be easily handled using importance sampling. In contrast, posterior shift inevitably introduces bias when incorporating historical data -- a challenge we specifically focus on addressing in this paper.
>
>    - Posterior shift primarily concerns changes in the conditional distribution of the reward given a covariate-policy pair. Since this distribution is quite general, and our main interest lies in minimizing the mean squared error (MSE) of the estimator, we restrict attention to its first two moments: the conditional mean and variance of the reward. The conditional mean is captured by the mean reward difference function $b(s)$, while the conditional variances of the historical and experimental rewards are denoted by $(\sigma^{(e)})^2$ and $(\sigma^{(h)})^2$, respectively.
>
>    - By focusing on these two key quantities, all relevant scenarios can be systematically categorized as follows, based on whether one dominates the other or whether they are of comparable magnitude:
>
>      (i) **$\sigma^{(h)}$ deminates both $|b(s)|$ and $\sigma^{(e)})$**: This corresponds to the heavy-tailed historical reward scenario (Scenario (i)) considered in the paper.
>
>      (ii) **$\sigma^{(e)}$ deminates both $|b(s)|$ and $\sigma^{(h)})$**: This corresponds to the heavy-tailed experimental reward scenario (Scenario (ii)).
>
>      (iii) **$\sigma^{(e)}$ and $\sigma^{(h)})$ are comparable and both dominate $|b(s)|$**: This corresponds to the small mean shift scenario (Scenario (iii)).
>
>      (iv) **All three quantities are comparable**: This corresponds to the moderate mean shift scenario (Scenario (iv)).
>
>      (v) **$|b(s)|$ dominates both $\sigma^{(e)}$ and $\sigma^{(h)})$**: This corresponds to the large mean shift scenario (Scenario (v)).
>
>    - Accordingly, all the aforementioned scenarios -- or closely related settings -- are systematically studied in this paper. In summary, our analysis considers all three types of distributional shifts: shifts in the covariate distribution, the policy distribution, and the reward distribution. For the reward distribution, we focus specifically on changes in its first two moments (mean and variance). While this does not exhaustively cover all possible posterior shift scenarios, we believe it captures the most practically relevant scenarios for the purpose of MSE minimization.
> * **Differences from Theorem 1 in Li et al.[3].**
> Li et al. [3] analyze constant weights without pessimism. Their result relies on $\hat{w}$, a data-dependent random variable that can induce high variance. In contrast, our conclusion considers a pessimistic, state-dependent constant weight $w$, which avoids such randomness and reduces variance. Moreover, we explicitly address heavy-tailed settings, which are not covered in Li et al. [3].
>
> * **Disconnect between the method and theory with the experimental results**. We emphasize that the theoretical foundations of importance sampling and doubly robust estimation are closely linked in assumptions and statistical properties. To better align theory with experiments, we have expanded the discussion of the doubly robust framework in our revision, clarifying its relation to our method. This strengthens the connection between our theoretical claims and empirical performance. For further details, please see our response to **Reviewer TPvk** due to character limits.
>
> * **Extension to Historical Treatment & Control Data**
> Our method is initially developed for scenarios where only control outcomes are available in the historical dataset—a common case in A/B testing. It can be naturally extended to more general settings where **both treatment and control outcomes** are observed.
> In this case, we estimate individual treatment effects for historical samples and construct a corresponding estimator. Each experimental unit is assigned a personalized weight $w(S^{(e)})$, while each historical unit receives a complementary weight $1 - w(S^{(h)})$. This enables flexible integration of both datasets in the estimation process and allows us to compute a pessimistic MSE accordingly.
>
> * **Conclusion.** We will include the following conclusion in the final version.
>   - We propose a pessimistic data integration method for causal effect estimation that leverages both experimental and historical datasets. Our method formulates sample-wise weight assignment as a learnable policy. This allows us to apply pessimistic policy optimization to minimize the estimator’s mean squared error .The resulting state-adaptive weights adjust flexibly to individual covariates, improving efficiency and balancing bias-variance tradeoffs. We evaluate the method theoretically and empirically across five scenarios, showing that it achieves near-optimal performance and superior robustness under distribution shift or limited experimental data.
>
> Thank you again for your thoughtful review. We look forward to further discussion.

---

> > ### Comment · Reviewer_KjHs · 2025-08-05
> >
> > Thank you for your rebuttal. It has addressed most of my concerns on extensions to the time-dependent setting and the differences with previous work from Li et al. I still believe that the novelty of top of related work is relatively light so that I don't feel confident to increase my score at this time.

---

> > > ### Author Response · Authors · 2025-08-06
> > >
> > > We sincerely thank the reviewer for acknowledging our efforts in addressing their concerns. Your remaining concern appears to relate to the novelty of our work beyond the existing literature, which we address by clarifying the differences between our work and existing studies below.
> > >
> > > Existing related works can be roughly categorized into two types: (i) methods that correct for unobserved confounding using experimental data, and (ii) methods that improve efficiency under distributional shift. Our work falls into the latter category. The most closely related work is that of Li et al. [3], and **the major difference lies in that we allow our weight function to to vary with the state** (in sequential settings, weights can vary over time).
> > >
> > > Although this may appear to be a subtle change, it substantially enhances the flexibility of the model. In particular, it effectively **enables the use of expressive function classes such as deep neural networks to more accurately capture the importance of each individual sample**, resulting in a substantial reduction in MSE.
> > >
> > > * Theoretically, we establish the optimality of our method across **all five considered scenarios, which exhaustively cover most practically interesting settings**. In contrast, the method proposed by Li et al., which relies on consistent (non-varying) weights, only achieves optimality in cases of **moderate or large mean shifts**.
> > >
> > > * Empirically, our proposed estimator **consistently outperforms existing approaches**.
> > >
> > > We hope this addresses your concern.

---

> > > > ### Author Response · Authors · 2025-08-08
> > > >
> > > > As the discussion phase nears its end, we just wanted to follow up to see if our rebuttal has sufficiently addressed your comments. Please let us know if anything remains unclear — we’d be glad to clarify.
> > > > Thank you again for your thoughtful review.

---

### Note · Authors · 2025-08-14

Thank you to the reviewers and AC for the careful discussion. This remark summarizes what changed, what is clarified, and where the paper’s value is strongest.


**Core contribution & relation to prior work.** Our estimator treats the integration weight as a *covariate-adaptive policy* learned pessimistically to minimize MSE, in contrast to constant-weight schemes (e.g., Li et al.). This adaptive view is the key novelty: it lets the method flex to local heterogeneity and recover the best scenario-specific behavior . We clarified these differences and their implications for sequential settings.

**Theory corrections.** In response to concerns about Theorem 1, we strengthened Assumption 1 to ensure *uniform* concentration over the weight class, clarified the conditioning in the MSE definition, and provided a more rigorous proof sketch. We also outlined two constructions—empirical-process bounds and high-dimensional multiplier bootstrap—to obtain uniform upper bounds for pessimistic bias/variance. Reviewers indicated these additions improved credibility.

**DR vs IS alignment.** We clarified that the main theorem applies beyond IS; the DR implementation used in experiments satisfies analogous assumptions, with explicit DR corollaries. This bridges the earlier gap between theory and practice.

**Scope of shifts & adaptivity.** We organized distribution shift into covariate, policy, and posterior components, focusing on the posterior mean/variance to cover five practically salient regimes. The estimator adapts: it borrows when safe and defaults to EDO in heavy-tailed or large-shift settings—precisely the scenarios where borrowing should be muted.

**Positioning.** We added real-data-based experiments derived from ACTG175 to improve external validity. Reviewers noted that our strongest gains are in *low-to-moderate posterior shift*—we agree and will temper wording accordingly.

**Inference.** We described a sample-splitting DR procedure that yields valid large-sample p-values under standard rate conditions, together with simulations showing well-calibrated type-I error and good power.

**Commitments.** We will move the uniform-bound construction and DR corollaries into the main text and add a clear conclusion.

We appreciate the constructive dialogue; several reviewers raised their scores following these clarifications, and we believe the paper now presents a precise, useful, and careful contribution for policy evaluation under distribution shift.

---

### Decision · Program_Chairs · 2025-09-17

**Decision:**

Accept (poster)

**Comment:**

This paper introduces a novel weighted Average Treatment Effect (ATE) estimator designed to integrate experimental data (in which a binary treatment is randomized) with observational data (collected from a control treatment only). The key contribution of this work is a new technique that linearly combines experimental and observational estimators using data-dependent weights optimized to minimize MSE. The authors evaluated the empirical performance of the proposed method in synthetic and semi-synthetic datasets, with results indicating that the proposed estimator achieves lower MSE than competitors.

---
All reviewers agreed that this paper tackles an important and challenging problem. They found the proposed method to be simple, intuitive, and novel, highlighting, for example, that the use of contextual bandits to optimize mixing weights is an interesting and unexplored direction in the literature. The reviewers also praised the strong empirical results, with the proposed estimator achieving superior performance compared to baselines.

---
During the discussion phase, reviewers raised a number of concerns. Most were relatively minor and were adequately addressed by the authors in their thorough rebuttal. That said, two central concerns remained and were the primary focus of the discussion. First, two reviewers commented on the perceived limited novelty of this method relative to the one introduced by Li et al. [3]. The authors provided a detailed response, clearly explaining both similarities and the ways in which their approach extends Li et al.’s. The proposed estimator, for instance, dynamically determines how to integrate data through a learned covariate-adaptive policy explicitly optimized to minimize MSE. This contrasts with the constant-weight scheme leveraged by Li et al., which merely minimizes a pessimistic bound on MSE. In addition, the proposed approach extends Li et al.’s framework by enabling the estimator to adapt locally to distributional heterogeneity. Achieving this capability required substantial new mathematical analysis, which resulted in a technique capable of improving efficiency under distributional shift through a scheme that adapts combination weights as a function of the state and time—a key capability for addressing sequential settings. Finally, the authors emphasized in the rebuttal that their estimator is optimal across five key scenarios, thus covering a wide range of practically relevant cases. Taken together, these distinctions support the paper’s theoretical and methodological novelty.

The second major concern raised during the discussion phase related to the correctness of one of the paper’s theoretical results—in particular, Theorem 1. The authors fully addressed this point by strengthening Assumption 1 to ensure uniform concentration over the weight class. They also provided a more rigorous proof sketch and outlined two construction strategies for obtaining uniform upper bounds on bias and variance.

---
Overall, the reviewer–author exchanges were constructive, and the authors appeared to provide strong and clear responses to all key concerns raised. Post-rebuttal, the reviewers indicated that their main questions were resolved. The reviewers encouraged the authors to revise the manuscript to address all major points raised during the discussion and to incorporate key requested clarifications (for instance, by more explicitly discussing how this work’s Theorem 1 differs from the one introduced by Li et al.).